# Investigating the radiative effect of Arctic cirrus measured in situ during the winter 2015/2016

Andreas Marsing[1], Ralf Meerkötter[1], Romy Heller[1], Stefan Kaufmann[1], Tina Jurkat-Witschas[1], Martina Krämer[2,3], Christian Rolf[2], and Christiane Voigt[1,3]

[1]Institute of Atmospheric Physics, German Aerospace Center (DLR), Oberpfaffenhofen, Germany
[2]Institute for Energy and Climate Research (IEK-7), Research Center Jülich, Jülich, Germany
[3]Institute for Atmospheric Physics, Johannes Gutenberg University, Mainz, Germany

**Correspondence:** Andreas Marsing (andreas.marsing@dlr.de)

**Abstract.** The radiative energy budget in the Arctic undergoes a rapid transformation compared to global mean changes. Understanding the role of cirrus in this system is vital, as they interact with short- and long-wave radiation and the presence of cirrus can be decisive as to a net gain or loss of radiative energy in the polar atmosphere.

In an effort to derive radiative properties of cirrus in a real scenario in this sensitive region, we use in situ measurements of ice water content (IWC) performed during the POLSTRACC aircraft campaign in the boreal winter and spring 2015/2016 employing the German research aircraft HALO. A large dataset of IWC measurements of mostly thin cirrus at high northern latitudes was collected in the upper troposphere and also frequently in the lowermost stratosphere. From this dataset we select vertical profiles that sampled the complete vertical extent of cirrus cloud layers. These profiles exhibit a vertical IWC structure that will be shown to control the instantaneous radiative effect both in the long and short wavelength regimes in the polar winter.

We perform radiative transfer calculations with the UVSPEC model from the libRadtran program package in a one-dimensional column between the surface and the top of the atmosphere (TOA), taking as input the IWC profiles, as well as the state of the atmospheric column at the time of measurement, as given by weather forecast products. In parameter studies, we vary the surface albedo and solar zenith angle in ranges typical for the Arctic region. We find the strongest (positive) radiative forcing up to about $48\,\mathrm{W\,m^{-2}}$ of cirrus over bright snow, whereas the forcing is mostly weaker and even ambiguous, with a rather symmetric range of values down to $-35\,\mathrm{W\,m^{-2}}$, over the open ocean in winter and spring. The IWC structure over several kilometres in the vertical affects the irradiance at the TOA through the distribution of optical thickness. We show in how far coarser vertically resolved IWC profiles can reflect this effect. Further, a highly variable heating rate profile within the cloud is found which drives dynamical processes and contributes to the thermal stratification at the tropopause.

Our case studies highlight the importance of a detailed resolution of cirrus clouds and consideration of surface albedo for estimations of the radiative energy budget in the Arctic.

# 1 Introduction

Cirrus clouds are ice clouds that exist at temperatures below the homogeneous freezing threshold of about 235 K. The required temperatures and humidity conditions are found in the upper troposphere and sometimes lowermost stratosphere, leading to an average global coverage of at least 20 %. Cirrus clouds form in ice super-saturated regions in the presence of suitable nuclei that may be aerosol of different kind, solid or liquid, or by freezing of (supercooled) liquid droplets in ascending air masses (Heymsfield et al., 2017).

The impact of cirrus on the radiative energy budget of the atmosphere can be differentiated by direct and indirect effects. The direct effect is absorption and re-emission of thermal (longwave) radiation and scattering of solar (shortwave) radiation. Indirect or secondary effects arise from the redistribution of water vapour and through heterogeneous physical (trapping, release, e.g. Kärcher and Voigt, 2006; Kärcher et al., 2009; Voigt et al., 2006) and chemical reactions with radiatively active compounds, such as ozone-depleting substances (e.g. von Hobe et al., 2011; Solomon et al., 1997; Borrmann et al., 1996).

Cirrus occurrence and radiative properties are prone to anthropogenic influence through the introduction of ice nucleating particles and precursors (Hoose and Möhler, 2012), including aircraft emissions (Kärcher, 2018), and through the evolution of atmospheric humidity as a feedback to changing climate as a whole. It is known that the microphysical properties of cirrus, such as particle number concentration, size distribution and shape, depend on the nuclei and the environmental conditions during nucleation, such as updraft speeds (e.g. Kärcher and Jensen, 2017; Joos et al., 2014; Krämer et al., 2020).

Cirrus are more abundant and thicker in the tropics and mid-latitudes, but there is also considerable coverage in the high latitude and polar regions. There, the low elevation of the sun and complete absence of sunlight in polar nights reduce the shortwave effect (Hong and Liu, 2015). In addition, the Arctic surface climate currently undergoes a much more rapid warming transition than the rest of the globe, known as Arctic amplification, which also changes the spatial and temporal patterns of the surface albedo as sea ice and continental ice retreat (Wendisch et al., 2017; Shupe et al., 2022). Therefore the question has not yet been answered satisfactorily how Arctic cirrus impact the radiation budget now and in the future. Further it is unknown in which direction and extent, and under which circumstances the cirrus feed back to the current warming.

Although cirrus are regularly observed from space with active (e.g. Sassen et al., 2008; Pitts et al., 2018) and passive (e.g. Strandgren et al., 2017) sensors, in situ measurements are needed for the closure of the physical appearance of cirrus ice crystals and their bulk radiative properties (Thornberry et al., 2017; Krisna et al., 2018; Ewald et al., 2021). There have been some in situ studies on Arctic cirrus clouds in the past: the Polar Stratospheric Aerosol Experiment (POLSTAR) missions 1997/1998 (Schiller et al., 1999), the European Polar Stratospheric Cloud and Lee Wave Experiment (EUPLEX) in January-February 2003 and the ENVISAT validation experiments in October 2002/March 2003, with measurements of ice water content (IWC) up to 68° N (summary in Schiller et al., 2008); the Stratospheric Aerosol and Gas Experiment (SAGE) III Ozone Loss and Validation Experiment (SOLVE) in 1999–2000 with measurements of IWC and particle size distribution (Schiller et al., 2002; Hallar et al., 2004). The mentioned missions accumulate to a total of 15 flights and contributed a significant part to our current picture of cirrus in the Arctic. Nevertheless these are few studies compared to much more comprehensive observations in the tropics and at mid latitudes (e.g. Voigt et al., 2007; Thornberry et al., 2017; Heymsfield et al., 2017; Krämer et al., 2016, 2020).

An earlier study by Feofilov et al. (2015) uses spaceborne radar, lidar and radiance measurements to derive the radiative impact at the top of the atmosphere (TOA) of differently shaped IWC profiles including global variations. While we also examine this effect for the inhomogeneous vertical IWC distributions in our measurements, in contrast, our study focuses on high latitude cirrus and we demonstrate the radiative effects, including heating rate, in the entire atmospheric column. This also constitutes a link to cloud dynamics modelling studies that investigate the interaction of heating rate profiles with ice supersaturation, vertical velocity and particle nucleation and growth (Fusina et al., 2007; Fusina and Spichtinger, 2010). Bucholtz et al. (2010) present directly measured heating rates in thin tropical cirrus and highlight the importance of further observations to improve the understanding of the radiative effects of cirrus.

The present work introduces a novel dataset of in situ IWC measurements carried out during the Polar Stratosphere in a Changing Climate (POLSTRACC) campaign employing the German High Altitude and Long Range research aircraft HALO (Oelhaf et al., 2019). The campaign was based in Kiruna, Sweden (67.5° N, 20.3° E) and spanned the boreal winter/spring season from 17 December 2015 to 18 March 2016 with 18 science flights and 156 flight hours. The campaign accommodated a number of different science goals, including trace gas composition in the upper troposphere and lower stratosphere (Marsing et al., 2019; Johansson et al., 2019; Braun et al., 2019; Keber et al., 2020; Ziereis et al., 2022), gravity waves (Krisch et al., 2020) and polar stratospheric clouds (Voigt et al., 2018). Therefore, the flight strategy often needed to avoid extended sections in cirrus clouds. Nevertheless, a total of 7.2 h or 5600 km inside cirrus during 44 individual events were recorded.

The flights included several climb or descent profiles through cirrus clouds, predominantly located just below the thermal tropopause. In this study, we take advantage of these IWC profiles and embed them in the corresponding atmospheric column of temperature and trace gases from European Centre for Medium-Range Weather Forecasts (ECMWF) model analysis. We investigate the location of cirrus in relation to the local tropopause and the characteristic of a tropopause inversion layer (TIL) above. In a second step, this enables radiative transfer calculations that help understand the effect of these high latitude ice clouds on the atmospheric radiation budget as a whole and on the heating rate profile inside the cloud layer in detail. This constitutes the link between realistic profiles (as they are directly measured) and their impact on local dynamics and radiative energy budget.

The manuscript is laid out as follows: Section 2 contains the experimental acquisition of in situ IWC data. An overview and statistical characteristics of sampled Arctic cirrus clouds are given in Sec. 3, whereas Sec. 4 breaks the observations down to the detailed profiles. Section 5 elaborates on the radiative transfer calculations. Section 6 contains the discussion of and conclusions from the material. Section 7 closes with a short outlook.

## 2 In situ measurements of IWC

IWC is a derived quantity from our bulk measurements of the total atmospheric water content (TWC), which consists of the gaseous (vapour) phase and condensed phases (liquid, ice). In the present closed-path principle, a sample air flow was led into the aircraft cabin and heated, in order to evaporate all water content. Then, the TWC measurement was performed with the WAter vapoR ANalyzer (WARAN) laser hygrometer. It derives the concentration of water vapour in the sample flow

by using the absorption of the 1.37 µm line from a InGaAs tunable diode laser (TDL) in a closed measurement cell (Voigt et al., 2014). The sensor is the commercial WVSS-II by SpectraSensors Inc. with a modified stainless steel inlet line and connected to a forward-facing Trace Gas Inlet (TGI) with 9.55 mm diameter at 31.9 cm distance from the fuselage of HALO. The measurement range is 50–10000 ppmv, with a precision of 5 % or 50 ppmv, whichever is greater. Values are sampled at a frequency of 0.3–0.4 Hz.

Whereas the instrument design and data treatment compensate for drifts due to changing environmental conditions, such as temperature and pressure (Heller, 2018), slow degradation requires repeated calibration within months. A calibration for the POLSTRACC campaign was performed afterwards on 28 July 2016 using a gauged dew point mirror (MBW 373-LX). The result is a linear correction function

$$H_2O_{calibrated} = a + b \cdot H_2O_{WARAN}, \tag{1}$$

where $H_2O_{WARAN}$ denotes the output water vapour mixing ratio from WARAN and $H_2O_{corrected}$ is the calibrated mixing ratio, both in ppmv. The calibration coefficients $a$ and $b$ are determined piecewise:

$$
\begin{aligned}
a &= 3.756\,\text{ppmv}, \quad b = 1.045 \quad \text{for} \quad H_2O_{WARAN} \leq 70\,\text{ppmv}, \\
a &= 12.619\,\text{ppmv}, \quad b = 0.944 \quad \text{for} \quad H_2O_{WARAN} > 70\,\text{ppmv}.
\end{aligned}
\tag{2}
$$

In cirrus clouds that contain only ice, the ice water content (IWC) can be calculated by subtracting the gas phase water content (GWC) from the measurement. Due to a lack of an independent GWC measurement in clouds during the POLSTRACC campaign, the saturation mixing ratio is used as a first order approximation, calculated from the Clausius-Clapeyron equation, as in Heller (2018). In reality, the relative humidity with respect to ice (RH$_i$) inside cirrus mostly assumes values between 80 – 120 %. Stronger sub- and particularly supersaturations, up to the homogeneous freezing threshold (about 150 – 165 %), are possible wherever particle growth rates are too slow to mitigate the excess relative humidity (Krämer et al., 2020). Such pronounced supersaturation has an increased probability at temperatures below 200 K, where cirrus are often characterised by initially low particle number concentrations (Krämer et al., 2020). High supersaturation is strongly correlated with high updraft speeds, which eventually trigger homogeneous nucleation (Krämer et al., 2009) with a rapid increase in particle number concentration (Mitchell et al., 2018). Updraft speeds for the selected cases in the sections below reach rather high values up to about 0.5 – 1 m s$^{-1}$. This increases the expected quasi steady-state relative humidity (Krämer et al., 2009) in the relevant temperature range to up to 120 – 130 %. Krämer et al. (2009) note that at the same time, relaxation times towards this state are reduced at higher updrafts from tens to a few minutes, so supersaturation depends strongly on the timing of the observation within the cirrus lifecycle. This natural climatological variability propagates into the uncertainty of calculated IWC, but the effect is strongly mitigated by the particle sampling characteristics explained in the following:

Sub-isokinetic sampling due to a lower inlet flow velocity (around 3 m s$^{-1}$) compared the speed of airflow around the fuselage (typically around 230 – 240 m s$^{-1}$) leads to an inertial enhancement of particles in the inlet flow. Therefore, a correcting "enhancement factor" (EF) needs to be applied to the IWC measurement. Following the rationale in Heller (2018), the EF for typical high latitude cirrus, where significant contributions to the IWC are contained in particles with radii larger than 20 µm

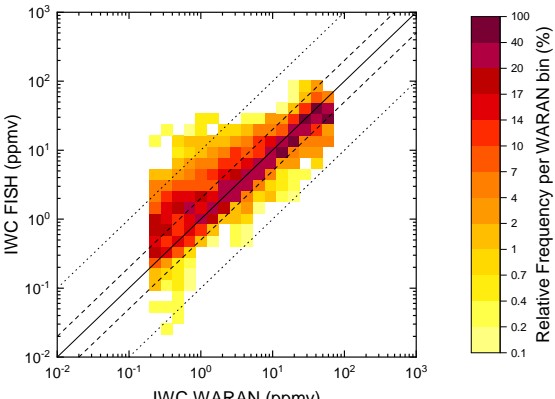

**Figure 1.** Comparison of ice water content (IWC) from the WARAN and FISH instruments for cirrus measurements during the POLSTRACC campaign. The shading indicates the distribution of data within each WARAN bin. Note the double logarithmic axes and the logarithmic colour scale. The solid line represents the 1:1 identity. The dashed lines denote a deviation of factor 2, and the dotted lines a deviation of factor 10. Accounting for the detection limit, WARAN data are clipped below $0.2$ ppmv.

(Wolf et al., 2018), can be calculated as

$$EF = \frac{u_a}{u_0} \tag{3}$$

with the known airspeed $u_a$ and inlet velocity $u_0$ (Schiller et al., 1999). This approximation is independent of the precise particle size distribution and takes values of about 70 to 80 for the POLSTRACC measurements. Such high EF values effectively lower
125 the detection limit of the corrected IWC to below $1$ ppmv. The error in calculating the EF is $\pm 5\,\%$, leading to a precision of $10\,\%$ for IWC measurements. With this enhancement of IWC over GWC, if we were to assume a hypothetical supersaturation with $RH_i = 160\,\%$ instead of $100\,\%$, the IWC in the cases below would be reduced by a factor of 0.002–0.03 on 25 January 2016 (0.04–0.14 on 09 March 2016). This is the uncertainty associated with the saturation assumption made above, ranging from negligible within the instrument errors to about $14\,\%$ in case of maximum supersaturation.

130    Another major uncertainty in airborne IWC measurements arises from further enhancement or depletion of particles of different sizes in the flow around the aircraft until the sampling position. Afchine et al. (2018) studied this effect in detail, including data from the HALO mission ML-CIRRUS (Voigt et al., 2017), and found particular ramifications for cabin instruments with inlet positions at the fuselage. Afchine et al. (2018) found that the IWC from WARAN was mostly systematically higher than the IWC from the Fast In-situ Stratospheric Hygrometer (FISH, Meyer et al., 2015), a Lyman-$\alpha$ hygrometer that
135 samples total water in a similar way, but at a different position. This comparison is repeated with the POLSTRACC data in Fig. 1. Here, a systematic deviation is not obvious, as the correlation between both instruments is well centred on the 1:1 line and there are no pronounced off-centred branches. More than $50\,\%$ of the measurements coincide within a factor 2 above $1$ ppmv. In a linear regression, the coefficient of determination $R^2 = 0.73$ is smaller compared to Afchine et al. (2018), but the intercept

$y_0 = -0.040$ is closer to zero and the slope $m = 0.952$ closer to identity. This enhanced comparability is attributed to improved calibration and a precise synchronisation of data streams. It also shows that the different positions of both inlets, which are about 3 m apart along the length of the fuselage, have a less pronounced impact on the observable IWC than previously thought. Nevertheless, we need to clarify that the IWC measurements from both instruments through roof-mounted TGIs are strongly affected by enrichment or loss of ice particles due to their inertia in the flow around the aircraft's fuselage. Afchine et al. (2018) could quantify this effect for the HALO aircraft with comparing IWC and particle size distribution measurements at the much less affected wing probe position, where IWC deviations by up to one order of magnitude were observed at a significant frequency. This flaw cannot be easily overcome for the POLSTRACC data without knowledge about the particle size distribution. In the study of the impact on radiation of the observed IWC distributions in Sec. 5, we therefore address the effects of underestimated IWC sampling. There, we test the sensitivity of our results to an underestimated IWC by re-evaluating all calculations with IWC profiles multiplied by a factor of 5, which corresponds to the average deviation seen for the FISH instrument in the worst particle size regime above 25 μm (Afchine et al., 2018).

## 3 Statistics of Arctic cirrus sampling

The POLSTRACC mission accommodated several scientific objectives that made use of the high ceiling altitude, long range and heavy payload capabilities of the HALO research aircraft (Oelhaf et al., 2019). For the observation of trace gas composition and gravity waves, the flight strategy often needed to avoid extended in-cloud sections. Therefore the data set represents a relatively random sampling of cirrus at the middle to high latitudes. Nevertheless, the 18 science flights spanning 156 flight hours yielded an accumulated residence time of 7.2 h inside cirrus clouds during 44 individual events, as calculated from the number of WARAN IWC measurements at temperatures below 235 K.

Figure 2 shows the IWC measurements as coloured sections of the POLSTRACC flight tracks. It can be seen that there are many shorter sections around the landing sites during the campaign: Oberpfaffenhofen, Germany (EDMO); Kiruna, Sweden (ESNQ); Kangerlussuaq, Greenland (BGSF). Cloud layers there were often crossed during approaches and departures. Some more extended sections in mid-flight are found intermittently. The colour coding shows the range of temperatures during these cirrus encounters. The minimum temperature recording during an IWC event was 195.8 K.

For the IWC as an accessible bulk property, previous studies set up climatologies that relate the IWC to temperature (Schiller et al., 2008; Krämer et al., 2016) and other properties such as effective radius and number concentration (e.g. Krämer et al., 2020; Luebke et al., 2013; Liou et al., 2008), and broke them down to different latitudes on Earth and different cloud formation pathways. Schiller et al. (2008) and later Luebke et al. (2013) collected the range of airborne cirrus measurements from different campaigns as a function of latitude and temperature. In Fig. 3 the POLSTRACC measurements are added to this scheme found in Luebke et al. (2013). The broad latitudinal range of the POLSTRACC measurements can be seen. While many domains have been covered by previous missions, the figure shows that the new data also extend into gaps that were not filled before. This concerns in particular the lower temperatures below 210 K down to 198 K at the mid latitudes between 45 to 60° N. A 4 h fraction of the campaign data was sampled north of 60° N and a significant extension to existing data could be made at

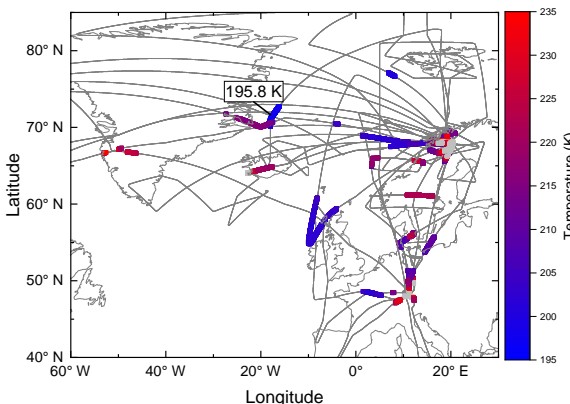

**Figure 2.** Map of the North Atlantic showing the flight tracks of the POLSTRACC campaign. Coloured sections denote IWC measurements from the WARAN instrument. The colour coding indicates the temperature during these sections, where red is warmer than blue, and grey colors are above 235 K.

$65°$ N over the whole temperature range. Measurements north of $70°$ N are generally sparse, and POLSTRACC can extend the coverage slightly for the cold cirrus below 200 K. Still there is much room left open temperaturewise, but also in the latitudinal coverage between $70°$ N and the pole.

Since the creation of the original figure by Luebke et al. (2013), the POLSTRACC observations represent the only addition of airborne in situ data at northern polar latitudes. Further observations not included in this figure comprise for example the already mentioned ML-CIRRUS campaign in mid-latitudes (Voigt et al., 2017).

This study focuses on cirrus near the tropopause. We are going to examine the distribution of cirrus sampling along different tropopause-related vertical coordinates, which helps to assess the role of geometrical location and extent, as well as temperature, in the cloud radiative effect, which at the tropopause becomes especially relevant in view of thermal stratification. Also different formation pathways and ice particle properties are expected for tropospheric and stratospheric cirrus. The following discussion is limited to observations north of $60°$ N. Figure 4a displays the cirrus observations in two vertical scales: the abscissa indicates the potential temperature difference $d_{TP}$ of the observation location to the 2 PVU (potential vorticity unit, $1 \, \mathrm{PVU} = 10^{-6} \, \mathrm{K \, m^2 \, kg^{-1} \, s^{-1}}$) isosurface of potential vorticity (often denoted as dynamical tropopause), which is typically located at a potential temperature of 300 to 320 K (but further $\pm 10$ K are possible). The ordinate axis indicates the squared Brunt-Väisälä frequency $N^2$ as a measure for the static stability of the atmospheric thermal stratification. The colour coding indicates the temperature as in Fig. 2. Close to the tropopause, roughly linear correlations between $d_{TP}$ and $N^2$ can be observed, connecting the free, statically less stable troposphere with $N^2 < 2 \cdot 10^{-4} \, \mathrm{s^{-2}}$ to the stratosphere with $N^2 > 4 \cdot 10^{-4} \, \mathrm{s^{-2}}$. This correlation is not compact but spans a corridor with a width of 7 to 10 K. The lower-right branch belongs to January mid-winter measurements, whereas the upper-left branch stems from the March late-winter measurements. This may be interpreted as an

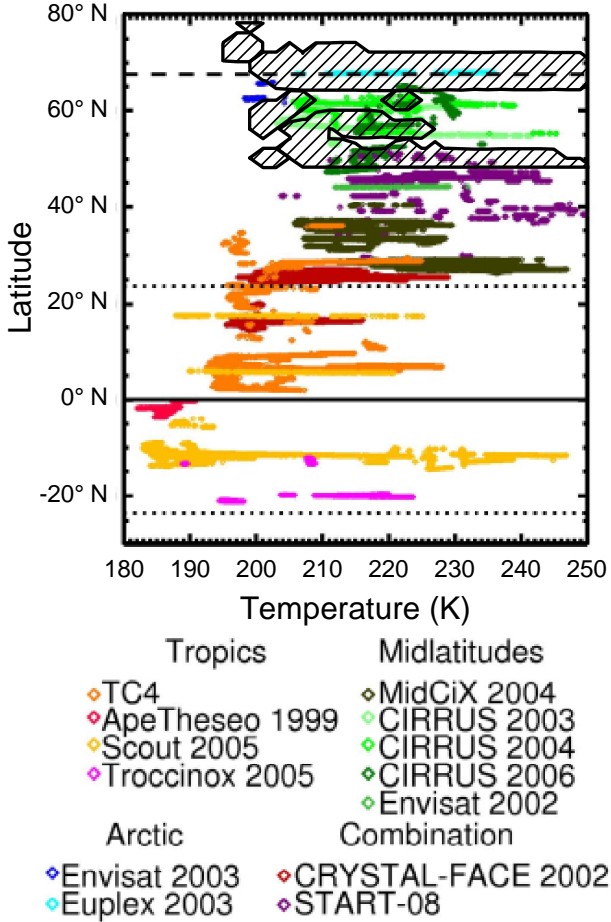

**Figure 3.** The black contour and diagonal stripes sketch the coverage of cirrus measurements during POLSTRACC with respect to latitude and temperature. For this work, the original of this figure from Luebke et al. (2013) is adapted, showing the coverage of literature airborne in situ cirrus missions.

increasing stabilization of the tropopause region within two months due to the steady descending motion in the stratosphere (Manney and Lawrence, 2016; Birner, 2010). The blue colours highlight the occurrence of cold cirrus at $T \lesssim 200\,\mathrm{K}$ above the dynamical tropopause. These cirrus clouds on the stratospheric side have in most cases no continuation in the troposphere. They build up a lower branch of polar stratospheric ice clouds (PSCs), and do not rely on tropospheric water vapour supply. Ice nucleation, particle size and habit as well as chemical constitution differ significantly from the tropospheric tropopause cirrus and they play a crucial role in heterogeneous chemistry in the lowermost stratosphere, impacting especially the budgets of ozone-depleting substances. Panel b shows the distribution of measurements with respect to $N^2$. Two major accumulation regions can be identified, one at 2 to $3 \cdot 10^{-4}\,\mathrm{s}^{-2}$, and another at 4 to $5 \cdot 10^{-4}\,\mathrm{s}^{-2}$. The former can be attributed to tropospheric


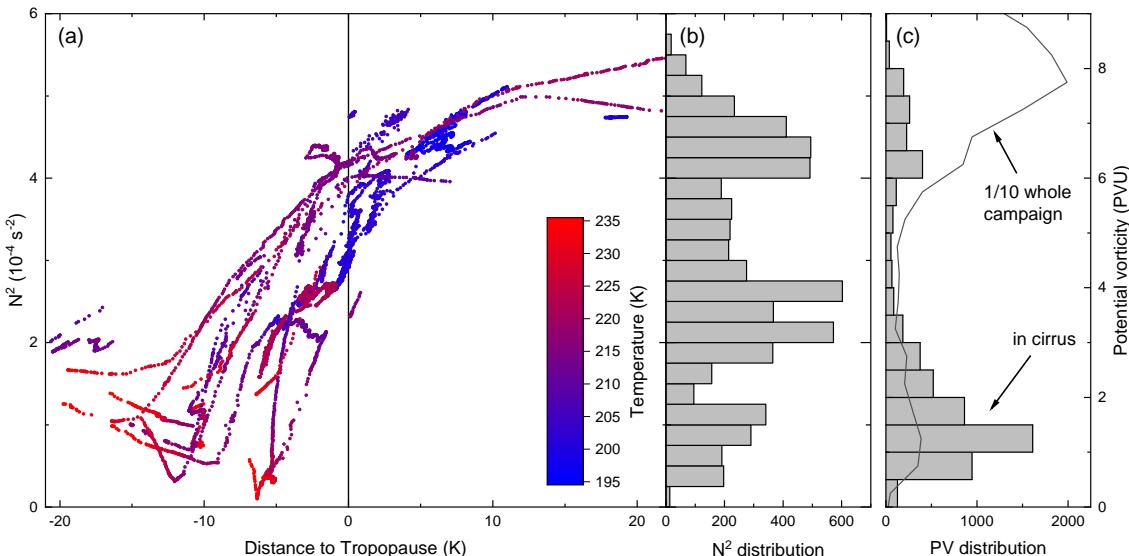

**Figure 4.** Statistics on the distribution of cirrus sampling below 235 K during POLSTRACC at high latitudes north of 60° N. (a) Distribution over distance to tropopause (given in K of potential temperature) and squared Brunt-Väisälä frequency $N^2$. The temperature is indicated in the colour coding. (b) Marginal distribution in $N^2$. (c) Distribution in potential vorticity. Cirrus sampling in grey bars, and the total flight campaign flight time (divided by 10) in the black line.

cirrus extending up until the thermal tropopause (according to the definition by the World Meteorological Organization, 1957)
and the latter to stratospheric cirrus with no or only a weak connection to the troposphere.

The distribution of cirrus observations can also be related directly to potential vorticity (PV) as seen in the bars in Fig. 4c. As already indicated above, a PV of 2 PVU is usually chosen as the dynamical tropopause which follows other tropopause definitions adequately in areas of weak horizontal wind shear, although located systematically below the thermal tropopause (Gettelman et al., 2011). For the extratropics a value of 3.5 PVU is often more consistent with the thermal tropopause (Hoinka,
1997). In the figure, the low counts in the 0 to 0.5 PVU range may suggest a low cloud occurrence in the mid troposphere. This is not necessarily the case and may be a result of the artificial limitation of the data to temperatures below 235 K, which has been applied in order to exclude false hits from mixed-phase clouds. So certainly more clouds were encountered there, composed of either pure ice, mixed phases or liquid water. The highest cirrus occurrence is found below the 2 PVU isosurface with a maximum at 1 to 1.5 PVU, which is unambiguously located in the troposphere, where the available water vapour
content is still at least one order of magnitude above stratospheric values. Above, cirrus sampling steadily decreases towards a rather uniform distribution above the 3.5 PVU isosurface. This reflects a random distribution of cirrus observations in the stratosphere, where PV varies strongly in the vertical direction and no single level is preferred for cirrus occurrence. Among the generally low IWC sampling in the stratosphere, the apparent rise between 6 to 8 PVU is most likely due to the overall distribution of flight time in the stratosphere during the campaign, as can be seen by the black line in Fig. 4c.

**Table 1.** List of complete cirrus profiles sampled during the POLSTRACC aircraft campaign. The given latitude is an average value for the profile. The acronyms ML/HL are used to categorise the profiles as mid/high latitude profiles. $z$ denotes the vertical span, $T$ is the temperature range and the maximum IWC (in molar mixing ratio and in mass concentration) is read from the WARAN measurements. The ice water path (IWP) is calculated by integrating the IWC measurements over the whole profile.

| Date | Time (UTC) | Lat. (°N) | $z$ (km) | $T$ (K) | max. IWC (ppmv) / (mg m$^{-3}$) | IWP (g m$^{-2}$) |
|---|---|---|---|---|---|---|
| 13.12.2015 | 8:12 – 8:33 | 50.2 (ML) | 8.8 – 12.6 | 205 – 230 | 11 / 3.2 | 1.2 |
| 13.12.2015 | 10:32 – 10:33 | 51.2 (ML) | 11.3 – 11.8 | 206 – 208 | 1 / 0.3 | 0.04 |
| 25.01.2016 | 12:39 – 13:09 | 67.8 (HL) | 7.3 – 12.0 | 201 – 235 | 43 / 9.9 | 9.2 |
| 31.01.2016 | 14:29 – 14:41 | 66.1 (HL) | 3.9 – 8.8 | 206 – 248 | 10 / 3.4 | 4.3 |
| 06.03.2016 | 5:15 – 5:23 | 67.8 (HL) | 6.8 – 9.7 | 211 – 230 | 1 / 0.4 | 0.3 |
| 09.03.2016 | 14:48 – 14:51 | 67.2 (HL) | 5.9 – 7.9 | 219 – 233 | 51 / 17.9 | 10.2 |
| 18.03.2016 | 11:09 – 11:14 | 55.0 (ML) | 10.0 – 11.6 | 206 – 217 | 9 / 2.1 | 0.8 |
| 18.03.2016 | 17:20 – 17:30 | 50.4 (ML) | 9.6 – 11.1 | 208 – 218 | 26 / 6.3 | 1.5 |

The following Secs. 4 and 5 focus on one subclass of IWC measurements that represent complete vertical profiles of cirrus clouds at the tropopause, seen as line-forming dots of data in Fig. 4a.

## 4   Complete profiles of cirrus at the high latitude tropopause

In order to investigate the radiative impact of cirrus at the tropopause on the profile of temperature and stability, it is necessary to include the complete profile of the respective cloud in the analysis. The POLSTRACC IWC dataset has been filtered to this
end and a selection of eight events was found that witnessed the required profiles. These events are listed in Tab. 1. There are some caveats towards the use of the respective cloud profiles. First, very low IWC values at or below about 10 ppmv are below the associated water vapour concentrations. Although thin cirrus are common in the Arctic (e.g. Hong and Liu, 2015), this complicates the distinction of the precise cloud impact in face of the surrounding atmosphere. Second, some profiles are interrupted by short periods of horizontal flight, extending the time between the cloud bottom and top crossing to well above
10 min. This exacerbates the uncertainty as to how well the measurements actually correspond to a real vertical cloud profile. Third, the mid latitude profiles south of 60° N (Germany, Denmark, southern Sweden) are of lesser interest for this study.

Following these criteria, the profiles on 25 January 2016 and 09 March 2016 are the best candidates for the present study. The January profile still includes two horizontal sections with an accumulated duration of 18 min, which equals about 230 km in distance.
Focusing on these two profiles, Fig. 5a, c display respective time-altitude cross sections (curtains). The cirrus profiles are marked by the thicker section of the shown flight paths in orange. The background shading displays the squared Brunt-Väisälä frequency $N^2$ from ECMWF reanalysis as an indicator for the vertical static stability of the atmosphere. Troposphere and stratosphere can be clearly distinguished by dark blue colours and greenish/yellow tones, respectively. Troposheric enhance-

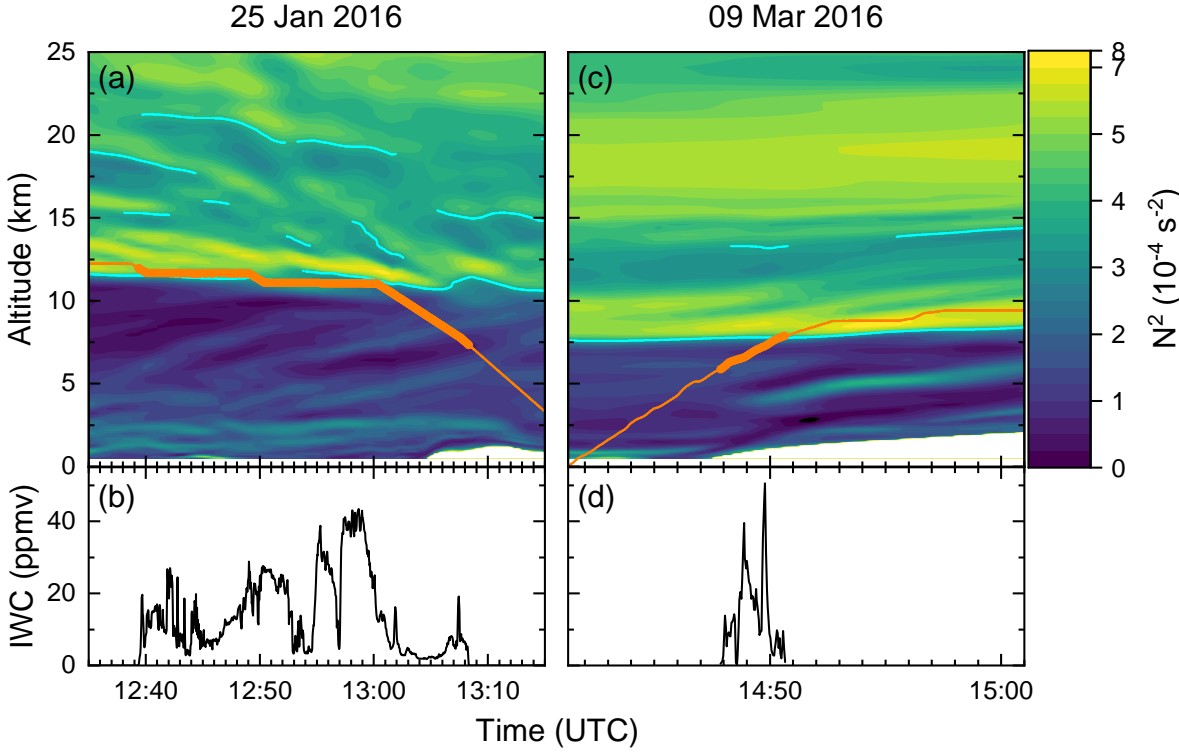

**Figure 5.** Time series of the cirrus cloud profiles on 25 January and 09 March 2016. (a) and (c) The orange line indicates the aircraft flight path. The thicker section highlights the location of the cloud profile. The background shading displays a vertical cross section of squared Brunt-Väisälä frequency $N^2$ from ECMWF reanalysis data interpolated along the flight track. Cyan lines indicate thermal tropopause altitudes according to the WMO definition. (b) and (d) Time series of IWC measurements from the WARAN instrument.

ments with $N^2$ up to $4.0 \, \mathrm{s}^{-2}$ in the boundary layer and at $5 \, \mathrm{km}$ altitude are identifiable. The strongest $N^2$ enhancement is
found directly above the thermal tropopause (cyan line, representing the WMO definition), marking a pronounced tropopause inversion layer in both cases, despite very different tropopause altitudes. The extended vertical profiles of the reanalysis data show different patterns in both profiles: On 25 January, the stratospheric profile features a fine-scaled structure of static stability that trigger multiple thermal tropopause identifications (cyan lines). On 09 March instead, one broad stratospheric inversion above $13 \, \mathrm{km}$ can be seen. Both these observations are not uncommon in regions influenced by orographic gravity waves and
in the vicinity of the jet. Missing $N^2$ data at low altitudes is due to orographic cutoff of the reanalysis data above terrain in Scandinavia or Greenland.

Panels b and d contain the in situ IWC measurements from the WARAN instrument, where the time series are aligned with the curtain plots above. On 25 January, the measurements show a considerable variability in the horizontal flight sections, reflecting the overall patchy atmospheric structure. In contrast, the descent towards the end of the profile is characterised
by IWC below $4 \, \mathrm{ppmv}$ before again enhanced IWC at the cloud bottom below $8.4 \, \mathrm{km}$. On 09 March the profile exhibits an

almost symmetrical double-peaked structure of IWC distribution inside the cloud. In both cases the IWC varies frequently by up to one order of magnitude. From these slant profiles, we derive vertical IWC profiles for the later sections just by stacking the measurements vertically, neglecting any horizontal inhomogeneity in the clouds. Horizontal flight path sections are condensed into one mean IWC value. Although this no longer represents the actual whole (but unknown) three-dimensional IWC distribution inside the cloud, we deem these artificial one-dimensional profiles still as realistic. Especially in the critical 25 January case, a very similar profile would result if we only considered the last section of the flight path that contains the majority of vertical coverage.

We note that possible lower-level clouds might exist below the observed cirrus or also along ground-reaching flight paths. Beyond possible visual identification, such clouds were excluded from sampling (through inlet flow cutoff) during the POL-STRACC campaign to avoid impairment of the targeted measurements of low concentrations of water vapour and cloud water in the upper troposphere and above.

## 5 Radiative transfer calculations

Radiative transfer calculations focus on the ice clouds that have been probed on 25 January 2016 and 09 March 2016. Both days show significant differences in the shape of the $IWC(z)$ and in the geometrical thickness of the ice cloud (Fig. 6). In particular, the profiles $IWC(z)$ reveal vertical fine structures.

The chapter describes the radiative transfer model, its input and treated radiation parameters. It is shown how the specific properties of the measured profiles $IWC(z)$ affect irradiances, the radiative forcing, as well as heating rate profiles. The radiative effects of detected IWC fine structures are discussed by comparing to results which would emerge for IWC profiles of vertically coarser resolutions. The mentioned uncertainty with the IWC measurements, likely an underestimation, is addressed by repeated calculations using the IWC profiles times 5.

### 5.1 Methods

#### 5.1.1 Radiative transfer model and input parameters

Radiative transfer calculations are based on the routine UVSPEC from the program package libRadtran (Mayer and Kylling, 2005). The 1D six-stream radiative transfer solver DISORT was used to provide static profiles of irradiances and heating rates in the shortwave (SW = 0.24 – 5.0 μm) and the longwave spectral range (LW = 2.5 – 100 μm). Figure 6 shows IWC profiles measured on 25 January 2016 and on 09 March 2016 together with the temperature profiles serving as input. Apart from the IWC, radiative transfer calculations need the effective radius ($r_{eff}$). While the IWC is based on in situ measurements, assumptions are made for $r_{eff}$. The vertical profiles $r_{eff}(z)$ have been generated by using a parameterization after Liou et al. (2008) which describes $r_{eff}$ as a function of the IWC by a polynomial fit of observed data. The data have been collected at Arctic latitudes during the DOE's ARM MPACE experiment at the ARM's North Slope of Alaska site in Fall 2004. The

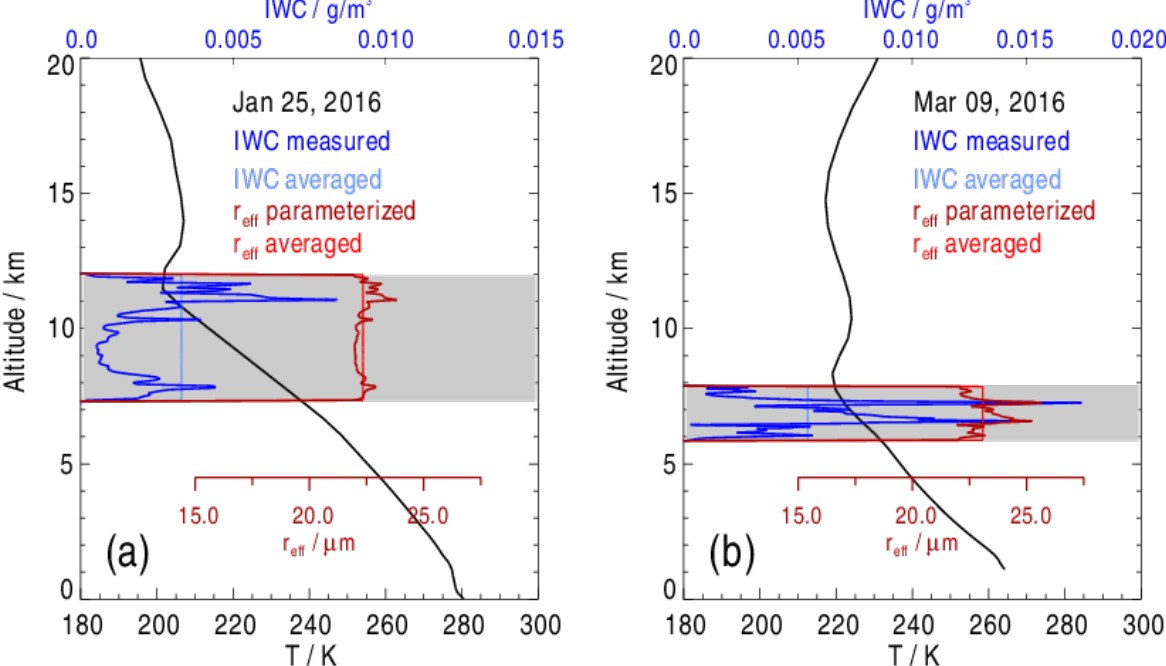

**Figure 6.** Vertical profiles of the measured ice water content IWC on (a) 25 January and (b) 09 March 2016 (dark blue) and the effective radius $r_{\text{eff}}$ (dark red) derived from a parametrization after Liou et al. (2008) describing $r_{\text{eff}}$ as a function of IWC. Vertical averages of IWC and $r_{\text{eff}}$ are indicated as lines in light blue and light red. Dark lines show the temperature profiles, grey shading illustrates the total vertical extent of the measured ice clouds.

effective radii, $r_{\text{eff}}$, very probably do not exactly correspond to those present in the ice clouds measured during POLSTRACC, but this way their values may at least be realistically limited.

After Liou et al. (2008), the dependence of $r_{\text{eff}}$ on IWC is only weak for Arctic ice clouds and for IWC > 0.0015 g m$^{-3}$. Consequently, Fig. 6 shows relatively small altitude dependent variations of the $r_{\text{eff}}$ and similarly, $r_{\text{eff}}$ changes by less than a 280 factor 2 with IWC times 5. Thus, the vertical profile of the ice cloud optical thickness is only highly correlated with the profile of the IWC itself. For IWC < 0.0015 g m$^{-3}$ values of the polynomial fit have been linearly extrapolated to avoid an unrealistic increase of $r_{\text{eff}}$.

The translation of IWC and $r_{\text{eff}}$ into the optical properties follows Baum et al. (2005, 2007). Due to the lack of in situ measurements of ice crystal habits during POLSTRACC the UVSPEC setting of a general habit mixture (GHM) is taken. 285 The GHM assumes size-dependent fractions of different habits including for example pristine crystals like droxtals, columns and plates, bullet rosettes as well as aggregates of columns and plates. Under these assumptions the optical thickness of the ice cloud on 25 January 2016 results in $\tau_{0.55\,\mu m}$ = 0.65 and in $\tau_{0.55\,\mu m}$ = 0.68 for the ice cloud on 09 March 2016, which is moderately opaque (e.g. Kienast-Sjögren et al., 2016). Horizontally homogeneous model clouds are resolved vertically into 72 layers in order to adequately represent measured IWC fine structures.

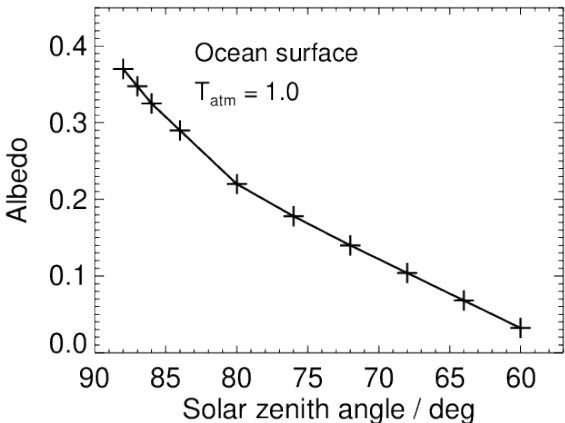

**Figure 7.** Broadband albedo at the ocean surface as a function of solar zenith angle after Payne (1972) for an atmospheric transmission transmission of $\tau_{\mathrm{atm}} = 1.0$. These values have been taken as input to the radiative transfer model.

Concerning the reflection properties of the lower boundary, surface types typical for Arctic regions are considered: a snow/ice free ocean, a snow-covered surface and one land surface assumed to represent a mix of snow-covered and snow-free areas. In principal, the reflection properties of these surface types are highly variable in time and space and depend on various parameters. The reflection of the ocean, for example, is affected by the surface wind speed and by the type and concentration of existing hydrosols. The reflection properties of snow depend on the grain size of the snow, its age, and the degree of pollution.

Although the term albedo is used in the following and for the sake of simplification, the lower boundary condition of the model atmosphere is described by a spectrally constant and isotropic bidirectional reflection distribution function (BRDF) which depends on solar zenith angle (sza) in case of the ocean and is independent of the sza for snow and the mixed surface. The BRDF is a pure property of the surface and not affected by atmospheric conditions.

     Simplifications are made for numerical simulations. In case of the ocean, a broadband albedo following measurements of
Payne (1972) is used which depends on sza and represents a mean surface roughness at $7.5\,\mathrm{kn}$ ($3.9\,\mathrm{m\,s^{-1}}$) windspeed. The ocean albedo increases significantly for large sza values. For this study data corresponding to the curve in Fig. 7 are taken. As an approximation the snow surface is represented by a constant broadband albedo of $\alpha_{\mathrm{snow\,1}} = 0.60$ and $\alpha_{\mathrm{snow\,2}} = 0.85$, representing old and fresh snow, respectively (e.g. Gardner and Sharp, 2010). Since the variation of the snow albedo is less pronounced for sza $\geq 60°$ compared to the ocean albedo, a sza-dependence is not taken into account. Furthermore, to give
an orientation about the albedo effects of Arctic land surface composed of different components, as for example a mixture of snow and snow-free (rocky) areas, a constant value $\alpha_{\mathrm{mix}} = 0.30$ is used. In the LW range a broadband emissivity of $\varepsilon = 0.99$ is assumed for the ocean and the mixed surface and $\varepsilon = 1.0$ for the snow surfaces according to Wilber et al. (1999).

     For the aerosol a UVSPEC setting is selected which describes a standard maritime haze in the lower $2\,\mathrm{km}$ in the winter season. The vertical profiles of the meteorological parameters and the trace gases are described in the following section.

### 5.1.2 Meteorological data and trace gases

For the one-dimensional radiative transfer calculations, a complete profile of temperature and radiatively relevant species in the troposphere and stratosphere is required. We therefore use data from the European Centre for Medium-Range Weather Forecasts Integrated Forecast System (ECMWF IFS) analysis product. Distributions of methane (CH4), nitrogen monoxide (NO) and nitrogen dioxide (NO2) are taken from the Copernicus Atmospheric Monitoring Service (CAMS). The ozone (O3) distribution is taken from the Global and regional Earth-system Monitoring using Satellite and in situ data (GEMS). High resolution hourly data on 137 model levels are provided for the meteorological quantities including specific humidity (3-hourly and 60 levels for CAMS/GEMS). The model data are interpolated in space and time on the flight paths of the HALO aircraft during the POLSTRACC campaign.

In situ values of meteorological quantities like atmospheric pressure and temperature as well as geolocation are provided by the Basic HALO Measurement And Sensor System (BAHAMAS) at a 1 Hz frequency (Giez et al., 2017). The BAHAMAS temperature and humidity data and in situ-measured ozone are only used for comparison in order to assess the consistency of the model with the in situ data during the relevant flight sections. Temperature is captured quite well by the model, with a slight tendency to underestimate the values by up to 2 K. The vertical structure in (slant) profiles is consistent, especially the height of the thermal tropopause coincides from both sources. Specific humidity in the vicinity of the IWC measurements (i.e., in neighbouring cloud-free sections) is found to agree within about 5 % and lacking a systematic bias. This also translates to a good representation of specific humidity (for context, e.g. Kaufmann et al., 2018). The GEMS ozone product is consistent with the in situ measurements from the on-board FAIRO instrument (Zahn et al., 2012) near the surface and for the most part in the stratosphere. In the regime above 500 ppbv O3, the model does not capture the small scale variability of the observations with typical deviations of 15 to 20 %. Due to the overall satisfactory agreement, model and in situ data are not merged for single atmospheric parameters to avoid possible edge artifacts. Instead we use the model data for the whole column for all quantities except from IWC.

### 5.1.3 Radiative quantities

With a view to the following sections the definitions of five radiative quantities are given. A basic model output is the upward and downward directed irradiance ($F_{\mathrm{up},\Delta\lambda}$, $F_{\mathrm{down},\Delta\lambda}$) at each model layer. These irradiances are integrated over a wavelength interval $\Delta\lambda$, here over the SW and LW range, and are given in $\mathrm{W\,m^{-2}}$. The balance of incoming and outgoing irradiance at each model layer is defined as:

$$F_{\Delta\lambda} = F_{\mathrm{down},\Delta\lambda} - F_{\mathrm{up},\Delta\lambda} \ . \tag{4}$$

Balancing the irradiances $F_{\Delta\lambda}$ over the SW and LW spectral intervals gives the net radiation budget or net irradiance in $\mathrm{W\,m^{-2}}$, hereinafter referred to as $F_{\mathrm{net}}$:

$$F_{\mathrm{net}} = F_{\mathrm{SW}} + F_{\mathrm{LW}} \ . \tag{5}$$

The net forcing $\mathrm{RF}_{\mathrm{net}}$ in $\mathrm{W\,m}^{-2}$ is defined as the difference of $F_{\mathrm{net}}$ calculated for the atmosphere with an embedded ice cloud minus $F_{\mathrm{net}}$ for the cloud-free atmosphere:

$$\mathrm{RF}_{\mathrm{net}} = F_{\mathrm{net,ice\ cloud}} - F_{\mathrm{net,cloud\text{-}free}} \ . \tag{6}$$

The heating rate $H_{\Delta\lambda}$ integrated over the wavelength interval $\Delta\lambda$ describes the temperature change of a model layer in $\mathrm{K\,d}^{-1}$ (Kelvin per day) and is defined as:

$$H_{\Delta\lambda} = -1/(\rho_{\mathrm{air}}\,c_p) \cdot \partial F_{\Delta\lambda}/\partial z \ . \tag{7}$$

$\rho_{\mathrm{air}}$ denotes the air density, $c_p$ the specific heat capacity of air at constant pressure. $\partial F_{\Delta\lambda}/\partial z$ is the vertical divergence of the irradiance $F_{\Delta\lambda}$.

Balancing $H_{\Delta\lambda}$ over the SW and LW spectral interval gives the net heating rate in $\mathrm{K\,d}^{-1}$, hereinafter referred to as $H_{\mathrm{net}}$:

$$H_{\mathrm{net}} = H_{\mathrm{SW}} + H_{\mathrm{LW}} \ . \tag{8}$$

## 5.2 Results

### 5.2.1 TOA net irradiances

Net irradiances at the TOA ($F_{\mathrm{net,TOA}}$) as a function of the sza based on IWC profiles measured on 25 January and on 09 March 2016 are shown in Fig. 8. Dashed curves result for the cloud-free atmosphere. Solar zenith angles in the range of $88° \geq \mathrm{sza} \geq 60°$ occur at Arctic latitudes north of $66°\,\mathrm{N}$ within the period from 10 September to 31 March. Horizontal gray lines calculated with $\varepsilon = 1.0$ additionally show the LW component of the irradiance $F_{\mathrm{LW,TOA}}$. The SW component $F_{\mathrm{SW,TOA}}$ is the difference between the curves for $F_{\mathrm{net,TOA}}$ and $F_{\mathrm{LW,TOA}}$. Figures 8a, b compare $F_{\mathrm{net,TOA}}$ over an ice-free ocean and over surfaces assumed to be covered with aged ($\alpha_{\mathrm{snow\ 1}} = 0.60$) and fresh snow ($\alpha_{\mathrm{snow\ 2}} = 0.85$). The curve for the albedo $\alpha_{\mathrm{mix}} = 0.30$ is, as mentioned, intended to give an orientation about $F_{\mathrm{net,TOA}}$ for land surfaces with partial snow cover.

Note, in order to estimate the diurnal course of net irradiances for selected geographical locations and days from Fig. 8 (and following figures), simply the sza has to be determined as a function of latitude, longitude, day, and daytime. If the local minimum sza is known, the possible irradiance range lies to the left of this sza.

For all curves in Fig. 8, $F_{\mathrm{net,TOA}}$ is minimal at highest sza and increases with decreasing sza. In other words, with increasing sza $F_{\mathrm{net,TOA}}$ converges to the longwave component $F_{\mathrm{LW,TOA}}$ (gray lines).

For a better understanding of sza dependent irradiances in the shortwave range, $F_{\mathrm{SW}}$, it should be noted: With increasing sza the light paths through the atmosphere become longer. As a consequence, the absolute value of the contribution of the downward directed irradiance $F_{\mathrm{SW,down}}$ to $F_{\mathrm{net,down}}$ decreases in all layers within the atmosphere. The proportion of the direct component $F_{\mathrm{SW,down,dir}}$ in each atmospheric layer depends on the optical thickness of the atmosphere between this layer and the TOA. Within ice clouds, a large part of $F_{\mathrm{SW,down,dir}}$ is scattered in the forward direction by the ice crystals, but the proportion of direct radiation at the cloud base nevertheless decreases in favour of the diffuse component $F_{\mathrm{SW,down,diff}}$ with increasing sza. To give an example, on 25 January 2016 and at $\mathrm{sza} = 64°$ the proportion of $F_{\mathrm{SW,down,dir}}$ transmitted through

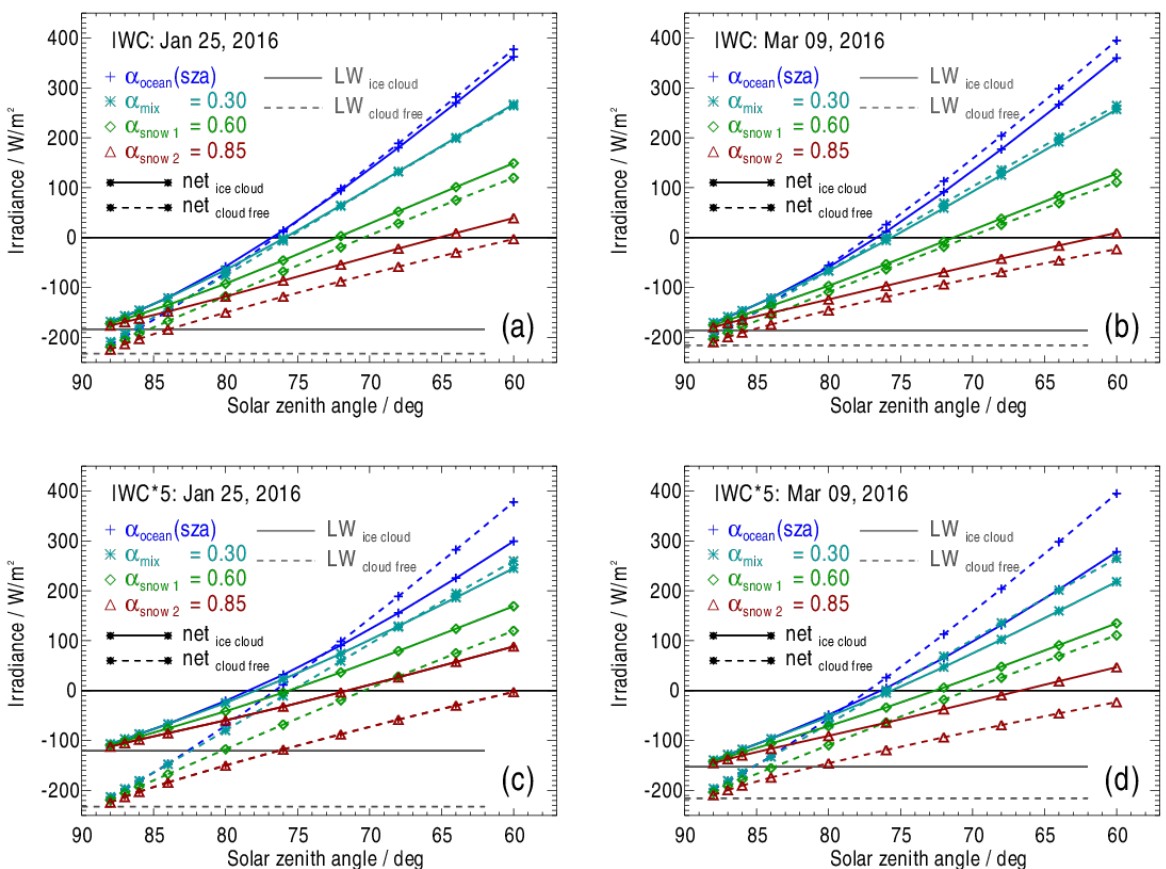

**Figure 8.** (a) Irradiance at the top of the atmosphere on 25 January 2016 as a function of solar zenith angle over an ice-free ocean, over snow-covered surfaces and over a mixed surface with albedo values as indicated. (b) As in (a), but for the atmosphere on 09 March 2016. Solid curves result for the atmosphere with an embedded ice cloud, dashed curves for the same but cloud-free atmosphere. (c) and (d) as in (a) and (b), but for ice clouds with higher optical thicknesses obtained by multiplying the measured profile IWC($z$) with a factor of 5. Horizontal gray lines, valid for $\varepsilon = 1.0$, indicate the LW component of the irradiance.

the cloud is about 25 %, that of diffuse radiation correspondingly 75 %. For sza = 88° $F_{\mathrm{SW,down,dir}}$ is negligible. At the cloud top, about 26 % of the incident solar irradiance $F_{\mathrm{SW}}$ is reflected at sza = 64°, whereas at sza = 88° it is 66 %. At cloud top the absolute values of upward directed irradiances $F_{\mathrm{SW,up}}$ strongly depend on the surface albedo and are highly correlated with

$F_{\mathrm{net,TOA}}$.

     Although at the maximum sza = 88° the difference between $F_{\mathrm{net,TOA}}$ over the ocean and $F_{\mathrm{net,TOA}}$ over the snow surfaces is relatively small, e.g. about $10\,\mathrm{W\,m^{-2}}$ for $\alpha_{\mathrm{ocean}} = 0.37$ and $\alpha_{\mathrm{snow}} = 0.85$, the curves diverge quickly with decreasing sza (Figs. 8a–d). Reason is that in the case of atmospheres containing semi-transparent ice clouds or under cloud-free conditions, the contribution of the (positive) $F_{\mathrm{SW,TOA}}$ increases with decreasing surface albedo and with decreasing sza. This effect is

further enhanced by differences between the sza-dependent ocean albedo and the constant snow albedo which increase with decreasing sza. As a consequence, the transition from negative to positive values of $F_{\mathrm{net,TOA}}$, equivalent to a change from a loss to a gain of radiative energy, occurs at a greater sza over the ocean than over aged snow ($\alpha_{\mathrm{snow\ 1}} = 0.60$) or fresh snow ($\alpha_{\mathrm{snow\ 2}} = 0.85$). Furthermore, over snow surfaces the transition from a negative to a positive $F_{\mathrm{net,TOA}}$ is shifted towards a higher sza when compared to results for a cloud-free atmosphere.

In case of the mixed surface ($\alpha_{\mathrm{mix}} = 0.30$) the differences of $F_{\mathrm{net,TOA}}$ between the cloud-free atmosphere and the atmosphere containing an ice cloud are smallest for about sza $< 82°$ (Fig. 8a). This is a constellation where the specific ice cloud properties result in a negligible net radiative forcing $\mathrm{RF}_{\mathrm{net}}$ (Eq. 6).

    Comparing the curves of $F_{\mathrm{net,TOA}}$ in Figs. 8a and b, their behaviour as a function of sza and surface albedo is qualitatively the same. Differences between cloudy and cloud-free atmospheres, resulting in the radiative forcing, $\mathrm{RF}_{\mathrm{net}}$, are further

discussed in Sec. 5.2.3.

    It was also investigated to what extent higher optical thicknesses of the ice clouds affect $F_{\mathrm{net,TOA}}$. For this purpose measured profiles IWC($z$) were multiplied by a factor of 5 with $r_{\mathrm{eff}}$ being adjusted according to the parametrization of Liou et al. (2008) in each layer. Higher values of IWC($z$) result in optical thicknesses of $\tau_{0.55\,\mu\mathrm{m}} = 2.94$ and $\tau_{0.55\,\mu\mathrm{m}} = 2.85$ for 25 January and 09 March, respectively. In the SW, higher optical thicknesses of the ice clouds lead to a larger contribution of the upward directed

reflected radiation to $F_{\mathrm{net,TOA,SW}}$. At the same time, increased optical thicknesses of ice clouds increase the cloud LW effect, i.e., rendering more surface emitted LW radiation to be absorbed by ice clouds and hence $F_{\mathrm{net,LW,TOA}}$ less negative (Figs. 8c, d vs. Figs. 8a, b). One net effect is an overall shift of $F_{\mathrm{net,TOA}}$ towards higher values, especially noticeable at large sza, where SW effects are increasingly negligible. For example, $F_{\mathrm{net,TOA}}$ is increased by about $62\,\mathrm{W\,m^{-2}}$ on 25 January and by about $32\,\mathrm{W\,m^{-2}}$ on 09 March at sza $= 88°$.

## 5.2.2   Surface net irradiances


Figure 9 presents net irradiances at the surface, or bottom of the atmosphere, $F_{\mathrm{net,BOA}}$. The effect of various surface albedo values on the relative course of the curves to each other corresponds to that in Fig. 8. However, in comparison to $F_{\mathrm{net,TOA}}$ (Fig. 8) the decrease in $F_{\mathrm{net,BOA}}$ with increasing sza is less pronounced. The reason for this is the generally reduced LW radiation emitting from the surface, $F_{\mathrm{LW,BOA}}$, as the values at sza $= 88°$ show. $F_{\mathrm{LW,BOA}}$ is the limiting factor for $F_{\mathrm{net,BOA}}$. At the other

end of the $x$-axis at small sza values, the curves of $F_{\mathrm{net,BOA}}$ (Fig. 9) and $F_{\mathrm{net,TOA}}$ (Fig. 8) tend to converge for the same albedo values, a consequence of increasing forward scattering of incoming SW radiation at the ice crystals. Generally, with decreasing sza an increasing shortwave component $F_{\mathrm{SW,BOA}}$ increases $F_{\mathrm{net,BOA}}$, an effect that becomes stronger the smaller the surface albedo is. For $\alpha_{\mathrm{snow\ 2}} = 0.85$ the difference between the downward and upward directed components of $F_{\mathrm{SW,BOA}}$ is minimal leading to a weak dependence of $F_{\mathrm{net,BOA}}$ with sza (Figs. 9a–d). As can also be seen from Fig. 9, the ice cloud lowers the

positive $F_{\mathrm{net,BOA}}$ at the dark ocean surface within a wide range of the sza when compared to the cloud-free atmosphere. Reason is a dominating shadowing effect of the ice clouds, i.e. downward directed $F_{\mathrm{SW,BOA}}$ is reduced more strongly than the surface LW emission $F_{\mathrm{LW,BOA}}$ changes (see gray lines). Over a bright snow surface with $\alpha_{\mathrm{snow\ 2}} = 0.85$, SW contributions to $F_{\mathrm{net,BOA}}$ become smaller leading to correspondingly smaller changes of $F_{\mathrm{net,BOA}}$. In most cases the curves representing the

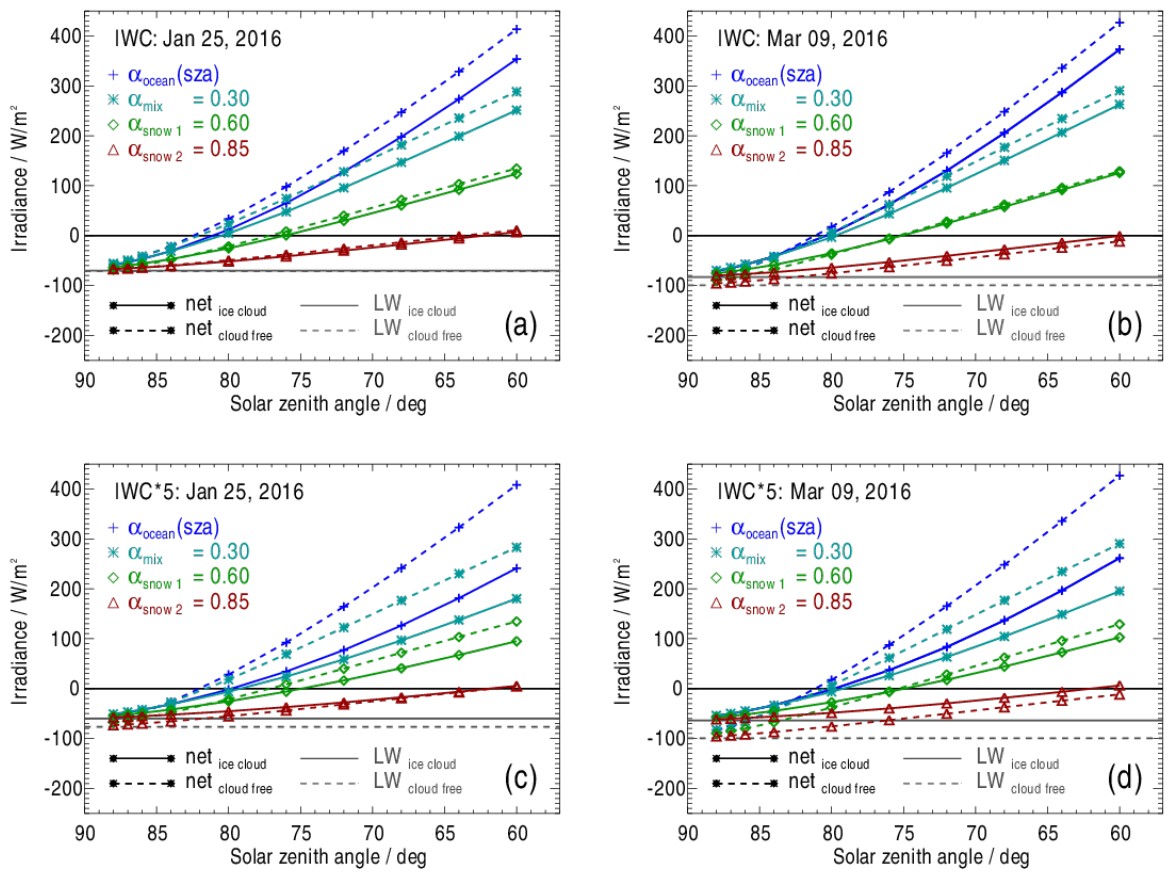

**Figure 9.** Irradiance at the bottom of the atmosphere (surface), otherwise as in Fig. 8.

cloud-free atmosphere are even below those of the cloudy atmosphere. $F_{net,BOA}$ based on IWC times 5 is shown in Figs. 9c,
d.

### 5.2.3 Radiative forcing

Net radiative forcings at the TOA ($RF_{net,TOA}$) as defined by Eq. 6, are displayed as a function of sza for 25 January 2016 and
09 March 2016 in Figs. 10a–d. Over bright snow surfaces $RF_{net,TOA}$ is positive on both days within the entire range $88° \geq$ sza
$\geq 60°$.

At sza = 88° the net forcing is mainly determined by the LW forcing resulting in an energy gain of the system. With
decreasing sza in the range of about 88° > sza > 70° an ice cloud over snow (and partly over the mixed surface) increases the
amount of reflected SW radiation leading to a reduction of the atmospheric energy gain. For sza < 70° the amount of reflected
SW radiation decreases due to the pronounced forward scattering of ice crystals, herewith causing an increase of $RF_{net,TOA}$.

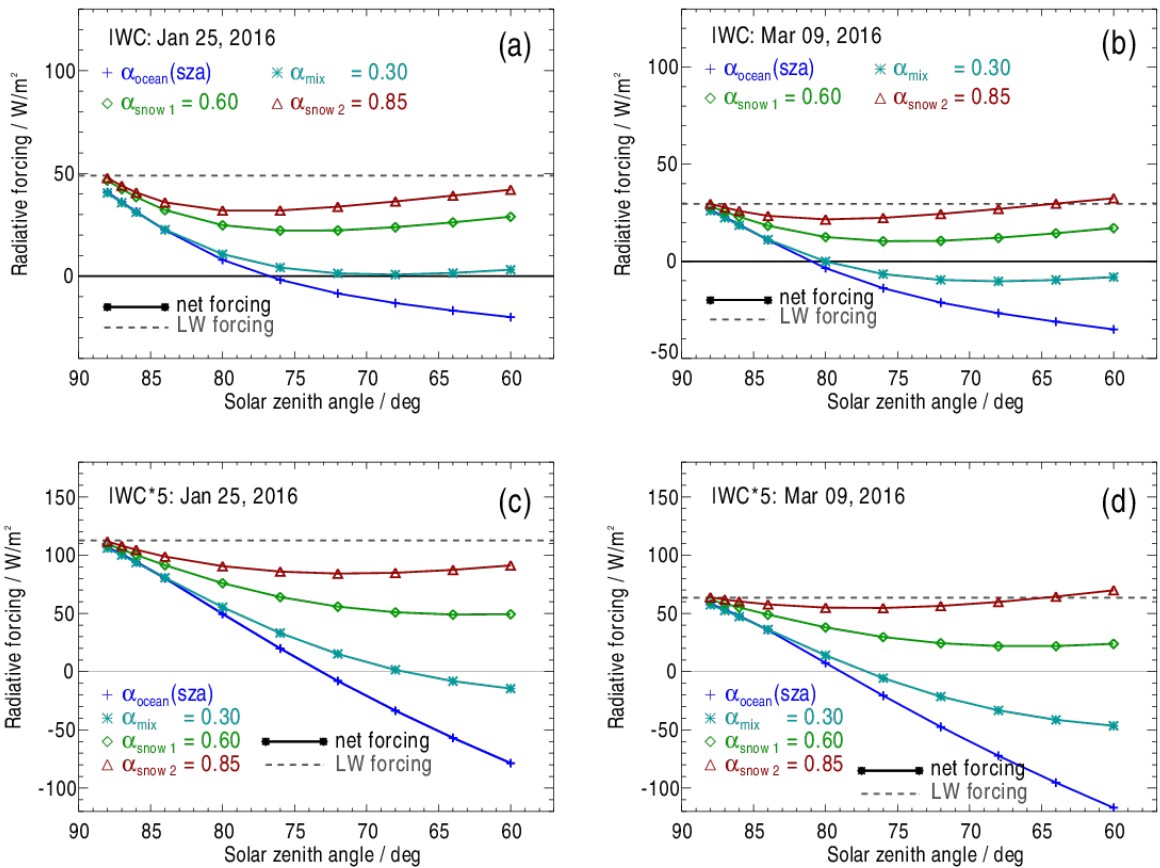

**Figure 10.** Radiative forcing at the top of the atmosphere as a function of the solar zenith angle on (a) 25 January 2016 and on (b) 09 March 2016 for different surface albedo assumptions as indicated. (c) and (d) as in (a) and (b), but for ice clouds with higher optical thicknesses obtained by multiplying the measured profile IWC($z$) with a factor of 5. Horizontal dashed lines, valid for $\varepsilon = 1.0$, indicate the LW component of the forcing.

Over the ocean, positive $RF_{net,TOA}$ result for sza > 68° on 25 January 2016 and for sza > 78° on 09 February 2016. For
smaller sza values $RF_{net,TOA}$ even turns into the negative range. One reason for a negative $RF_{net,TOA}$ is that at a large sza the ocean albedo is relatively high, but decreases significantly with decreasing sza (Fig. 7). The smaller the ocean albedo becomes with decreasing sza, the larger is the relative contribution of outgoing SW radiation that reduces $RF_{net,TOA}$. In general it can be said that the sign of $RF_{SW,TOA}$ is predominantly determined by whether the ice cloud albedo is greater or smaller than the surface albedo. Furthermore, ice cloud layers emit at higher temperatures on 09 March 2016 with the consequence that the LW
component of the radiative forcing is about $15 - 20\,\mathrm{W\,m^{-2}}$ smaller than on 25 January 2016. All curves are affected, over the ocean this means that the transition of $RF_{net,TOA}$ into negative values is shifted towards a larger sza (Figs. 10a, b.). Curves for

$RF_{net,TOA}$ with IWC times 5 are displayed in Figs. 10c, d. Their greater spread is a consequence of increased ice cloud optical thicknesses.

### 5.2.4 IWC fine structures

As mentioned, IWC profiles measured during the POLSTRACC campaign show vertical fine structures (Figs. 6a, b). It is investigated how such structures affect the profiles of the net irradiances $F_{net}(z)$ and heating rates $H(z)$. As a reference, additional radiative transfer calculations have been carried out for synthetic ice clouds with a vertically constant IWC profile, $IWC(z)_{const}$. In a second step, IWC profiles with different vertical resolutions are used as a reference. All synthetic IWC profiles together with the assigned profiles of the effective radius $r_{eff}(z, IWC(z))$ after Liou et al. (2008) are designed in such

a way that the optical thickness is the same as for $IWC(z)_{meas}$. Note, profiles of the temperature affecting the LW irradiance profiles are displayed in Fig. 6.

Figures 11a, e, i, m compare SW and LW irradiance profiles, $F_{SW}(z)$ and $F_{LW}(z)$, calculated for the profiles $IWC(z)_{meas}$ and $IWC(z)_{const}$. Altitude-dependent differences of $F_{SW}(z)$ and $F_{LW}(z)$ (solid and dashed line) follow the corresponding difference between $IWC(z)_{meas}$ and $IWC(z)_{const}$. Note, apart from the influence of trace gases and aerosols and in view of a

relatively weak variability of the profiles of $r_{eff}(z, IWC(z))$ (Fig. 6), the LW thermal emission of a layer is proportional to the ice water content, and herewith proportional to the cloud optical thickness of the layer. In addition, the LW emission depends on the temperature distribution within the ice cloud layer. At the large solar zenith angle sza = 88°, LW radiation dominates the altitude-dependent changes of $F_{LW}(z)$ (Figs. 11a, i).

On 25 January 2016 LW irradiances at the cloud top, $F_{LW,TOC}$, are smaller for the measured profile $IWC(z)_{meas}$ when

compared to $F_{LW,TOC}$ which are related to $IWC(z)_{const}$ (Fig. 11a). Reason is a smaller LW emission from the IWC maximum near the cloud top due to relatively cold temperatures in combination with an increased optical thickness here. Below the upper IWC peak, $F_{LW}$ calculated for $IWC(z)_{meas}$ is larger than for $IWC(z)_{const}$ because the upward directed emission from the surface and from the IWC maximum near the cloud base are balanced with smaller downward directed emissions from colder layers above. Major differences between irradiances for the $IWC(z)_{meas}$ and $IWC(z)_{const}$ are confined to the layers within

the cloud and reach up to about $5\,W\,m^{-2}$. Thus, on 25 January 2016, it is the combination of the vertical inhomogeneity of the IWC profile (two different maxima) and the geometrical thickness of the entire ice cloud associated with larger vertical temperature differences that cause pronounced altitude dependent LW irradiance differences. The ice cloud causes irradiance differences in the order of a few $W\,m^{-2}$ which penetrate up to the TOA (Tab. 2).

On 09 March 2016 the geometrical thickness of the ice cloud is reduced showing two closely spaced maxima in the middle

of the IWC profile. The overall shape is more symmetric than on 25 January 2016 (Fig. 6b). Switching from the measured to the constant IWC profile does not lead to a stronger weighting of layers emitting LW radiation with significantly different temperatures. As a consequence, LW irradiance differences inside the ice cloud are smaller than on 25 January 2016 (about $2\,W\,m^{-2}$) and even smaller at the TOA and BOA (Fig. 11i, Tab. 2).

A smaller solar zenith angle of sza = 72° increases the contribution of SW radiation significantly (Figs. 11e, m). Irradiance

differences due to the influence of differently shaped IWC profiles are mainly a result of scattering processes. SW downward

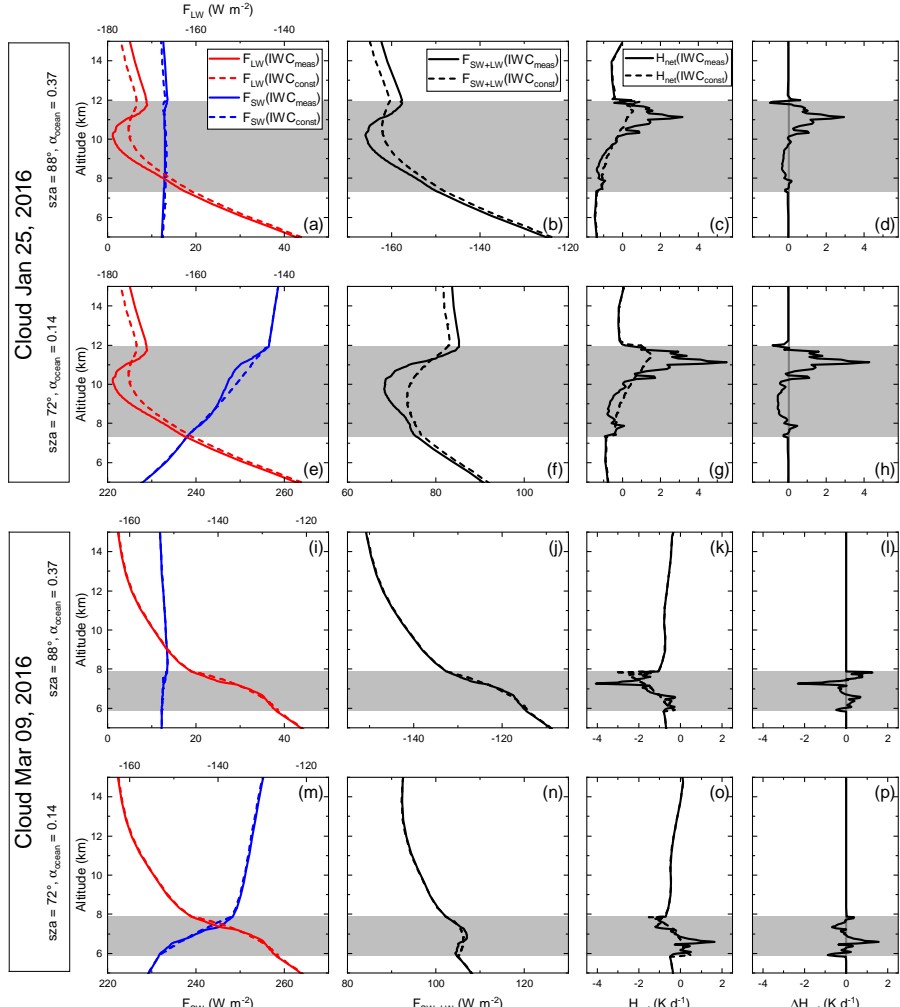

**Figure 11.** Irradiance profiles of measured (solid) and constant (dashed) IWC profiles for selected values of solar zenith angle and albedo as indicated. First column: LW (red, top axis scale) and SW (blue, bottom axis scale) irradiances. The top axis is shown in negative numbers since $F_{LW}$ is upward directed. Second column: Net irradiances. All irradiance scales span a range of $50\,\mathrm{W\,m^{-2}}$, albeit at different absolute positions. Third column: Net heating rate profiles of the measured and the constant IWC profiles. Fourth column: Difference between the net heating rate profiles of the measured and the constant IWC profiles.

directed irradiances $F_{SW}(z)$ calculated for $IWC(z)_{meas}$ are smaller below the relative IWC maxima. The reason is that the cloud optical thickness, and herewith the extinction due to scattering, is higher at altitudes of the IWC maxima when compared to values of the constant profile $IWC(z)_{const}$ at the same altitude (Figs. 11e, m and Fig. 6). On 25 January 2016 the vertically extended maximum of the IWC produces greatest differences. On 09 March 2016 small enhancements of downward directed

$F_{SW}(z)$ result for $IWC(z)_{meas}$ at the upper edge of the maxima due to multiple scattering effects (Fig. 11m).

**Table 2.** Net irradiances at the top of the atmosphere and at the surface, $F_{\mathrm{net,TOA}}$ and $F_{\mathrm{net,BOA}}$, calculated for measured and vertically constant profiles of the ice water content ($\mathrm{IWC}(z)_{\mathrm{meas}}$ and $\mathrm{IWC}(z)_{\mathrm{const}}$). $\Delta F_{\mathrm{net}}$ is the difference $F_{\mathrm{net}}(\mathrm{IWC}_{\mathrm{meas}})$ minus $F_{\mathrm{net}}(\mathrm{IWC}_{\mathrm{const}})$. Selected are results for four solar zenith angles (sza) and for the days 25 January 2016 and 09 March 2016.

| Date | sza | $F_{\mathrm{net,TOA}}$ / $F_{\mathrm{net,BOA}}$ for $\mathrm{IWC}(z)_{\mathrm{meas}}$ | $F_{\mathrm{net,TOA}}$ / $F_{\mathrm{net,BOA}}$ for $\mathrm{IWC}(z)_{\mathrm{const}}$ | $\Delta F_{\mathrm{net,TOA}}$ / $\Delta F_{\mathrm{net,BOA}}$ (measured - constant) |
|---|---|---|---|---|
| | (°) | (W m$^{-2}$) | (W m$^{-2}$) | (W m$^{-2}$) |
| $\alpha_{\mathrm{ocean}}$ (sza), $\varepsilon_{\mathrm{ocean}} = 0.99$ | | | | |
| 25 Jan 2016 | 88 | -168.85 / -59.13 | -171.34 / -58.47 | 2.49 / -0.66 |
| 25 Jan 2016 | 80 | -58.07 / 11.76 | -60.06 / 12.20 | 1.99 / -0.44 |
| 25 Jan 2016 | 72 | 94.40 / 127.86 | 92.40 / 128.26 | 2.00 / -0.40 |
| 25 Jan 2016 | 64 | 269.80 / 273.55 | 267.76 / 273.94 | 2.04 / -0.39 |
| 09 Mar 2016 | 88 | -171.20 / -70.84 | -171.10 / -70.71 | 0.79 / -0.99 |
| 09 Mar 2016 | 80 | -59.41 / 4.41 | -59.49 / 4.56 | 0.08 / -0.15 |
| 09 Mar 2016 | 72 | 91.46 / 130.10 | 91.37 / 130.19 | 0.09 / -0.09 |
| 09 Mar 2016 | 64 | 266.84 / 286.94 | 266.72 / 287.04 | 0.12 / -0.10 |
| $\alpha_{\mathrm{snow}} = 0.85$, $\varepsilon_{\mathrm{snow}} = 1.00$ | | | | |
| 25 Jan 2016 | 88 | -176.31 / -67.48 | -178.97 / -66.97 | 2.66 / -0.51 |
| 25 Jan 2016 | 80 | -118.07 / -52.54 | -120.04 / -52.08 | 1.97 / -0.46 |
| 25 Jan 2016 | 72 | -53.83 / -30.74 | -55.79 / -30.28 | 1.96 / 0.46 |
| 25 Jan 2016 | 64 | 9.20 / -6.11 | 7.21 / -5.65 | 1.99 / -0.46 |
| 09 Mar 2016 | 88 | -179.44 / -79.97 | -179.32 / -79.82 | -0.12 / -0.15 |
| 09 Mar 2016 | 80 | -123.87 / -64.14 | -123.94 / -63.99 | 0.07 / -0.15 |
| 09 Mar 2016 | 72 | -69.12 / -40.56 | -69.19 / -40.41 | 0.07 / -0.15 |
| 09 Mar 2016 | 64 | -15.85 / -14.07 | -15.93 / -13.92 | 0.08 / -0.15 |

Figures 11b, f, j, n compare profiles of net irradiances $F_{\mathrm{net}}(z)$ balanced over the SW and LW range based on $\mathrm{IWC}(z)_{\mathrm{meas}}$ and $\mathrm{IWC}(z)_{\mathrm{const}}$. As is to be expected, $F_{\mathrm{net}}(z)$ reflects that the specific properties of the profile $\mathrm{IWC}(z)_{\mathrm{meas}}$ on 25 January 2016 cause greatest differences to the results obtained for $\mathrm{IWC}(z)_{\mathrm{const}}$. Table 2 contains data of $F_{\mathrm{net}}(\mathrm{IWC}(z)_{\mathrm{meas}})$ and $F_{\mathrm{net}}(\mathrm{IWC}(z)_{\mathrm{const}})$ as well as their differences $\Delta F_{\mathrm{net}} = F_{\mathrm{net}}(\mathrm{IWC}(z)_{\mathrm{meas}})$ - $F_{\mathrm{net}}(\mathrm{IWC}(z)_{\mathrm{const}})$ at the TOA and at the BOA.

Numbers show that for the individual day $\Delta F_{\mathrm{net,TOA}}$ and $\Delta F_{\mathrm{net,BOA}}$ are of the same order of magnitude independent of the two surface types considered. In most cases absolute values of $\Delta F_{\mathrm{net}}$ are smaller than $1\,\mathrm{W\,m^{-2}}$. An exception is again the results for 25 January 2016 at TOA. The reason is already given by Figs. 11a, e, showing that at cloud top the upward directed $F_{\mathrm{LW,TOC}}$ calculated for $\mathrm{IWC}(z)_{\mathrm{const}}$ differs by about $2.5\,\mathrm{W\,m^{-2}}$ from $F_{\mathrm{LW,TOC}}$ related to $\mathrm{IWC}(z)_{\mathrm{meas}}$. Results obtained with IWC times 5 are presented in Tab. A1.

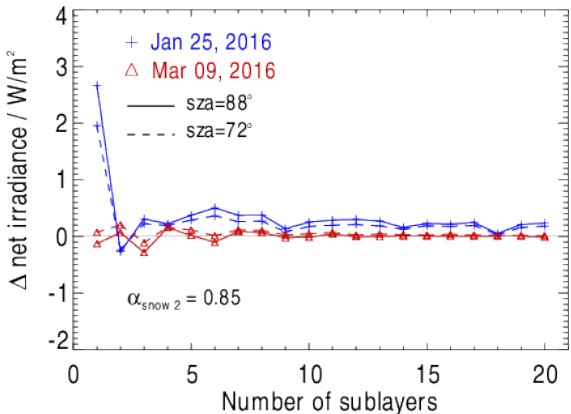

**Figure 12.** Differences between net irradiances $F_{\mathrm{net,TOA}}$ calculated for the measured IWC profile and IWC profiles of different vertical resolution. The x-axis shows the number of layers into which the entire ice cloud is divided.

480  Vertical profiles of the IWC are not always available in high vertical resolution. In models, as for example climate models, the vertical resolution of ice clouds near the tropopause is usually in the order of several hundred meters. How different vertical resolutions of the IWC affect $F_{\mathrm{net,TOA}}$ is illustrated in Fig. 12. It shows differences between $F_{\mathrm{net,TOA}}$ based on IWC measurements on 25 January 2016 and 09 March 2016 and $F_{\mathrm{net,TOA}}$ calculated for IWC profiles resulting from a division of the entire ice cloud into an increasing number of sublayers. Within these sublayers the IWC is vertically averaged. As shown in Fig.

485 12, the curves of $\Delta F_{\mathrm{net,TOA}}$ converge relatively quickly to zero with the number of sublayers, but differently for 25 January 2016 and 09 March 2016. The reason again is given by the different IWC profiles on both days. On 25 January a resolution of the ice cloud into 10 sublayers ($\sim 460\,\mathrm{m}$) results in $\Delta F_{\mathrm{net,TOA}} < 0.4\,\mathrm{W\,m^{-2}}$, whereas on 09 March $\Delta F_{\mathrm{net,TOA}} < 0.2\,\mathrm{W\,m^{-2}}$ is already reached for a resolution into 5 sublayers ($\sim 400\,\mathrm{m}$). This is valid for sza = 88° as well as for sza = 72°.

### 5.2.5 Heating rates

490 According to Eq. 7 wavelength integrated net heating rate profiles are proportional to the vertical divergence of the net irradiance $\partial F_{\mathrm{net}}(z)/\partial z$ and can be read from the profiles shown in Figs. 11b, f, j, n (results in Figs. 11c, g, k, o). $\partial F_{\mathrm{net}}(z)/\partial z$ is mainly affected by the profile of IWC$(z)$ within the ice cloud, so that their differences due to IWC$(z)_{\mathrm{meas}}$ and IWC$(z)_{\mathrm{const}}$ are particularly pronounced there and are minimal above and below the ice cloud layer. As a consequence, differences in net heating rate profiles $\Delta H_{\mathrm{net}}(z)$ associated with IWC$(z)_{\mathrm{meas}}$ and IWC$(z)_{\mathrm{const}}$ occur within the ice cloud on both days, 25

495 January, and 09 March 2016, and for both solar zenith angles sza = 88° and sza = 72° (Figs. 11d, h, l, p). On 25 January 2016 the maximum difference $\Delta H_{\mathrm{net}}$ is $4.27\,\mathrm{K\,d^{-1}}$ at sza = 72° whereas the minimum $\Delta H_{\mathrm{net}}$ results in $-0.99\,\mathrm{K\,d^{-1}}$ at sza = 88°. On 09 March 2016 corresponding maximum and minimum values are $1.57\,\mathrm{K\,d^{-1}}$ and $-2.33\,\mathrm{K\,d^{-1}}$, respectively. Within the

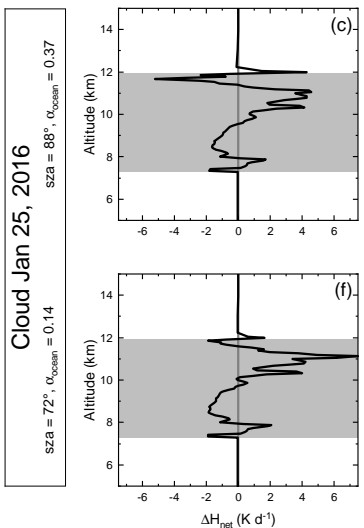
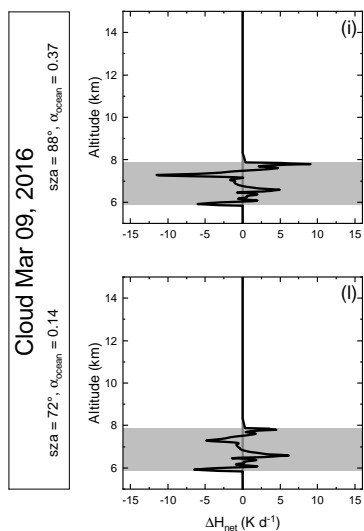

**Figure 13.** Same as in Fig. 11c, f, i, l, but for ice clouds with higher optical thicknesses as a result of multiplying the entire profile IWC($z$) with a factor of 5.

cloud, $\Delta H_{\mathrm{net}}$ is in the order of magnitude of the net heating rate $H_{\mathrm{net}}(z)$, whereas above and below the cloud $\Delta H_{\mathrm{net}}(z)$ is negligible.

Figure 11 presents curves that result for an ocean surface as the lower boundary. Comparing heating rate differences $\Delta H_{\mathrm{net}} = H_{\mathrm{net}}(\mathrm{IWC}_{\mathrm{meas}}) - H_{\mathrm{net}}(\mathrm{IWC}_{\mathrm{const}})$ over the ocean with those over snow ($\alpha_{\mathrm{snow\ 2}} = 0.85$) reveals a difference of $0.02\,\mathrm{K\,d^{-1}}$ (-$0.15\,\mathrm{K\,d^{-1}}$) at the altitude of the maximum $\Delta H_{\mathrm{net}}$ for sza = 88° (sza = 72°) on 25 January 2016 (not shown). The corresponding change at the minimum $\Delta H_{\mathrm{net}}$ results in $0.002\,\mathrm{K\,d^{-1}}$ (-$0.05\,\mathrm{K\,d^{-1}}$). Also, on 09 March 2016 changes at the maximum and minimum $\Delta H_{\mathrm{net}}$ are very small. At the BOA all changes are in the order of $10^{-4}\,\mathrm{K\,d^{-1}}$. Thus, the transition from the ocean surface to a bright snow surface results in small changes in the heating rate profiles, within the ice cloud and outside.

When multiplying the IWC by 5 (Fig. 13), $\Delta H_{\mathrm{net}}(z)$ changes significantly in shape. On 25 January the maximum is $7.48\,\mathrm{K\,d^{-1}}$ for sza = 72° and the minimum $\Delta H_{\mathrm{net}}$ is -$5.21\,\mathrm{K\,d^{-1}}$ for sza = 88°. On 09 March maximum and minimum values of $\Delta H_{\mathrm{net}}(z)$ are $9.06\,\mathrm{K\,d^{-1}}$ and -$11.54\,\mathrm{K\,d^{-1}}$, which both result for sza = 88°. The change in shape or vertical dependency is due to the high optical thicknesses of the clouds and manifests itself mainly in two features: First, the scaling of $\Delta H_{\mathrm{net}}(z)$ with $z$ is stronger at lower altitudes than up higher, due to the higher absolute concentration of ice water in the thicker air at lower altitude. Second, the influence of the SW contribution is more limited to the uppermost cloud layers. This leads to a reduced warming impact of the SW regime on the overall heating rate profile, especially in the 09 March case, where the majority of the IWC distribution is more centred in the cloud profile. Nonetheless, $\Delta H_{\mathrm{net}}$ still remains negligible above and below the clouds.

## 6 Discussion and conclusions

Our radiative transfer simulations for selected profiles of the IWC measured during the POLSTRACC campaign indicate that an ice cloud of extended geometrical thickness and a specific vertically asymmetric fine structure of the IWC changes the radiation budget at the TOA when compared to the results of a vertically constant IWC profile of equal optical thickness. Effects are significant and for our cases in the order of a few $\mathrm{W\,m^{-2}}$. Fine structures in geometrically thin ice clouds having a more symmetric IWC profile do not lead to a noticeable modification of the TOA radiation budget when compared to the case of a vertically constant IWC. This is consistent with the results by Feofilov et al. (2015), who performed a statistical analysis of cirrus IWC profiles and their radiative effects from active and passive space-borne remote sensing products. Similar to our above result, they conclude that the LW effect of a certain profile (at the TOA) is modulated by an "effective radiative layer" with respect to a constant IWC profile, which reflects internal inhomogeneity in the vertical IWC distribution. We show that the uncertainty arising from possible coarser vertical resolution is reduced to below $20\,\%$ for the inhomogeneous IWC profile if layers are thinner than about $500\,\mathrm{m}$. The study by Feofilov et al. (2015) is dominated by low and midlatitude cirrus properties with higher IWP and optical thickness. In contrast, our analyses are focused on thinner (still opaque) Arctic cirrus clouds which exist in darkness or at high solar zenith angles such that their net radiative forcing may be of alternating sign.

A significant impact on net irradiances at the top and the bottom of the atmosphere is found to originate from the albedo of typical Arctic surface types, such as the open ocean or snow covered regions, implying also effects on the radiative forcing. Effects depend on the solar zenith angle, which are strong for $F_{\mathrm{net,TOA}}$ and mostly strong or moderate for $F_{\mathrm{net,BOA}}$, but weak for $F_{\mathrm{net,BOA}}$ under conditions of fresh fallen snow. Age and pollution of snow seem important factors in this context. For example, the transition from negative to positive values of the TOA radiation budget, equivalent to a change from a loss to a gain of radiative energy, occurs at a greater solar zenith angle over the ocean than for example over aged snow ($\alpha_{\mathrm{snow}}$ = 0.60). Furthermore, over aged snow the transition is shifted towards a higher solar zenith angle when compared to results for a cloud-free atmosphere. This means that the presence of the cirrus leads to an enhanced gain of radiative energy in the Arctic atmosphere in most cases during autumn to spring, and over bright surfaces. Therefore in scenarios with an overall negative trend of surface albedos in the Arctic, the positive radiative forcing effect of cirrus is expected to decrease. This decrease is relevant in view of the Arctic amplification (e.g. Wendisch et al., 2017; Perovich and Polashenski, 2012; Thackeray and Hall, 2019) as it constitutes a possible counteracting effect. Notwithstanding, the (direct) impact on surface net irradiances and therefore on surface temperature is much less pronounced.

Instantaneous heating rate profiles of the atmosphere are affected by fine structures of the IWC mainly within the cloud layer and remain negligible below and above the cloud. The effect of different surface albedo values on instantaneous heating rate profiles is only very weak. Depending on the profile of IWC or optical thickness, the heating rate profile can contribute to a strengthening or weakening of a given structure in the temperature profile, especially concerning the cold point of the tropopause, as well as the strength of the above inversion layer. The strong tropospheric lapse rate in some of our observed cloud profiles induces a stabilizing radiative effect of the cloud in most of the tropospheric heating rate profile inside the cloud layer, with an overall higher (LW) heating rate at the cloud top (warming) than at the cloud bottom (less warming or cooling).

However, a heating rate profile first of all will trigger cloud internal processes through evaporation or latent heat release, which drive vertical motion, modify the temperature gradient and change the microphysical properties of the ice particles.

Concerning a probable underestimation of IWC in the measurements, we derived that a potentially higher true IWC profile would lead, as can be expected, to a more warming impact of the cirrus on the TOA radiation budget and dampen the effect of different surface albedo values. At the same time, the heating rate profiles would be more dominated by the LW contribution

due to the strongly enhanced optical thickness. Still we conclude that our original calculations are of value because there exist (and were observed) many thinner cirrus clouds in the Arctic to which our results would apply accordingly, beyond the two cases that we analysed.

## 7 Outlook

Some aspects of Arctic ice clouds have not been treated in this work, notably effects of the modification of size and number

density by specific cloud formation regimes. Likewise, the effects of different ice crystal habits or habit compositions on SW and LW irradiances, radiative forcings, and heating rates were not in the focus of this study. Based on measurements of a subtropical cirrus, Wendisch et al. (2005) and Wendisch et al. (2007) for example show that irradiances are significantly sensitive to different ice crystal shapes in the SW and LW depending on solar zenith angle, location above or below the cloud, cirrus optical thickness, and spectral region within the SW and LW range. For certain constellations, maximum changes of

irradiances even reach 26 % in the SW and up to 70 % in the LW. Intercompared are the effects of specific habits, i.e. hexagonal columns, plates, rosettes, aggregates, and spheres. As mentioned, radiative transfer calculations in this study have been carried out under the assumption of a general habit mixture (GHM) due to the lack of in situ measurements of ice crystal habits during the POLSTRACC campaign. It cannot be ruled out that different habit mixes occur in the Arctic, in which pristine crystals, as for example hexagonal plates or columns, play a more dominant role. Nevertheless, we choose the GHM because

the assumption that an ice cloud consists of one habit type only seems also not to be realistic.

The flight and measurement strategies pose additional sources of uncertainty. Slant pseudo-vertical IWC profiles, or even more those with intermediate level flight sections, extend horizontally over several 10 to 100 km and likely do not adequately account for lateral inhomogeneity within the cirrus cloud field. This uncertainty might be reduced with higher vertical speed, at the cost of reduced vertical resolution or sensor accuracy, or better quantified by repeated sampling in different directions or

collocated satellite observations. Similar studies would also greatly benefit from information on potential lower cloud coverage below the cirrus.

Recent and current observational activities will provide additional data on cloud microphysical properties to better represent real Arctic cirrus in future work, aiming for a closure of the physics of particle-scale processes with the radiation field. In this respect, the in situ measurements of the recent CIRRUS-HL (cirrus at high latitudes) mission are very promising. Further

useful contributions are expected to come from ongoing and planned campaign activities.

**Table A1.** Same as in Tab. 2, but for ice clouds with higher optical thicknesses as a result of multiplying the entire profile IWC($z$) with a factor of 5.

| Date | sza | $F_{\text{net,TOA}}$ / $F_{\text{net,BOA}}$ for IWC($z$)$_{\text{meas}}$ | $F_{\text{net,TOA}}$ / $F_{\text{net,BOA}}$ for IWC($z$)$_{\text{const}}$ | $\Delta F_{\text{net,TOA}}$ / $\Delta F_{\text{net,BOA}}$ (measured - constant) |
|---|---|---|---|---|
| | (°) | (W m$^{-2}$) | (W m$^{-2}$) | (W m$^{-2}$) |
| $\alpha_{\text{ocean}}$ (sza), $\varepsilon_{\text{ocean}} = 0.99$ | | | | |
| 25 Jan 2016 | 88 | -107.23 / -51.47 | -110.88 / -50.67 | 3.65 / -0.80 |
| 25 Jan 2016 | 80 | -20.67 / -1.89 | -24.73 / -1.03 | 4.06 / -0.86 |
| 25 Jan 2016 | 72 | 90.56 / 77.13 | 86.28 / 77.91 | 4.28 / -0.78 |
| 25 Jan 2016 | 64 | 225.38 / 181.32 | 220.87 / 181.97 | 4.51 / -0.65 |
| 09 Mar 2016 | 88 | -139.51 / -54.57 | -139.33 / -54.63 | -0.18 / 0.06 |
| 09 Mar 2016 | 80 | -48.44 / -1.91 | -48.66 / -1.89 | 0.22 / -0.02 |
| 09 Mar 2016 | 72 | 65.27 / 83.32 | 64.73 / 83.36 | 0.54 / -0.04 |
| 09 Mar 2016 | 64 | 202.65 / 196.49 | 201.82 / 196.47 | 0.83 / 0.02 |
| $\alpha_{\text{snow}} = 0.85$, $\varepsilon_{\text{snow}} = 1.00$ | | | | |
| 25 Jan 2016 | 88 | -112.31 / -57.59 | -115.96 / -56.78 | 3.65 / -0.81 |
| 25 Jan 2016 | 80 | -59.48 / -45.40 | -63.58 / -44.58 | 4.10 / -0.82 |
| 25 Jan 2016 | 72 | -3.45 / -27.72 | -7.70 / -26.89 | 4.25 / -0.83 |
| 25 Jan 2016 | 64 | 57.36 / -6.18 | 52.97 / -5.37 | 4.39 / -0.81 |
| 09 Mar 2016 | 88 | -145.25 / -61.36 | -145.03 / -61.37 | -0.22 / -0.01 |
| 09 Mar 2016 | 80 | -90.55 / -48.52 | -90.78 / -48.51 | 0.23 / -0.01 |
| 09 Mar 2016 | 72 | -37.16 / -29.60 | -37.75 / -29.56 | 0.59 / -0.04 |
| 09 Mar 2016 | 64 | 18.82 / -6.41 | 17.97 / -6.36 | 0.85 / -0.05 |

# Appendix A: Appendix

This appendix includes a table similar to Tab. 2, showing further results from radiative transfer calculations that were carried out with IWC profiles scaled by a factor 5, as described in the main text.

*Data availability.* In situ observational data are available at the HALO database: https://halo-db.pa.op.dlr.de/mission/3. ECMWF model data are available at https://www.ecmwf.int. Results from the radiative transfer calculations are available at https://doi.org/10.5281/zenodo.7387075.


*Author contributions.* AM and RM developed the concept for and conducted the study. They also drafted the manuscript. RM executed the radiative transfer calculations. SK, RH, TJ, AM, CV, MK and CR performed the in situ measurements. All authors contributed to the intense discussion and to the writing of the manuscript.

*Competing interests.* The authors declare that they have no competing interests.

*Acknowledgements.* The authors are grateful to Silke Groß, Yi Huang and three anonymous referees for their reviews of the manuscript, that helped to improve the manuscript substantially. The authors thank Sonja Gisinger for kindly preparing the EMCWF and CAMS data.

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
