# Peer review of "Investigating the radiative effect of Arctic cirrus measured in situ during the winter 2015/2016"

_Atmospheric Chemistry and Physics, 2022_

## Referee Comment (RC1)

The authors are to be congratulated on showing the thin ice cloud radiative effects over the polar region using two profiles obtained from aircraft measurements during the Polar Stratosphere in a Changing Climate (POLSTRACC) campaign. I enjoyed reading this manuscript, particularly the instrumentation and data processing section which I am unfamiliar with. Obviously, only two ice water content (IWC) profiles cannot fully describe the cloud variability over the Arctic. However, by making use of the two profiles, the authors study the sensitivities of cloud radiative effects to solar zenith angle and surface albedo variations. The authors also study the difference in computed cloud radiative effects between using IWC aircraft measurement and optically equivalent constant IWC. How the aircraft measurement data are used to prepare the ice cloud description for the radiation computations is nicely presented in detail. All the assumptions made in the radiation computations are clearly stated. The only problem I have is that the computed longwave irradiance variation within the cloud layer looks strange to me and appears to contrast with the results in numerous previous studies. Hence, I would like to suggest the authors double check the longwave radiation computations in this study. Beyond this problem, this manuscript is clear, organized, and well-written and I would suggest it be accepted for publication after some revisions if needed. My specific comments are as follows:

**Major comments:**

Lines 171-173. Is the air in the (polar) stratosphere generally descending? What is the situation in the troposphere, particularly the upper troposphere that is of interest in this study? Is the air in the troposphere also generally descending? If the upper tropospheric air was also descending during the two months of interest, why do you think the mass accumulation near the tropopause during the two months was a result of the descending motion in the stratosphere? If the steady descending motion in the stratosphere exists and persists, will the static stability of the tropopause become increasingly higher as time goes by?

Figure 12. Numerous studies have shown that cloud longwave radiative heating/cooling rate have a vertical gradient from cloud base to cloud top, i.e., heating at the cloud base and cooling at the cloud top (e.g., Fu et al., 1997; Ren et al., 2020, 2021; Wall et al., 2020). However, the red curves in Fig. 12(a) and 12(d) show the opposite, i.e., longwave radiative cooling at the cloud base and heating at the cloud top. I used the longwave version of the rapid radiative transfer model (RRTM; Iacono et al., 2000) with scattering included (Tang et al., 2018) to do a quick check. The resultant longwave irradiance ($I_{LW}$) and radiative heating/cooling rate ($H_{LW}$) profiles are shown in the figure below:

[Figure]

Figure C1. Longwave irradiance ($I_{LW}$) and radiative heating/cooling rate ($H_{LW}$) profiles in an RRTM experiment.

In this RRTM experiment, a homogeneous ice cloud layer with constant IWC of 0.0032 g m$^{-3}$ is placed between 7.25 and 12 km; a constant effective radius of 23 $\mu$m is assumed; a subarctic winter atmospheric profile is adopted with surface temperature set to 280 K and surface emissivity set to 1. Such settings in the RRTM experiment resemble the case of January 25, 2016 in this study. As shown in the above figure, the decrease of $I_{LW}$ with height is slower in the lower portion of the cloud layer, whereas in Fig. 12(a) and 12(d) the decrease of $I_{LW}$ with height is slower in the upper portion of the cloud layer. Would it be helpful to double check your longwave radiation computations? What radiative transfer solver in libRadtran did you use for the longwave radiation computations? Will the $I_{LW}$ result change if you switch to using another solver, such as DISORT?

References

Fu, Q., Liou, K., Cribb, M., Charlock, T., & Grossman, A. (1997). Multiple scattering parameterization in thermal infrared radiative transfer. *Journal of the Atmospheric Sciences, 54*(24), 2799-2812. https://doi.org/10.1175/1520-0469(1997)054<2799:MSPITI>2.0.CO;2

Iacono, M. J., Mlawer, E. J., Clough, S. A., & Morcrette, J. J. (2000). Impact of an improved longwave radiation model, RRTM, on the energy budget and thermodynamic properties of the NCAR community climate model, CCM3. *Journal of Geophysical Research: Atmospheres, 105*(D11), 14873-14890. https://doi.org/10.1029/2000JD900091

Ren, T., Li, D., Muller, J., & Yang, P. (2021). Sensitivity of radiative flux simulations to ice cloud parameterization over the equatorial western Pacific Ocean region. *Journal of the Atmospheric Sciences, 78*(8), 2549-2571. https://doi.org/10.1175/JAS-D-21-0017.1

Ren, T., Yang, P., Schumacher, C., Huang, X., & Lin, W. (2020). Impact of cloud longwave scattering on radiative fluxes associated with the Madden-Julian Oscillation in the Indian

Ocean and Maritime Continent. *Journal of Geophysical Research: Atmospheres, 125*(13), e2020JD032591. https://doi.org/10.1029/2020JD032591

Tang, G., Yang, P., Kattawar, G. W., Huang, X., Mlawer, E. J., Baum, B. A., & King, M. D. (2018). Improvement of the simulation of cloud longwave scattering in broadband radiative transfer models. *Journal of the Atmospheric Sciences, 75*(7), 2217–2233. https://doi.org/10.1175/JAS-D-18-0014.1

Wall, C. J., Norris, J. R., Gasparini, B., Smith Jr, W. L., Thieman, M. M., & Sourdeval, O. (2020). Observational evidence that radiative heating modifies the life cycle of tropical anvil clouds. *Journal of Climate, 33*(20), 8621-8640. https://doi.org/10.1175/JCLI-D-20-0204.1

**Minor comments:**

Line 29. "trough" or "through"?

Line 80. It looks "CIRRUS-HL" is a contraction, but it is not spelled out when it first appears in this article here.

Lines 102-104. Is water vapor mixing ratio much higher than the saturation mixing ratio in a homogenous ice nucleation environment? I notice that as shown in Fig. 3 the POLSTRACC cirrus measurements were taken at temperatures between 195 and 250 K. Do you know if there are previous studies of the importance of homogenous vs. heterogenous ice nucleation in the polar atmosphere with this temperature range?

Line 118. It looks "FISH" is a contraction, but it is not spelled out when it first appears in this article here.

Line 174. Does the "latter" refer to "dynamical tropopause"? Why is the dynamical tropopause a transport barrier of air masses?

Lines 175-176. You mean the observed cirrus clouds above the dynamical tropopause will eventually become polar stratospheric clouds (PSCs)? How are PSCs defined? What are the criteria used to judge whether a cloud is a PSC?

Lines 180-182. Is the thermal tropopause the local temperature minimum? However, as shown in Fig. 4, what the authors refer to as "stratospheric cirrus" show the lowest temperatures, making me wonder if these "stratospheric cirrus" clouds are also below the thermal tropopause?

Lines 298-299. The agreement between aircraft and reanalysis water vapor data surprises me. This result suggests that the quality of ECMWF IFS analysis water vapor data is very good, at least over Northern Europe.

Lines 355-357. These two sentences read awkward to me. Correct me if I am wrong, basically you wanted to say increased optical thicknesses of ice clouds increase the cloud LW effect (forcing), i.e., making more surface emitted LW radiation absorbed by ice clouds and hence $F_{net,TOA,LW}$ less negative?

Lines 362-363. I cannot understand this sentence "a transition that is shifted towards larger sza when compared to the curves in Figs. 8a, b".

Line 419. "shadowing effects"? What do you mean by "shadowing effects"? Didn't you set the cloud fraction to 1 in each cloud layer in your radiation computations? In other words, every cloud layer is overcast in your computations, isn't it? If so, cloud overlapping does not matter in your computations.

Lines 421-422. Does this sentence talk about the case on March 9, 2016?

---

## Referee Comment (RC2)

Review of ACP manuscript "acp-2022-395"
**Title**:      Investigating the cloud radiative effect of Arctic cirrus
**Authors**:  Andreas Marsing et al.

**General comments**

This is an interesting study. However, I would think it requires major revisions before it can be considered for publication in ACP. In this paper, IWC profiles of two specific cases are used to investigate the sensitivity of cirrus radiative effects on different parameters using radiative transfer simulations. The resulting dependence of the radiative effects of the cirrus on solar zenith angle and surface albedo is not particularly exciting, this is almost common knowledge. What I found original is the comparison of simulations assuming homogeneous clouds vs. using IWC profiles measured with high vertical resolution. I would consider this the most interesting part of the paper. Therefore, the study could become even more relevant, when additional artificial profiles would be considered. The paper already shows, that measured profiles result in different radiative effects compared to homogeneous cirrus. However, the question is how representative the few measured profiles are. How would the results change if IWC increases/decreases with altitude? This could lead to some more general conclusions of the paper.

**Specific comments**

   1.    Introduction
- The title should already indicated, that this is a case study. Otherwise the reader expects more than the paper can deliver.
- The introduction mixes state of the art with data description and even already conclusions from this study. This is confusing and should be clearly separated.
- Line 15: No numbers are given to quantify the importance of the cirrus radiative effect. Switching from negative to positive is important, but not, if we switch between +- 0.01 W m$^{-2}$
- Line 32: What means "unavoidable"?
- Line 63: Flight hours depend on flight speed and do not reflect the amount and representativeness of the data. Also different instruments sample different volumes during the same time. Horizontal distance, sampled volume, total particle number would be more meaningful.
- Line 68: What means "thin"? IWP or optical thickness measurements should be quoted here. Or you characterize the general cirrus conditions during the campaign earlier.
- Line 73: This is an example of mixing conclusions into the Introduction. This was not shown yet. Needs to be demonstrated.
- Lines 79-80: should be part of the conclusion section.

   2.  In situ measurements of IWC
- Line 86: How do you measure IWC? The first two paragraphs read as if WARAN is measuring water vapor only. First introduce the concept, which allows also to derive IWP from the instrument (heating, evaporation, etc)
- Line 103: This means, that you assume saturated air within the cirrus. Provide a justification, that this assumption is valid in most of the cases.
- Line 110: What is the inlet velocity and how large is it? Is this measured in parallel or is this combining all the enhancement effects?
- Line 118: Why can FISH serve as a reference? Is it more precise? Why not using FISH data for this study?

3. Statistics from Arctic cirrus sampling

- Fig. 3: Add the legend from the original publication.
- Line 142 ff: Are measurements in mid-latitudes included in this "Arctic" study? Below in the text it is separated. But I suggest to remove all mid-latitude data from the beginning. The study aims for Arctic cirrus. So there is no need to discuss mid-latitude cirrus in this paper.
- Extending Fig. 3 might also not be the main objective of this study. Consider to focus on the Arctic cirrus.
- Line 162: The analysis of Fig. 4 should be better motivated with respect to the main objective of the paper. e.g.: temperature might affect emission - thermal IR radiative effect.
- Line 166: Explain PVU, please.

4. Complete profiles of cirrus at the high latitude tropopause
- Line 204: These very thin cirrus clouds are common for the Arctic: Really? Only 4 profiles are from Arctic locations.
- Table 1: Leave out the mid-latitude cirrus, give IWC in $g\ m^{-3}$
- Line 210: What do 18 min mean in distance?

5. Radiative transfer calculations
- First paragraph: Partly a repetition of the introduction. As a reader, I expect a focused and detailed description of what has been done in this section and not another general overview.
- Line 253: The radiative transfer solver is not given in this section. DISORT or two-stream? For SW: UV radiation is neglected? How large is the contribution of UV, which is not covered by the simulations?
- Line 258: More details on this parametrization are needed. What does $R_{eff}$ depend on? Is the Liou parametrization consistent with the parameterization of optical properties by Baum et al?
- Lines 261-262: Even if there are no in situ measurements of crystal shapes available for the cases investigated here, shape effects should at least be discussed. You may refer to the following two papers that would fit here:
    o For the thermal-infrared: Wendisch, M., P. Yang, and P. Pilewskie, 2007: Effects of ice crystal habit on thermal infrared radiative properties and forcing of cirrus. J. Geophys. Res., 112, D08201, doi:10.1029/2006JD007899
    o For the solar: Wendisch, M., P. Pilewskie, J. Pommier, S. Howard, P. Yang, A. J. Heymsfield, C. G. Schmitt, D. Baumgardner, and B. Mayer, 2005: Impact of cirrus crystal shape on solar spectral irradiance: A case study for subtropical cirrus. J. Geophys. Res., 110, D03202, doi:10.1029/2004JD005294.
- Line 276: It seems, that broadband albedo data of fixed values are used. In that case, I don't understand, why not simply changing surface albedo in 0.1 steps between 0 and 0.9? Such simulations can be attributed to specific surface types afterwards.
- Line 281: Snow and ocean do not reach an emissivity of 1.0. There is no need to make this approximation.
- Line 286: It is not clear, what temperature and humidity data are finally used for the simulations. HALO, IFS or GEMS? Or did you merge the data?
- Line 300: Are the measurements only used to validate the model or are they merged with the model for the final input of the UVSPEC?
- Eq. 4: I always thought that net irradiance is the difference between downward and upward irradiance. I understand that you define upward to be negative, this seems odd, at least from a measurement point of view (negative irradiances?).

- Line 315: Using symbol "*I*" for irradiance is uncommon. Now you use "*F*" as a second symbol for the same quantify. Net irradiance has the same unit as the irradiance. I recommend to use only one symbol.
- Line 333: What is the motivation to use different surface albedo for the second case? Hard to compare the impact of IWP of the two clouds, when the albedo is different.
- Line 353: Both cirrus have almost the same optical thickness. What is the gain to analyze a second case with similar optical thickness? What was different between both days? Should be motivated when introducing the cases.
- Line 369: To me the conclusion looks different: There is simply no significant solar radiative effect by the cirrus, if the surface albedo is high. Thus, no SZA dependence is visible.
- Fig. 11 very similar to Fig. 6, please combine them.
- Line 392: similar discussion for Fig. 6.
- Line 393: Temperature profile is not discussed. But this might have an impact on LW radiation.
- Line 438: 4.47 k day-1: What is the layer thickness the heating rates are related to? The absolute values strongly depend on this. Without knowing the thickness, the values are not comparable to other studies.
- Fig. 12: The heating rate profiles do not seem to fit to the net irradiance profiles. Decrease of net irradiance should always result in a negative heating rate. Or is this only the cloud contribution to the heating rate?
- Line 446: This should be discussed with the differences between solar and IR wavelength. The heating rates likely are dominated by the IR irradiances. There is no chance, that the surface albedo makes a big difference.
- Line 449: If Fig. A3 is discussed here in the main part of the paper, then the figure should also be placed within the main text. Readers should not be forced to move forward to the appendix while reading.

**6  Discussion and outlook**
- Why no conclusions?
- Discuss, if the measurement strategy/flight pattern affects the conclusions given here. The IWC profiles are not measured at a single location and might be affected by horizontal inhomogeneity of the cirrus. This should be mentioned and discussed here.
- The effect of the temperature profiles and the location of the cirrus with the profile was not discussed. As the longwave heating is dominating over the solar effect, this is more important than changing surface albedo.

**Appendix**
- As mentioned above: I would prefer to have the plots in the main text, where they are obviously needed.

---

## Referee Comment (RC4)

ACPD MS No.: acp-2022-395

Title: Investigating the cloud radiative effect of Arctic cirrus

Author(s): Andreas Marsing, Ralf Meerkötter, Romy Heller, Stefan Kaufmann, Tina Jurkat-Witschas, Martina Krämer, Christian Rolf, and Christiane Voigt

MS type: Research article
Iteration: Initial submission

General Comments:

The cloud radiative effect (CRE) of cirrus clouds tends to be strongest in the Polar Regions since cirrus cloud emissivity tends to be greater than the corresponding albedo, and longwave (LW) radiation tends to dominate over shortwave (SW) radiation in the Polar Regions.  This gives Arctic cirrus a potentially elevated status in terms of radiative impact on climate.  Moreover, cirrus clouds having visual optical depths $\tau_{vis}$ between 0.3 and 3.0 have the greatest frequency of occurrence (Hong and Liu, 2015, JClim), have a CRE representative of cirrus clouds overall (Hong and Liu, 2016, JClim), and appear to be most abundant in the Arctic during winter (DJF; Mitchell et al., 2018).  Thus, the CRE of winter Arctic cirrus might be particularly strong, making the topic of this journal submission of interest.

However, this manuscript was written with a focus on SW radiation with LW radiation arguably secondary in importance.  While the SW radiation is more interesting in many respects, the uniqueness of Arctic cirrus in terms of LW radiation should not be ignored.  In the results section, it might be instructive to show net irradiance for these surface albedo (and cloudy vs. clear) conditions as a function of time over a 24 hour period.  Relating TOA $F_{net}$ (same as CRE) to solar zenith angle is fine but this focus might detract from the fact that most of the time during Arctic winter the sun is not present and $F_{net}$ is determined only by LW radiation.  A representative latitude (based on in situ sampling) could be selected for this.  This would add perspective for those readers seeking a more representative understanding of Arctic cirrus radiative effects.

The paper is well written and organized and presents results that appear to be unique.  After some minor revisions, it should be appropriate for publication in ACP.

Major Comments:

1.  Figure 9:  The results in Fig. 9 (especially 9a) appear to contradict the results in Fig. 17 of Hong and Liu (2015, J. Climate), where $F_{net}$ at the surface is comparable with TOA $F_{net}$ for the same $\tau_{vis}$ used here.  Please attempt to explain this discrepancy.

2. Lines 352-354:  There are evidently some errors in this sentence.  The visible optical depths ($\tau_{vis}$) for the Jan. and March case studies are 0.65 and 0.68, respectively (line 265) but here it says both $\tau_{vis}$ are identical.  Moreover, $\tau_{vis}$ = 3 IWP/($\rho_i D_e$), and multiplying the IWC profiles by a factor of 5 should also increase IWP by this factor, and thus increase $\tau_{vis}$ by a factor of 5.  That being so, the 5-fold $\tau_{vis}$ stated for these two case studies should be 3.25 and 3.40 (not 2.94 and 2.85 as stated in the text).

3. Lines 379-380:  Note this is due only to changes in SW radiation.  Please provide an explanation to conceptually understand this.  For example, is this due to the greater "effective" optical depth of the cirrus when incident reflected SW radiation enters cloud base at oblique angles?

4. Lines 382-384:  But $\tau_{vis}$ is almost the same for both case studies (0.65 vs. 0.68).  Are you sure that a 0.03 change in $\tau_{vis}$ can account for the shift in the snow albedo curves?

Technical Comments:

1. Line 29: trough => through?

2. Figure 9:  Fig. 8 => Fig. 8a,b?

---

## Author Comment (AC1)

**Response to peer reviews**

of the manuscript ACP-2022-395: "Investigating the cloud radiative effect of Arctic cirrus" by Marsing, Meerkötter et al.

*Dear reviewers and dear editor,*

*We thank all reviewers for their careful assessment of the manuscript. The comments raised several important open questions and greatly helped in streamlining the overall analysis, presentation and findings of our study.*

*Below you find part of our authors' answers to the reviews. The reason why this is not complete at this point is twofold: First, as we checked several results of the radiative transfer calculations using the DISORT solver instead of two-stream (used before), we originally did not notice much difference. However, in the last few days, we found that deviations are stronger and more prevalent than thought, leading us to redo all RT calculations. We would like to be consistent and careful here. This has been taking some time, especially to transfer the results to all figures and to adapt all text and answers. Second, just now we stepped into some technical issues due to maintenance work on our instute network, which renders access to all necessary files quite difficult.*

*Therefore, we would like to hand in the completed answers and the revised manuscript in the course of the next week. We hope and ask for your generous understanding to the delay. This has also been announced beforehand to the editor.*

*Yours sincerely,*

*Andreas Marsing, on behalf of all authors*

*The reviews are written in black. The authors' response to the reviews is given below each paragraph in blue italic font.*

**Review by anonymous reviewer #1**

The authors are to be congratulated on showing the thin ice cloud radiative effects over the polar region using two profiles obtained from aircraft measurements during the Polar Stratosphere in a Changing Climate (POLSTRACC) campaign. I enjoyed reading this manuscript, particularly the instrumentation and data processing section which I am unfamiliar with. Obviously, only two ice water content (IWC) profiles cannot fully describe the cloud variability over the Arctic. However, by making use of the two profiles, the authors study the sensitivies of cloud radiative effects to solar zenith angle and surface albedo variations. The authors also study the difference in computed cloud radiative effects between using IWC aircraft measurement and optically equivalent constant IWC. How the aircraft measurement data are used to prepare the ice cloud description for the radiation computations is nicely presented in detail. All the assumptions made in the radiation computations are clearly stated. The only problem I have is that the computed longwave irradiance variation within the cloud layer looks strange to me and appears to contrast with the results in numerous previous studies. Hence, I would like to suggest the authors double check the longwave radiation computations in this study. Beyond this problem, this manuscript is clear, organized, and well-written and I would suggest it be accepted for publication after some revisions if needed.

*We thank the reviewer for his/her kind words and the positive assessment of the manuscript. The concern regarding our longwave irradiance profiles is addressed in the respective paragraph below.*

My specific comments are as follows:

**Major comments:**

Lines 171-173. Is the air in the (polar) stratosphere generally descending? What is the situation in the troposphere, particularly the upper troposphere that is of interest in this study? Is the air in the troposphere also generally descending? If the upper tropospheric air was also descending during the two months of interest, why do you think the mass accumulation near the tropopause during the two months was a result of the descending motion in the stratosphere? If the steady descending motion in the stratosphere exists and persists, will the static stability of the tropopause become increasingly higher as time goes by?

*In a zonal and multi-annual mean, there is a generally descending motion in the winter and spring polar stratosphere, which has also been observed, e.g. in terms of $N_2O$, in the winter 2015/2016 (Manney and Lawrence, 2016; Birner, 2010). In contrast, tropospheric eddies counteract descending motion (Birner, 2010), which also stabilize the height of the tropopause. Birner also points out how these opposing effects lead to the increasing static stability above the tropopause. We include these references in the manuscript.*

Figure 12. Numerous studies have shown that cloud longwave radiative heating/cooling rate have a vertical gradient from cloud base to cloud top, i.e., heating at the cloud base and cooling at the cloud top (e.g., Fu et al., 1997; Ren et al., 2020, 2021; Wall et al., 2020). However, the red curves in Fig. 12(a) and 12(d) show the opposite, i.e., longwave radiative cooling at the cloud base and heating at the cloud top. I used the longwave version of the rapid radiative transfer model (RRTM; Iacono et al., 2000) with scattering included (Tang et al., 2018) to do a quick check. The resultant longwave irradiance ($I_{LW}$) and radiative heating/cooling rate ($H_{LW}$) profiles are shown in the figure below:

[Figure]

Figure C1 Longwave irradiance ($I_{LW}$) and radiative heating/cooling rate ($H_{LW}$) profiles in an RRTM experiment.

In this RRTM experiment, a homogeneous ice cloud layer with constant IWC of 0.0032 g m$^{-3}$ is placed between 7.25 and 12 km; a constant effective radius of 23 μm is assumed; a subarctic winter atmospheric profile is adopted with surface temperature set to 280 K and surface emissivity set to 1. Such settings in the RRTM experiment resemble the case of January 25, 2016 in this study. As shown in the above figure, the decrease of $I_{LW}$ with height is slower in the lower portion of the cloud layer, whereas in Fig. 12(a) and 12(d) the decrease of $I_{LW}$ with height is slower in the upper portion of the

cloud layer. Would it be helpful to double check your longwave radiation computations? What radiative transfer solver in libRadtran did you use for the longwave radiation computations? Will the ILW result change if you switch to using another solver, such as DISORT?

*We thank the reviewer for the careful investigation of the LW irradiance profile. We suppose that the difference in the radiative transfer calculations results from the used atmospheric temperature profiles. In our original calculations, we tried to resemble the situation at the time of cirrus sampling as closely as possible by using temperature profiles from the ECMWF IFS model (as explained in the manuscript). In Figure A2 in this answer, we show on the left panel how the profile from the 25 January case (black) differs from the ==standard== subarctic winter profile (light blue) or from other synthetic profiles. The right panel gives the corresponding LW irradiance profiles. We observe that our $I_{LW}$ profile in the subarctic case closely resembles that from reviewer #1. We deduce that especially the strong inversion at the tropopause, which is only present in the "real" profile, causes the deviation in the irradiance profile, which naturally propagates into the heating rate profile. ==To be added: Sentence on RT solver.==*

[Figure]

*Figure A2 Left: temperature profiles, i.e. original POLSTACC profile (black) plus further synthetic profiles (red, dark blue) as well as the profile of the subarctic winter standard atmosphere (light blue). Right: corresponding LW irradiance profiles. Curves explain that the pronounced temperature inversion in the POLSTRACC profile produces the significant difference to the irradiance profile calculated by referee #1.*

References

Fu, Q., Liou, K., Cribb, M., Charlock, T., & Grossman, A. (1997). Multiple scattering parameterization in thermal infrared radiative transfer. Journal of the Atmospheric Sciences, 54(24), 2799-2812. https://doi.org/10.1175/1520-0469(1997)0542.0.CO;2

Iacono, M. J., Mlawer, E. J., Clough, S. A., & Morcrette, J. J. (2000). Impact of an improved longwave radiation model, RRTM, on the energy budget and thermodynamic properties of the NCAR community climate model, CCM3. Journal of Geophysical Research: Atmospheres, 105(D11), 14873-14890. https://doi.org/10.1029/2000JD900091

Ren, T., Li, D., Muller, J., & Yang, P. (2021). Sensitivity of radiative flux simulations to ice cloud parameterization over the equatorial western Pacific Ocean region. Journal of the Atmospheric Sciences, 78(8), 2549-2571. https://doi.org/10.1175/JAS-D-21-0017.1

Ren, T., Yang, P., Schumacher, C., Huang, X., & Lin, W. (2020). Impact of cloud longwave scattering on radiative fluxes associated with the Madden-Julian Oscillation in the Indian Ocean and Maritime Continent. Journal of Geophysical Research: Atmospheres, 125(13), e2020JD032591. https://doi.org/10.1029/2020JD032591

Tang, G., Yang, P., Kattawar, G. W., Huang, X., Mlawer, E. J., Baum, B. A., & King, M. D. (2018). Improvement of the simulation of cloud longwave scattering in broadband radiative transfer models. Journal of the Atmospheric Sciences, 75(7), 2217–2233. https://doi.org/10.1175/JAS-D-18-0014.1

Wall, C. J., Norris, J. R., Gasparini, B., Smith Jr, W. L., Thieman, M. M., & Sourdeval, O. (2020). Observational evidence that radiative heating modifies the life cycle of tropical anvil clouds. Journal of Climate, 33(20), 8621-8640. https://doi.org/10.1175/JCLI-D-20-0204.1

**Minor comments:**

Line 29. "trough" or "through"?

*"Through" is probably bad expression. We change it to "from".*

Line 80. It looks "CIRRUS-HL" is a contraction, but it is not spelled out when it first appears in this article here.

*We set a parenthesis to better express that CIRRUS-HL stands for cirrus in high latitudes. However, the occurrence of this acronym is now moved to the end of the manuscript.*

Lines 102-104. Is water vapor mixing ratio much higher than the saturation mixing ratio in a homogenous ice nucleation environment? I notice that as shown in Fig. 3 the POLSTRACC cirrus measurements were taken at temperatures between 195 and 250 K. Do you know if there are previous studies of the importance of homogenous vs. heterogenous ice nucleation in the polar atmosphere with this temperature range?

*The homogeneous freezing threshold in the relevant temperature range varies between about 150 – 165 % in terms of relative humidity with respect to ice ($RH_{ice}$) (e.g. Krämer et al., 2016), so there can indeed be considerable super-saturation in a homogeneous nucleation scenario. The prevalence of homogeneous or heterogeneous ice nucleation depends primarily on the updraft speed of the humid air mass (Krämer et al., 2020) associated with the meteorological situation. In polar regions, this includes primarily low and high pressure systems and warm conveyor belts (slow updrafts) and gravity waves and jet streams (fast updrafts). Without further analysis of the flow conditions and history, this may adversely affect the accuracy of the IWC values, which Is what we suppose to be the reviewer's main concern. We provide resolutions to this in the specific answers to reviewers #3 and #4.*

Line 118. It looks "FISH" is a contraction, but it is not spelled out when it first appears in this article here.

*We include the full name "Fast In-situ Stratospheric Hygrometer" in the text.*

Line 174. Does the "latter" refer to "dynamical tropopause"? Why is the dynamical tropopause a transport barrier of air masses?

*In many cases the dynamical tropopause is a transport barrier, but not necessarily in convection or when radiative heating or cooling takes place as it is the case inside clouds. To avoid confusion, we omit the half sentence "With the latter acting as a transport barrier of air masses".*

Lines 175-176. You mean the observed cirrus clouds above the dynamical tropopause will eventually become polar stratospheric clouds (PSCs)? How are PSCs defined? What are the criteria used to judge whether a cloud is a PSC?

*We rephrase this sentence, as it might be confusing. There are several PSC types, and some consist of water ice or contain a high fraction of water ice. In that sense, it is safe to say that ice clouds above the polar tropopause reflect a lower branch of PSCs.*

Lines 180-182. Is the thermal tropopause the local temperature minimum? However, as shown in Fig. 4, what the authors refer to as "stratospheric cirrus" show the lowest temperatures, making me wonder if these "stratospheric cirrus" clouds are also below the thermal tropopause?

*We clarify upon this by explicitly referring to the WMO tropopause. Accordingly, the thermal tropopause or lapse rate tropopause is the lowest level at which the lapse rate falls below 2 K km$^{-1}$ (with some additional constraints). Therefore, the thermal tropopause is (sometimes considerably) lower (in altitude) than the local temperature minimum.*

Lines 298-299. The agreement between aircraft and reanalysis water vapor data surprises me. This result suggests that the quality of ECMWF IFS analysis water vapor data is very good, at least over Northern Europe.

*We clarify on this a little bit more in the text, mentioning that we compare only exemplary sections of data near our cloud profiles of interest (not within, as we have no humidity data there). Indeed we find exceptional agreement there, but we cannot generalize this observation to a broader domain.*

Lines 355-357. These two sentences read awkward to me. Correct me if I am wrong, basically you wanted to say increased optical thicknesses of ice clouds increase the cloud LW effect (forcing), i.e., making more surface emitted LW radiation absorbed by ice clouds and hence Fnet,TOA,LW less negative?

*To be added*

Lines 362-363. I cannot understand this sentence "a transition that is shifted towards larger sza when compared to the curves in Figs. 8a, b". Line 419. "shadowing effects"? What do you mean by "shadowing effects"? Didn't you set the cloud fraction to 1 in each cloud layer in your radiation computations? In other words, every cloud layer is overcast in your computations, isn't it? If so, cloud overlapping does not matter in your computations.

*To be added*

Lines 421-422. Does this sentence talk about the case on March 9, 2016?

*To be added*

**Review by anonymous reviewer #2**

**General comments**

This is an interesting study. However, I would think it requires major revisions before it can be considered for publication in ACP. In this paper, IWC profiles of two specific cases are used to investigate the sensitivity of cirrus radiative effects on different parameters using radiative transfer

simulations. The resulting dependence of the radiative effects of the cirrus on solar zenith angle and surface albedo is not particularly exciting, this is almost common knowledge. What I found original is the comparison of simulations assuming homogeneous clouds vs. using IWC profiles measured with high vertical resolution. I would consider this the most interesting part of the paper. Therefore, the study could become even more relevant, when additional artificial profiles would be considered. The paper already shows, that measured profiles result in different radiative effects compared to homogeneous cirrus. However, the question is how representative the few measured profiles are. How would the results change if IWC increases/decreases with altitude? This could lead to some more general conclusions of the paper.

*We thank the reviewer for his/her thorough assessment of the manuscript and for the helpful comments. Taking all the reviewers' suggestions into account, we decided to follow reviewer #2nd's suggestion in part to expand the radiative effect study by a few more artificial profiles. The added "storyline" shows how the transition from simple to ever more realistic IWC profiles translates into the radiative effects at TOA, BOA, in-cloud. To be completed*

**Specific comments**

1. Introduction
   - The title should already indicate, that this is a case study. Otherwise the reader expects more than the paper can deliver.
     *This is something that had also gone through our minds. We agreed and change the title: The cloud radiative effect of Arctic cirrus measured in situ during the POLSTRACC campaign*
   - The introduction mixes state of the art with data description and even already conclusions from this study. This is confusing and should be clearly separated. *To be added.*
   - Line 15: No numbers are given to quantify the importance of the cirrus radiative effect. Switching from negative to positive is important, but not, if we switch between +- 0.01 W m$^{-2}$
     *This is part of the abstract and we need to provide a number here as soon as the final numbers are available.*
   - Line 32: What means "unavoidable"?
     *We remove the possibly confusing adjectives.*
   - Line 63: Flight hours depend on flight speed and do not reflect the amount and representativeness of the data. Also different instruments sample different volumes during the same time. Horizontal distance, sampled volume, total particle number would be more meaningful.
     *As suggested, we provide a distance, whereas meaningful values on sampled volume and particle number cannot be given. Still, flight hours are a common and interesting measure in the community to get an idea of the extent of a field campaign.*
   - Line 68: What means "thin"? IWP or optical thickness measurements should be quoted here. Or you characterize the general cirrus conditions during the campaign earlier.
     *We agree that the handwavy "thin" characterization is not adequate here. We elaborate on IWP and optical thickness later.*
   - Line 73: This is an example of mixing conclusions into the Introduction. This was not shown yet. Needs to be demonstrated.
     *We rephrase the sentence and move the finding to the conclusions. This should better express what we wanted to say, the intersection with and difference to the Feofilov study.*

- Lines 79-80: should be part of the conclusion section.

  *We move this to the conclusions.*

2. In situ measurements of IWC
   - Line 86: How do you measure IWC? The first two paragraphs read as if WARAN is measuring water vapor only. First introduce the concept, which allows also to derive IWP from the instrument (heating, evaporation, etc)

     *We adapt the storyline to begin with measured total water content (TWC).*

   - Line 103: This means, that you assume saturated air within the cirrus. Provide a justification, that this assumption is valid in most of the cases.

     *We acknowledge the lack of a justification why we pursue our study with such a simple assumption. We are well aware that (near) saturation is by far not the only scenario of relative humidity inside cirrus. Therefore, we change the text to explicitly quote this knowledge and explain why the impact on IWC values is less severe than one might think. The reason is the enhancement of ice particles in the sample flow.*

   - Line 110: What is the inlet velocity and how large is it? Is this measured in parallel or is this combining all the enhancement effects?

     *We give a number for the inlet velocity, which is very precisely known using the hygrometer's cell pressure and either a mass flow controller or a critical orifice with known characteristics. However, we assume that explanation of this detail is of lesser interest for the manuscript.*

   - Line 118: Why can FISH serve as a reference? Is it more precise? Why not using FISH data for this study?

     *FISH is not a reference, but another instrument that can provide IWC in a similar way. As an earlier comparison of both measurements showed a systematic bias employing data from an earlier campaign (Afchine et al., 2018), we repeat the comparsion which now shows substantially better agreement. This gives confidence to the accuracy of both instruments, but especially to WARAN. The precision in the studied range of TWC values is comparable, so there is no reason in this respect as to whether one instrument should e preferred over the other.*

3. Statistics from Arctic cirrus sampling
   - Fig. 3: Add the legend from the original publication.

     *We add the original legend.*

   - Line 142 ff: Are measurements in mid-latitudes included in this "Arctic" study? Below in the text it is separated. But I suggest to remove all mid-latitude data from the beginning. The study aims for Arctic cirrus. So there is no need to discuss mid-latitude cirrus in this paper.

     *We would like to keep the overview of mid-latitude cirrus as this dataset has not been shown elsewhere. Also, many important climatological studies of cirrus make a lesser separation along latitudes. As this study is founded in part on this common knowledge, and to give context to the measurements and why they are as they are, we would like to keep them in at this point.*

   - Extending Fig. 3 might also not be the main objective of this study. Consider to focus on the Arctic cirrus.

     *One of the objectives of the POLSTRACC campaign was to extend the in situ data on cirrus in this temperature/latitude parameter space. Therefore, we find it worthwhile to show what could be achieved in this respect, and to give an updated idea of available data.*

   - Line 162: The analysis of Fig. 4 should be better motivated with respect to the main objective of the paper. e.g.: temperature might affect emission - thermal IR radiative effect.

*We agree and rephrase the motivational sentence. In that, we clarify what might be relevant macrophysical properties of cirrus near the tropopause.*

- Line 166: Explain PVU, please.
  *We give the definition of the potential vorticity unit (PVU).*

4. Complete profiles of cirrus at the high latitude tropopause
   - Line 204: These very thin cirrus clouds are common for the Arctic: Really? Only 4 profiles are from Arctic locations.
     *This sentence was misleading. It is rephrased to express that thin cirrus are common for the Arctic (as stated in the literature). The optical thickness of the profiles at hand is assessed elsewhere.*
   - Table 1: Leave out the mid-latitude cirrus, give IWC in g m$^{-3}$
     *We would like to keep the mid-latitude profiles in until here, as motivated above. We give the IWC additionally in mg m$^{-3}$. Both representations (molar mixing ratio and mass concentration) have their justifications.*
   - Line 210: What do 18 min mean in distance?
     *We give the value which is about 230 km.*

5. Radiative transfer calculations
   - First paragraph: Partly a repetition of the introduction. As a reader, I expect a focused and detailed description of what has been done in this section and not another general overview.
     *To be added.*
   - Line 253: The radiative transfer solver is not given in this section. DISORT or two-stream? For SW: UV radiation is neglected? How large is the contribution of UV, which is not covered by the simulations?
     *To be added.*
   - Line 258: More details on this parametrization are needed. What does $R_{eff}$ depend on? Is the Liou parametrization consistent with the parameterization of optical properties by Baum et al?
     *To be added.*
   - Lines 261-262: Even if there are no in situ measurements of crystal shapes available for the cases investigated here, shape effects should at least be discussed. You may refer to the following two papers that would fit here:
     *To be added.*
     o For the thermal-infrared: Wendisch, M., P. Yang, and P. Pilewskie, 2007: Effects of ice crystal habit on thermal infrared radiative properties and forcing of cirrus. J. Geophys. Res., 112, D08201, doi:10.1029/2006JD007899
     o For the solar: Wendisch, M., P. Pilewskie, J. Pommier, S. Howard, P. Yang, A. J. Heymsfield, C. G. Schmitt, D. Baumgardner, and B. Mayer, 2005: Impact of cirrus crystal shape on solar spectral irradiance: A case study for subtropical cirrus. J. Geophys. Res., 110, D03202, doi:10.1029/2004JD005294.
   - Line 276: It seems, that broadband albedo data of fixed values are used. In that case, I don't understand, why not simply changing surface albedo in 0.1 steps between 0 and 0.9? Such simulations can be attributed to specific surface types afterwards.
     *To be added.*
   - Line 281: Snow and ocean do not reach an emissivity of 1.0. There is no need to make this approximation.
     *To be added.*

- Line 286: It is not clear, what temperature and humidity data are finally used for the simulations. HALO, IFS or GEMS? Or did you merge the data?
  *We rephrase some sentences to make clear that we use model (IFS and GEMS) data for everything except from IWC.*
- Line 300: Are the measurements only used to validate the model or are they merged with the model for the final input of the UVSPEC?
  *We use measurements (apart from IWC) only in order to validate the model.*
- Eq. 4: I always thought that net irradiance is the difference between downward and upward irradiance. I understand that you define upward to be negative, this seems odd, at least from a measurement point of view (negative irradiances?).
  *To be added.*
- Line 315: Using symbol "$I$" for irradiance is uncommon. Now you use "$F$" as a second symbol for the same quantify. Net irradiance has the same unit as the irradiance. I recommend to use only one symbol.
  *We thank the reviewer for discovering this inconsistency. We now write "F" everywhere for (net) irradiance.*
- Line 333: What is the motivation to use different surface albedo for the second case? Hard to compare the impact of IWP of the two clouds, when the albedo is different.
  *To be added.*
- Line 353: Both cirrus have almost the same optical thickness. What is the gain to analyze a second case with similar optical thickness? What was different between both days? Should be motivated when introducing the cases.
  *To be added.*
- Line 369: To me the conclusion looks different: There is simply no significant solar radiative effect by the cirrus, if the surface albedo is high. Thus, no SZA dependence is visible.
  *To be added.*
- Fig. 11 very similar to Fig. 6, please combine them.
  *To be added.*
- Line 392: similar discussion for Fig. 6.
  *To be added.*
- Line 393: Temperature profile is not discussed. But this might have an impact on LW radiation.
  *To be added.*
- Line 438: 4.47 k day-1: What is the layer thickness the heating rates are related to? The absolute values strongly depend on this. Without knowing the thickness, the values are not comparable to other studies.
  *To be added.*
- Fig. 12: The heating rate profiles do not seem to fit to the net irradiance profiles. Decrease of net irradiance should always result in a negative heating rate. Or is this only the cloud contribution to the heating rate?
  *To be added.*
- Line 446: This should be discussed with the differences between solar and IR wavelength. The heating rates likely are dominated by the IR irradiances. There is no chance, that the surface albedo makes a big difference.
  *To be added.*
- Line 449: If Fig. A3 is discussed here in the main part of the paper, then the figure should also be placed within the main text. Readers should not be forced to move forward to

the appendix while reading.

*The figure is moved to the main text.*

6. Discussion and outlook
   - Why no conclusions?

     *The section title is misleading. Of course we have conclusions. We rename the section to "Discussion and conclusions" and give an outlook separately afterwards.*
   - Discuss, if the measurement strategy/flight pattern affects the conclusions given here. The IWC profiles are not measured at a single location and might be affected by horizontal inhomogeneity of the cirrus. This should be mentioned and discussed here.

     *Good point, wilco. To be added.*
   - The effect of the temperature profiles and the location of the cirrus with the profile was not discussed. As the longwave heating is dominating over the solar effect, this is more important than changing surface albedo.

     *To be added.*

7. Appendix
   - As mentioned above: I would prefer to have the plots in the main text, where they are obviously needed.

     *All plots now appear in the main text. We decided to leave the last table in the appendix, though.*

**Review by anonymous reviewer #3**

This paper uses the measurements of a hygrometer (WARAN) to infer the water contents of cirrus clouds and then based on the inferred cloud water content to quantify the radiative effects of the cirrus in the arctic region. As the authors correctly state, cirrus clouds are frequent in the arctic and potentially play an important role in influencing the radiative balance in the region. However, it is difficult to ascertain their radiative effect because the effect depends not only on the cloud properties, which are difficult to measure, but sensitively on various environment variables such as solar zenith angle and surface albedo which can affect both the magnitude and sign of the radiative effect. I am convinced the topic and objective of this work are both important and think works like this one that base on data to assess the radiative effect of cirrus clouds in the arctic are much needed and should be encouraged. I also find the paper generally well written, providing a clear documentation of the research steps and results.

Although the research is well motivated, I found several critical issues with this work. These include the quantification of the ice water content and the configuration of the environmental profiles for the radiative assessment. These deficiencies limit the usefulness of this work and should be addressed before the paper is considered for publication.

*We thank the reviewer for his/her encouraging words and for the overall positive assessment of the manuscript. The important point regarding the IWC measurements was raised by all reviewers. We acknowledge that this aspect was not well represented in the original manuscript. Concerning the radiative assessment, and taking into account the other reviewers' comments, we decided to expand the study a bit and tried to make it more relevant to the community, as the reviewer understandably demanded. We elaborate on this in the specific answer below.*

**IWC**

Given that IWC is not directly measured but inferred in the total water measurements. The accuracy of the data are especially in need of validation. I found it unsatisfactory to only present a PDF summary (fig 1) of the WARAN vs FISH comparison, without explaining the different behaviour

documented here compared to the literature (overestimation of WARAN) or analyzing the biases pattern, e.g., under moist vs. dry conditions, at different times of the day (solar angles), association with underlying surface types (albedo), and collocated dynamical fields.

*We rephrase parts of the IWC measurement section in order to make the measurement process and derivation of IWC more traceable.*

*The instrumental paper by Afchine et al. (2018) comprehensively studies the capabilities, limitations, uncertainties and deviations of the used hygrometers and inlet configuration. We go no further than that. Having said that, we were actually quite surprised that the inter-instrumental comparison between FISH and WARAN could still be notably improved compared to Afchine et al. (2018), by means of some basic improvements in calibration and data treatment. This does not reduce the inherent deficiencies, but justifies the use of the relatively simple WARAN instrument for this study.*

Moreover, it seems the authors completely ignored the possibility of ice supersaturation in inferring the ice content from the total water measurement. Given how common the UTLS air is found to be in a supersaturated state and how the ice and supersaturated air are intrinsically related in influencing the radiation fields (e.g., Tan et al. 2016, https://doi.org/10.1002/2016GL071144), this is not acceptable. It is understood that independent data not available from the campaign, but at minimum this issue should be recognized and discussed, preferably using the statistics of the supersaturation or its relation to environment conditions obtained from other campaigns. In this regard, it appears especially hand-wavy, and possibly wrong, to inflate the IWC by 5 times in the radiative assessment.

*We acknowledge the lack of a justification why we pursue our study with such a simple assumption. We are well aware that (near) saturation is by far not the only scenario of relative humidity inside cirrus. Therefore, we change the text to explicitly quote this knowledge and explain why the impact on IWC values is less severe than one might think. The reason is the enhancement of ice particles in the sample flow. Therefore, we also effectively avoid to mistake supersaturation as ice crystals.*

**Radiative assessment**

*To be added.*

The authors correctly recognize that the radiative assessment is sensitive to the environment conditions coexisting with the cirrus, such as the solar angle and surface albedo. However, it doesn't appear logic to me that they extensively use idealized (nominal) values of these parameters rather than best estimates of them from appropriate datasets. Generally speaking, we don't need another set of sensitivity experiments to illustrate how complex the cirrus radiative effects are but are in great need of measurement data to nail down what exact effects are in the nature. The authors need to either provide convincing arguments as to how the sensitivity computations done here are new or useful (how it can be related to nature), or change their strategy and properly pair their cirrus data with the values of those parameters appropriate to the time and location in their assessment.

Also, these aspects of the assessment probably can be better documented or explained:

The configuration of the RT model, e.g., how many streams are used in the RT solver, how the scattering angles are discretized, … these aspects all affect the results. The sensitivity of cirrus effect to the solar zenith angle is not well explained in the current paper; unclear how the scattering angle effect (forward scattering) and light path effect respectively affect the result and which dominates.

Possibility of sub-cirrus cloud layers, which are often found in nature and are expected to strongly affect the assessment of the radiative impacts of cirrus.

Optical depth of the aerosol (haze) layer prescribed – how much does it affect the lower boundary reflectance, and how are the cirrus effect depends on this factor.

**Review by anonymous reviewer #4**

**General Comments:**

The cloud radiative effect (CRE) of cirrus clouds tends to be strongest in the Polar Regions since cirrus cloud emissivity tends to be greater than the corresponding albedo, and longwave (LW) radiation tends to dominate over shortwave (SW) radiation in the Polar Regions. This gives Arctic cirrus a potentially elevated status in terms of radiative impact on climate. Moreover, cirrus clouds having visual optical depths $\tau_{vis}$ between 0.3 and 3.0 have the greatest frequency of occurrence (Hong and Liu, 2015, JClim), have a CRE representative of cirrus clouds overall (Hong and Liu, 2016, JClim), and appear to be most abundant in the Arctic during winter (DJF; Mitchell et al., 2018). Thus, the CRE of winter Arctic cirrus might be particularly strong, making the topic of this journal submission of interest.

However, this manuscript was written with a focus on SW radiation with LW radiation arguably secondary in importance. While the SW radiation is more interesting in many respects, the uniqueness of Arctic cirrus in terms of LW radiation should not be ignored. In the results section, it might be instructive to show net irradiance for these surface albedo (and cloudy vs. clear) conditions as a function of time over a 24 hour period. Relating TOA Fnet (same as CRE) to solar zenith angle is fine but this focus might detract from the fact that most of the time during Arctic winter the sun is not present and Fnet is determined only by LW radiation. A representative latitude (based on in situ sampling) could be selected for this. This would add perspective for those readers seeking a more representative understanding of Arctic cirrus radiative effects.

The paper is well written and organized and presents results that appear to be unique. After some minor revisions, it should be appropriate for publication in ACP.

*We thank the reviewer for his/her positive assessment of the manuscript and for pointing out some aspects of the cloud radiative effect with regard to the LW/SW regimes that are helpful to improve the core messages of the manuscript. To be completed…*

**Major Comments:**

*To be added.*

1. Figure 9: The results in Fig. 9 (especially 9a) appear to contradict the results in Fig. 17 of Hong and Liu (2015, J. Climate), where Fnet at the surface is comparable with TOA Fnet for the same τvis used here. Please attempt to explain this discrepancy.
2. Lines 352-354: There are evidently some errors in this sentence. The visible optical depths (τvis) for the Jan. and March case studies are 0.65 and 0.68, respectively (line 265) but here it says both τvis are identical. Moreover, τvis = 3 IWP/(ρi De), and multiplying the IWC profiles by a factor of 5 should also increase IWP by this factor, and thus increase τvis by a factor of 5. That being so, the 5-fold τvis stated for these two case studies should be 3.25 and 3.40 (not 2.94 and 2.85 as stated in the text).
3. Lines 379-380: Note this is due only to changes in SW radiation. Please provide an explanation to conceptually understand this. For example, is this due to the greater "effective" optical depth of the cirrus when incident reflected SW radiation enters cloud base at oblique angles?
4. Lines 382-384: But τvis is almost the same for both case studies (0.65 vs. 0.68). Are you sure that a 0.03 change in τvis can account for the shift in the snow albedo curves?

**Technical Comments:**

1. Line 29: trough => through?

   *"Through" is probably bad expression. We change it to "from".*

2. Figure 9: Fig. 8 => Fig. 8a,b?

*Another note by anonymous reviewer #4 was issued in a separate comment as follows:*

Lines 103-104: Regarding the use of the saturation mixing ratio to estimate the gas phase water content (GWC), consider citing Kramer et al. (2009, ACP, Fig. 7; 2020, ACP, Figs. 6, 7 & 9) to defend this assumption (i.e., RHi ~ 100% inside cirrus clouds). That may be a better option than referencing Heller's PhD thesis. However, measurements in Kramer et al. (2009; 2020) are not representative of Arctic cirrus where homogeneous ice nucleation appears more prevalent (suggesting higher RHi); see Mitchell et al. (2018, ACP).

*We overhauled this section according to the comments made by all reviewers. In doing so, we thank the reviewer for pointing out how to further motivate the "average" 100% RHi-assumption using the already mentioned publications by Krämer et al. (2009, 2020). Mitchell et al. (2018) convincingly state that homogeneous ice nucleation is more prevalent at high latitudes (and therefore the occurrence of high RHi). We include a short discussion involving updraft speeds and conclude that while higher ice supersaturation might be probable in the Arctic, also relaxation times of supersaturation are reduced. We also calculate the overall error for IWC in a worst case assumption.*

**References**

*Afchine, A., Rolf, C., Costa, A., Spelten, N., Riese, M., Buchholz, B., et al. (2018). Ice particle sampling from aircraft – influence of the probing position on the ice water content. Atmos. Meas. Tech., 11, 4015–4031.*

*Birner, T. (2010). Residual Circulation and Tropopause Structure. J. Atmos. Sci., 67, 2582–2600.*

*Krämer, M., Schiller, C., Afchine, A., Bauer, R., Gensch, I., Mangold, A., et al. (2009). Ice supersaturations and cirrus cloud crystal numbers. Atmos. Chem. Phys., 9, 3505–3522.*

*Krämer, M., Rolf, C., Luebke, A., Afchine, A., Spelten, N., Costa, A., et al. (2016). A microphysics guide to cirrus clouds – Part 1: Cirrus types. Atmos. Chem. Phys., 16, 3463–3483.*

*Krämer, M., Rolf, C., Spelten, N., Afchine, A., Fahey, D., Jensen, E., et al. (2020). A microphysics guide to cirrus – Part 2: Climatologies of clouds and humidity from observations. Atmos. Chem. Phys., 20, 12569–12608.*

*Manney, G. L., & Lawrence, Z. D. (2016). The major stratospheric final warming in 2016: dispersal of vortex air and termination of Arctic chemical ozone loss. Atmos. Chem. Phys., 16, 15371–15396.*

*Mitchell, D. L., Garnier, A., Pelon, J., & Erfani, E. (2018). CALIPSO (IIR–CALIOP) retrievals of cirrus cloud ice-particle concentrations. Atmos. Chem. Phys., 18, 17325–17354.*

---

## Author Comment (AC2)

**Response to peer reviews**

of the manuscript ACP-2022-395: "Investigating the cloud radiative effect of Arctic cirrus" by Marsing, Meerkötter et al.

*The reviews are written in black. The authors' response to the reviews is given below each paragraph in blue italic font.*

**Review by anonymous reviewer #1**

The authors are to be congratulated on showing the thin ice cloud radiative effects over the polar region using two profiles obtained from aircraft measurements during the Polar Stratosphere in a Changing Climate (POLSTRACC) campaign. I enjoyed reading this manuscript, particularly the instrumentation and data processing section which I am unfamiliar with. Obviously, only two ice water content (IWC) profiles cannot fully describe the cloud variability over the Arctic. However, by making use of the two profiles, the authors study the sensitivies of cloud radiative effects to solar zenith angle and surface albedo variations. The authors also study the difference in computed cloud radiative effects between using IWC aircraft measurement and optically equivalent constant IWC. How the aircraft measurement data are used to prepare the ice cloud description for the radiation computations is nicely presented in detail. All the assumptions made in the radiation computations are clearly stated. The only problem I have is that the computed longwave irradiance variation within the cloud layer looks strange to me and appears to contrast with the results in numerous previous studies. Hence, I would like to suggest the authors double check the longwave radiation computations in this study. Beyond this problem, this manuscript is clear, organized, and well-written and I would suggest it be accepted for publication after some revisions if needed.

*We thank the reviewer for his/her kind words and the positive assessment of the manuscript. The concern regarding our longwave irradiance profiles is addressed in the respective paragraph below.*

My specific comments are as follows:

**Major comments:**

Lines 171-173. Is the air in the (polar) stratosphere generally descending? What is the situation in the troposphere, particularly the upper troposphere that is of interest in this study? Is the air in the troposphere also generally descending? If the upper tropospheric air was also descending during the two months of interest, why do you think the mass accumulation near the tropopause during the two months was a result of the descending motion in the stratosphere? If the steady descending motion in the stratosphere exists and persists, will the static stability of the tropopause become increasingly higher as time goes by?

*In a zonal and multi-annual mean, there is a generally descending motion in the winter and spring polar stratosphere, which has also been observed, e.g. in terms of $N_2O$, in the winter 2015/2016 (Manney and Lawrence, 2016; Birner, 2010). In contrast, tropospheric eddies counteract descending motion (Birner, 2010), which also stabilize the height of the tropopause. Birner also points out how these opposing effects lead to the increasing static stability above the tropopause. We include these references in the manuscript.*

Figure 12. Numerous studies have shown that cloud longwave radiative heating/cooling rate have a vertical gradient from cloud base to cloud top, i.e., heating at the cloud base and cooling at the cloud top (e.g., Fu et al., 1997; Ren et al., 2020, 2021; Wall et al., 2020). However, the red curves in Fig. 12(a) and 12(d) show the opposite, i.e., longwave radiative cooling at the cloud base and heating at the cloud top. I used the longwave version of the rapid radiative transfer model (RRTM; Iacono et al., 2000) with

scattering included (Tang et al., 2018) to do a quick check. The resultant longwave irradiance ($I_{LW}$) and radiative heating/cooling rate ($H_{LW}$) profiles are shown in the figure below:

[Figure]

Figure C1 Longwave irradiance ($I_{LW}$) and radiative heating/cooling rate ($H_{LW}$) profiles in an RRTM experiment.

In this RRTM experiment, a homogeneous ice cloud layer with constant IWC of 0.0032 g m$^{-3}$ is placed between 7.25 and 12 km; a constant effective radius of 23 μm is assumed; a subarctic winter atmospheric profile is adopted with surface temperature set to 280 K and surface emissivity set to 1. Such settings in the RRTM experiment resemble the case of January 25, 2016 in this study. As shown in the above figure, the decrease of $I_{LW}$ with height is slower in the lower portion of the cloud layer, whereas in Fig. 12(a) and 12(d) the decrease of $I_{LW}$ with height is slower in the upper portion of the cloud layer. Would it be helpful to double check your longwave radiation computations? What radiative transfer solver in libRadtran did you use for the longwave radiation computations? Will the $I_{LW}$ result change if you switch to using another solver, such as DISORT?

*We thank the reviewer for the careful investigation of the LW irradiance profile. We suppose that the difference in the radiative transfer calculations results from the used atmospheric temperature profiles. In our original calculations, we tried to resemble the situation at the time of cirrus sampling as closely as possible by using temperature profiles from the ECMWF IFS model (as explained in the manuscript). In Figure A2 in this answer, we show on the left panel how the profile from the 25 January case (black) differs from the standard subarctic winter profile (light blue) or from other synthetic profiles. The right panel gives the corresponding LW irradiance profiles. We observe that our $I_{LW}$ profile in the subarctic case closely resembles that from reviewer #1. We deduce that especially the strong inversion at the tropopause, which is only present in the "real" profile, causes the deviation in the irradiance profile, which naturally propagates into the heating rate profile.*

*The calculation of irradiances and related radiative quantities is now carried out by using the uvspec-solver DISORT (6 streams). We previously assumed that a two-stream solver is sufficient for calculating irradiances. This may be justified in a broad range of model atmospheres, but is, as we learnt, not appropriate in all cases, especially when a high albedo is combined with semitransparent ice clouds. The basic statements of our study are not affected by this, the values of the radiation parameters change somewhat.*

[Figure]

*Figure A2 Left: temperature profiles, i.e. original POLSTACC profile (black) plus further synthetic profiles (red, dark blue) as well as the profile of the subarctic winter standard atmosphere (light blue). Right: corresponding LW irradiance profiles. Curves explain that the pronounced temperature inversion in the POLSTRACC profile produces the significant difference to the irradiance profile calculated by referee #1.*

References

Fu, Q., Liou, K., Cribb, M., Charlock, T., & Grossman, A. (1997). Multiple scattering parameterization in thermal infrared radiative transfer. Journal of the Atmospheric Sciences, 54(24), 2799-2812. https://doi.org/10.1175/1520-0469(1997)0542.0.CO;2

Iacono, M. J., Mlawer, E. J., Clough, S. A., & Morcrette, J. J. (2000). Impact of an improved longwave radiation model, RRTM, on the energy budget and thermodynamic properties of the NCAR community climate model, CCM3. Journal of Geophysical Research: Atmospheres, 105(D11), 14873-14890. https://doi.org/10.1029/2000JD900091

Ren, T., Li, D., Muller, J., & Yang, P. (2021). Sensitivity of radiative flux simulations to ice cloud parameterization over the equatorial western Pacific Ocean region. Journal of the Atmospheric Sciences, 78(8), 2549-2571. https://doi.org/10.1175/JAS-D-21-0017.1

Ren, T., Yang, P., Schumacher, C., Huang, X., & Lin, W. (2020). Impact of cloud longwave scattering on radiative fluxes associated with the Madden-Julian Oscillation in the Indian Ocean and Maritime Continent. Journal of Geophysical Research: Atmospheres, 125(13), e2020JD032591. https://doi.org/10.1029/2020JD032591

Tang, G., Yang, P., Kattawar, G. W., Huang, X., Mlawer, E. J., Baum, B. A., & King, M. D. (2018). Improvement of the simulation of cloud longwave scattering in broadband radiative transfer models. Journal of the Atmospheric Sciences, 75(7), 2217–2233. https://doi.org/10.1175/JAS-D-18-0014.1

Wall, C. J., Norris, J. R., Gasparini, B., Smith Jr, W. L., Thieman, M. M., & Sourdeval, O. (2020). Observational evidence that radiative heating modifies the life cycle of tropical anvil clouds. Journal of Climate, 33(20), 8621-8640. https://doi.org/10.1175/JCLI-D-20-0204.1

**Minor comments:**

Line 29. "trough" or "through"?

*Thanks. We meant "through".*

Line 80. It looks "CIRRUS-HL" is a contraction, but it is not spelled out when it first appears in this article here.

*We set a parenthesis to better express that CIRRUS-HL stands for cirrus in high latitudes. However, the occurrence of this acronym is now moved to the end of the manuscript.*

Lines 102-104. Is water vapor mixing ratio much higher than the saturation mixing ratio in a homogenous ice nucleation environment? I notice that as shown in Fig. 3 the POLSTRACC cirrus measurements were taken at temperatures between 195 and 250 K. Do you know if there are previous studies of the importance of homogenous vs. heterogenous ice nucleation in the polar atmosphere with this temperature range?

*This issue was raised by all reviewers. The homogeneous freezing threshold in the relevant temperature range varies between about 150 – 165 % in terms of relative humidity with respect to ice (RH$_{ice}$) (e.g. Krämer et al., 2016), so there can indeed be considerable super-saturation in a homogeneous nucleation scenario. The prevalence of homogeneous or heterogeneous ice nucleation depends primarily on the updraft speed of the humid air mass (Krämer et al., 2020) associated with the meteorological situation. In polar regions, this includes primarily low and high pressure systems and warm conveyor belts (slow updrafts) and gravity waves and jet streams (fast updrafts). Without further analysis of the flow conditions and history, this may adversely affect the accuracy of the IWC values, which Is what we suppose to be the reviewer's main concern.  We provide resolutions to this in the specific answers to reviewers #3 and #4.*

Line 118. It looks "FISH" is a contraction, but it is not spelled out when it first appears in this article here.

*We include the full name "Fast In-situ Stratospheric Hygrometer" in the text.*

Line 174. Does the "latter" refer to "dynamical tropopause"? Why is the dynamical tropopause a transport barrier of air masses?

*In many cases the dynamical tropopause is a transport barrier, but not necessarily in convection or when radiative heating or cooling takes place as it is the case inside clouds. To avoid confusion, we omit the half sentence "With the latter acting as a transport barrier of air masses".*

Lines 175-176. You mean the observed cirrus clouds above the dynamical tropopause will eventually become polar stratospheric clouds (PSCs)? How are PSCs defined? What are the criteria used to judge whether a cloud is a PSC?

*We rephrase this sentence, as it might be confusing. There are several PSC types, and some consist of water ice or contain a high fraction of water ice. In that sense, it is safe to say that ice clouds above the polar tropopause reflect a lower branch of PSCs.*

Lines 180-182. Is the thermal tropopause the local temperature minimum? However, as shown in Fig. 4, what the authors refer to as "stratospheric cirrus" show the lowest temperatures, making me wonder if these "stratospheric cirrus" clouds are also below the thermal tropopause?

*We clarify upon this by explicitly referring to the WMO tropopause. Accordingly, the thermal tropopause or lapse rate tropopause is the lowest level at which the lapse rate falls below 2 K km$^{-1}$ (with some additional constraints). Therefore, the thermal tropopause is (sometimes considerably) lower (in altitude) than the local temperature minimum.*

Lines 298-299. The agreement between aircraft and reanalysis water vapor data surprises me. This result suggests that the quality of ECMWF IFS analysis water vapor data is very good, at least over Northern Europe.

*We clarify on this a little bit more in the text, mentioning that we compare only exemplary sections of data near our cloud profiles of interest (not within, as we have no humidity data there). Indeed we find exceptional agreement there, but we cannot generalize this observation to a broader domain.*

Lines 355-357. These two sentences read awkward to me. Correct me if I am wrong, basically you wanted to say increased optical thicknesses of ice clouds increase the cloud LW effect (forcing), i.e., making more surface emitted LW radiation absorbed by ice clouds and hence $F_{net,TOA,LW}$ less negative?

*Sentence is changed accordingly.*

Lines 362-363. I cannot understand this sentence "a transition that is shifted towards larger sza when compared to the curves in Figs. 8a, b".

*Paragraph is rewritten.*

Line 419. "shadowing effects"? What do you mean by "shadowing effects"? Didn't you set the cloud fraction to 1 in each cloud layer in your radiation computations? In other words, every cloud layer is overcast in your computations, isn't it? If so, cloud overlapping does not matter in your computations.

*The extinction (optical thickness) due to scattering is higher at altitudes of the IWC maxima when compared to values of the constant IWC profile at the same altitude. Sentence is changed.*

Lines 421-422. Does this sentence talk about the case on March 9, 2016?

*Yes. Date is added to the sentence.*

**Review by anonymous reviewer #2**

**General comments**

This is an interesting study. However, I would think it requires major revisions before it can be considered for publication in ACP. In this paper, IWC profiles of two specific cases are used to investigate the sensitivity of cirrus radiative effects on different parameters using radiative transfer simulations. The resulting dependence of the radiative effects of the cirrus on solar zenith angle and surface albedo is not particularly exciting, this is almost common knowledge. What I found original is the comparison of simulations assuming homogeneous clouds vs. using IWC profiles measured with high vertical resolution. I would consider this the most interesting part of the paper. Therefore, the study could become even more relevant, when additional artificial profiles would be considered. The paper already shows, that measured profiles result in different radiative effects compared to homogeneous cirrus. However, the question is how representative the few measured profiles are. How would the results change if IWC increases/decreases with altitude? This could lead to some more general conclusions of the paper.

*We thank the reviewer for his/her thorough assessment of the manuscript and for the constructive comments and suggestions.*

*A few measured profiles are of course not representative. However, the POLSTRACC measurements in the Arctic reveal fine structures of the IWC and the profiles on 25 January and 09 March differ significantly in shape and geometrical thickness. In comparison with a homogeneous cirrus the results at least indicate which differences are to be expected for individual cases. Note, title has been changed accordingly.*

*Inspired by another comment, it could also be shown that small differences between $F_{net,\ TOA}$ and $F_{net,\ BOA}$, which result for optical ice cloud parameters derived from satellite data and which were averaged over longer time periods (Hong and Liu, 2015), are not necessarily valid for individual ice clouds as observed during POLSTRACC.*

*Following the reviewer's recommendation, we consider further profiles. In view of usually low(er) vertical resolutions of ice cloud parameters in models (climate models) we perform radiative transfer calculations for IWC profiles that approach successively from the homogeneous case to the resolution of the measured IWC profile and present corresponding results in an additional figure. We also performed simulations for simple linear increases and decreases of the IWC, but it is not easy to justify their presentation. The number of possible cases is large and relating such results to any application is difficult.*

**Specific comments**

1. Introduction
    - The title should already indicate, that this is a case study. Otherwise the reader expects more than the paper can deliver.
      *This is something that had also gone through our minds. We agreed and change the title: Investigating the radiative effect of Arctic cirrus measured in situ during the winter 2015/2016*
    - The introduction mixes state of the art with data description and even already conclusions from this study. This is confusing and should be clearly separated.
      *We rearrange paragraphs in the introduction in order to better separate state of the art and data description, and move conclusions to the appropriate section at the end of the manuscript.*
    - Line 15: No numbers are given to quantify the importance of the cirrus radiative effect. Switching from negative to positive is important, but not, if we switch between +- 0.01 W m$^{-2}$
      *This is part of the abstract. We are now a little bit more precise and give numbers for the extreme values of radiative forcing to illustrate the observed range.*
    - Line 32: What means "unavoidable"?
      *We remove the possibly confusing adjectives.*
    - Line 63: Flight hours depend on flight speed and do not reflect the amount and representativeness of the data. Also different instruments sample different volumes during the same time. Horizontal distance, sampled volume, total particle number would be more meaningful.
      *As suggested, we provide a distance, whereas meaningful values on sampled volume and particle number cannot be given. Still, flight hours are a common and interesting measure in the community to get an idea of the extent of a field campaign.*
    - Line 68: What means "thin"? IWP or optical thickness measurements should be quoted here. Or you characterize the general cirrus conditions during the campaign earlier.
      *We agree that the handwavy "thin" characterization is not adequate here. We elaborate on IWP and optical thickness later.*
    - Line 73: This is an example of mixing conclusions into the Introduction. This was not shown yet. Needs to be demonstrated.
      *We rephrase the sentence and move the finding to the conclusions. This should better express what we wanted to say, the intersection with and difference to the Feofilov study.*
    - Lines 79-80: should be part of the conclusion section.
      *We move this to the conclusions.*

2. In situ measurements of IWC
   - Line 86: How do you measure IWC? The first two paragraphs read as if WARAN is measuring water vapor only. First introduce the concept, which allows also to derive IWP from the instrument (heating, evaporation, etc)
     *We adapt the storyline to begin with measured total water content (TWC).*
   - Line 103: This means, that you assume saturated air within the cirrus. Provide a justification, that this assumption is valid in most of the cases.
     *This issue was raised by all reviewers. We acknowledge the lack of a justification why we pursue our study with such a simple assumption. We are well aware that (near) saturation is by far not the only scenario of relative humidity inside cirrus. Therefore, we change the text to explicitly quote this knowledge and explain why the impact on IWC values is less severe than one might think. The reason is the enhancement of ice particles in the sample flow.*
   - Line 110: What is the inlet velocity and how large is it? Is this measured in parallel or is this combining all the enhancement effects?
     *We give a number for the inlet velocity, which is very precisely known using the hygrometer's cell pressure and either a mass flow controller or a critical orifice with known characteristics. However, we assume that explanation of this detail is of lesser interest for the manuscript.*
   - Line 118: Why can FISH serve as a reference? Is it more precise? Why not using FISH data for this study?
     *FISH is not a reference, but another instrument that can provide IWC in a similar way. As an earlier comparison of both measurements showed a systematic bias employing data from an earlier campaign (Afchine et al., 2018), we repeat the comparsion which now shows substantially better agreement. This gives confidence to the accuracy of both instruments, but especially to WARAN. The precision in the studied range of TWC values is comparable, so there is no reason in this respect as to whether one instrument should e preferred over the other.*
3. Statistics from Arctic cirrus sampling
   - Fig. 3: Add the legend from the original publication.
     *We add the original legend.*
   - Line 142 ff: Are measurements in mid-latitudes included in this "Arctic" study? Below in the text it is separated. But I suggest to remove all mid-latitude data from the beginning. The study aims for Arctic cirrus. So there is no need to discuss mid-latitude cirrus in this paper.
     *We would like to keep the overview of mid-latitude cirrus as this dataset has not been shown elsewhere. Also, many important climatological studies of cirrus make a lesser separation along latitudes. As this study is founded in part on this common knowledge, and to give context to the measurements and why they are as they are, we would like to keep them in at this point.*
   - Extending Fig. 3 might also not be the main objective of this study. Consider to focus on the Arctic cirrus.
     *One of the objectives of the POLSTRACC campaign was to extend the in situ data on cirrus in this temperature/latitude parameter space. Therefore, we find it worthwhile to show what could be achieved in this respect, and to give an updated idea of available data.*
   - Line 162: The analysis of Fig. 4 should be better motivated with respect to the main objective of the paper. e.g.: temperature might affect emission - thermal IR radiative effect.

> *We agree and rephrase the motivational sentence. In that, we clarify what might be relevant macrophysical properties of cirrus near the tropopause.*

- Line 166: Explain PVU, please.

  *We give the definition of the potential vorticity unit (PVU).*

4. Complete profiles of cirrus at the high latitude tropopause

- Line 204: These very thin cirrus clouds are common for the Arctic: Really? Only 4 profiles are from Arctic locations.

  *This sentence was misleading. It is rephrased to express that thin cirrus are common for the Arctic (as stated in the literature). The optical thickness of the profiles at hand is assessed elsewhere.*

- Table 1: Leave out the mid-latitude cirrus, give IWC in g m⁻³

  *We would like to keep the mid-latitude profiles in until here, as motivated above. We give the IWC additionally in mg m⁻³. Both representations (molar mixing ratio and mass concentration) have their justifications.*

- Line 210: What do 18 min mean in distance?

  *We give the value which is about 230 km.*

5. Radiative transfer calculations

- First paragraph: Partly a repetition of the introduction. As a reader, I expect a focused and detailed description of what has been done in this section and not another general overview.

  *The description of the paragraph has been further streamlined and now focuses on the content. The investigation of the influence of IWC fine structures has been emphasized.*

- Line 253: The radiative transfer solver is not given in this section. DISORT or two-stream? For SW: UV radiation is neglected? How large is the contribution of UV, which is not covered by the simulations?

  - *We now switched from two-stream to DISORT (6 streams for irradiances). That is the reason why results (numbers) partly deviate from those in the first version of the article.*

  - *Sorry, UV is not neglected. Calculations are based on the wavelength intervals: SW = 0.24 μm – 5.0 μm and LW = 2.5 – 100 μm. The indications of the SW and LW spectral intervals $0.4 – 4.0\,\mu m$ and $4.0 – 100\,\mu m$ in lines 253 and 254 are mistakenly taken, they are not correct here and misleading. Indicated wavelength ranges are changed accordingly.*

- Line 258: More details on this parametrization are needed. What does $R_{eff}$ depend on? Is the Liou parametrization consistent with the parameterization of optical properties by Baum et al?

  - *Unfortunately, $r_{eff}$ and ice crystal shapes have not been measured during POLSTRACC. Liou's parameterizations describes $r_{eff}$ as a function of the IWC by a polynomial fit to observed data which have been collected at Arctic latitudes during the DOE's ARM MPACE experiment at the ARM's North Slope of Alaska site in Fall 2004. The parameterization $r_{eff}(IWC)$ may not exactly be consistent with Baum et al., but we try to realistically limit $r_{eff}$ instead of making completely arbitrary assumptions. We add some sentences about the parameterization and on the origin of the underlying data.*

  - *Note, the profiles $r_{eff}(z)$ in Fig. 6 differ slightly from those in the first version of the manuscript. Reason is that Liou's fit has now been linearly extrapolated for very small values of the IWC, i.e. IWC < 0.0015 g m⁻². This also applies to the values of $F_{net}$ in the tables. However, all results are essentially unchanged.*

- Lines 261-262: Even if there are no in situ measurements of crystal shapes available for the cases investigated here, shape effects should at least be discussed. You may refer to the following two papers that would fit here:
    - For the thermal-infrared: Wendisch, M., P. Yang, and P. Pilewskie, 2007: Effects of ice crystal habit on thermal infrared radiative properties and forcing of cirrus. J. Geophys. Res., 112, D08201, doi:10.1029/2006JD007899
    - For the solar: Wendisch, M., P. Pilewskie, J. Pommier, S. Howard, P. Yang, A. J. Heymsfield, C. G. Schmitt, D. Baumgardner, and B. Mayer, 2005: Impact of cirrus crystal shape on solar spectral irradiance: A case study for subtropical cirrus. J. Geophys. Res., 110, D03202, doi:10.1029/2004JD005294.
- *Thanks for the references. A paragraph is now added (under Discussion) where shape effects, which are not in the focus of this study, are mentioned and where Wendisch et al. (2005, 2007) is cited. Furthermore, the assumption of using the GHM is justified.*
- Line 276: It seems, that broadband albedo data of fixed values are used. In that case, I don't understand, why not simply changing surface albedo in 0.1 steps between 0 and 0.9? Such simulations can be attributed to specific surface types afterwards.
  *When changing the SW albedo in steps of 0.1 and showing all curves, the figures would become confusing. Instead, we reduce the number of curves by showing the results for four surfaces: ocean, fresh and aged snow, and one mixed surface in all Figs. 8-10. This allows the influence of different albedo values to be classified reasonably well.*
- Line 281: Snow and ocean do not reach an emissivity of 1.0. There is no need to make this approximation.
  *Sorry, the number given for the LW emissivity in the text was not quite precise. The assumption is $\varepsilon = 0.99$ for the ocean and the mixed surfaces and $\varepsilon = 1.0$ for both snow surfaces following Wilber et al. (1999). There are certainly values slightly different from our $\varepsilon$ assumptions, but since these deviations are rather small, we have decided not to carry out new radiative transfer calculations for all cases. Numbers are corrected now in the text.*
- Line 286: It is not clear, what temperature and humidity data are finally used for the simulations. HALO, IFS or GEMS? Or did you merge the data?
  *We rephrase some sentences to make clear that we use model (IFS and GEMS) data for everything except from IWC.*
- Line 300: Are the measurements only used to validate the model or are they merged with the model for the final input of the UVSPEC?
  *We use measurements (apart from IWC) only in order to validate the model.*
- Eq. 4: I always thought that net irradiance is the difference between downward and upward irradiance. I understand that you define upward to be negative, this seems odd, at least from a measurement point of view (negative irradiances?).
  *The balance of upward and downward is now defined as the difference $F_{\Delta\lambda} = F_{down,\Delta\lambda} - F_{up,\Delta\lambda}$ in Eq. 4.*
- Line 315: Using symbol "/" for irradiance is uncommon. Now you use "$F$" as a second symbol for the same quantify. Net irradiance has the same unit as the irradiance. I recommend to use only one symbol.
  *We thank the reviewer for discovering this inconsistency. We now write "F" everywhere for (net) irradiance.*
- Line 333: What is the motivation to use different surface albedo for the second case? Hard to compare the impact of IWP of the two clouds, when the albedo is different.
  *Figures are changed, i.e. albedo assumptions for both days, 25 January and 09 February 2016 are now the same in all Figs. 8 – 10. Curves can actually be compared better now.*

- Line 353: Both cirrus have almost the same optical thickness. What is the gain to analyze a second case with similar optical thickness? What was different between both days? Should be motivated when introducing the cases.

  *The two clouds on 25 January and 09 March distinguish in geometrical thickness and in the shape of the IWC profile. Temperature profile together with geometrical thickness and IWC profile shape (symmetric and asymmetric) have an effect on the radiation fields, especially in the LW. This is now mentioned in the introductory sentences of chapter 5.*

- Line 369: To me the conclusion looks different: There is simply no significant solar radiative effect by the cirrus, if the surface albedo is high. Thus, no SZA dependence is visible.

  *Yes, that's correct and $F_{LW, BOA}$ (near sza=88°) is the limiting factor regarding the minimum values of $F_{net, BOA}$. The SW component is then added and depends on sza. For alb = 0.85 the SW contribution is small and shows almost no sza dependence. Explanation is changed accordingly and some sentences are deleted in the text.*

- Fig. 11 very similar to Fig. 6, please combine them.

  *Figs. 11 and 6 are merged into one Fig. 6, i.e. they now contain the T-profile, measured and vertically averaged IWC(z) as well as corresponding parameterized $r_{eff}$(IWC(z)).*

- Line 392: similar discussion for Fig. 6.

  *Corresponding sentences are deleted.*

- Line 393: Temperature profile is not discussed. But this might have an impact on LW radiation.

  *An introductory sentence indicating the influence of the temperature profile (shown in Fig. 6) is added. Discussion of the LW effects below.*

- Line 438: 4.47 k day-1: What is the layer thickness the heating rates are related to? The absolute values strongly depend on this. Without knowing the thickness, the values are not comparable to other studies.

  *As already indicated, the ice clouds on both days are divided into 72 layers, resulting in a vertical resolution of 64 m and 27 m on 25 January and 09 March, respectively.*

- Fig. 12: The heating rate profiles do not seem to fit to the net irradiance profiles. Decrease of net irradiance should always result in a negative heating rate. Or is this only the cloud contribution to the heating rate?

  *Probably a misunderstanding. Please note, that heating rate differences $\Delta H_{net} = H_{net}(IWC_{meas}) - H_{net}(IWC_{const})$ are shown in Figures 12c, f, i, l.*

- Line 446: This should be discussed with the differences between solar and IR wavelength. The heating rates likely are dominated by the IR irradiances. There is no chance, that the surface albedo makes a big difference.

  *Irradiance profiles shown in Fig. 12 are already discussed in section 5.2.4 with regard to the SW and LW contributions. Sentences around line 446 are actually intended to show that the results for the snow surface, which are not shown, do not significantly differ from those for the ocean (thus confirming the reviewer's comment).*

- Line 449: If Fig. A3 is discussed here in the main part of the paper, then the figure should also be placed within the main text. Readers should not be forced to move forward to the appendix while reading.

  *The curves calculated for IWC(z)\*5 are now presented as Figs. 8c, d, 9c, d, and 10 c, d in the main text.*

6. Discussion and outlook
   - Why no conclusions?

     *The section title is misleading. Of course we have conclusions. We rename the section to "Discussion and conclusions" and give an outlook separately afterwards.*

- Discuss, if the measurement strategy/flight pattern affects the conclusions given here. The IWC profiles are not measured at a single location and might be affected by horizontal inhomogeneity of the cirrus. This should be mentioned and discussed here.

  *In the outlook section, we discuss the uncertainty that arises from the non-vertical profiles. However, this cannot be quantified with the available observations. Future studies will make use of a more comprehensive picture of the cirrus cloud field.*

- The effect of the temperature profiles and the location of the cirrus with the profile was not discussed. As the longwave heating is dominating over the solar effect, this is more important than changing surface albedo.

  *Reacting to anonymous reviewer #4, we acknowledge that the LW range significantly contributes to the radiation budget at Arctic latitudes. Therefore, the LW radiation is now given more consideration in the discussion of the results. A different temperature profile would modify the LW effect, as laid out in an answer to anonymous reviewer #1. However, this study aims to reflect a realistic scenario of cirrus within a given atmospheric temperature and trace gas column, so we do not attempt to discuss sensitivities to changes of temperature or (vertical) location here.*

7. Appendix
   - As mentioned above: I would prefer to have the plots in the main text, where they are obviously needed.

     *All plots now appear in the main text. We decided to leave the last table in the appendix, though.*

**Review by anonymous reviewer #3**

This paper uses the measurements of a hygrometer (WARAN) to infer the water contents of cirrus clouds and then based on the inferred cloud water content to quantify the radiative effects of the cirrus in the arctic region. As the authors correctly state, cirrus clouds are frequent in the arctic and potentially play an important role in influencing the radiative balance in the region. However, it is difficult to ascertain their radiative effect because the effect depends not only on the cloud properties, which are difficult to measure, but sensitively on various environment variables such as solar zenith angle and surface albedo which can affect both the magnitude and sign of the radiative effect. I am convinced the topic and objective of this work are both important and think works like this one that base on data to assess the radiative effect of cirrus clouds in the arctic are much needed and should be encouraged. I also find the paper generally well written, providing a clear documentation of the research steps and results.

Although the research is well motivated, I found several critical issues with this work. These include the quantification of the ice water content and the configuration of the environmental profiles for the radiative assessment. These deficiencies limit the usefulness of this work and should be addressed before the paper is considered for publication.

*We thank the reviewer for his/her encouraging words and for the overall positive assessment of the manuscript. The important point regarding the IWC measurements was raised by all reviewers. We acknowledge that this aspect was not well represented in the original manuscript. Concerning the radiative assessment, and taking into account the other reviewers' comments, we decided to expand the study a bit and tried to make it more relevant to the community, as the reviewer understandably demanded. We elaborate on this in the specific answer below.*

**IWC**

Given that IWC is not directly measured but inferred in the total water measurements. The accuracy of the data are especially in need of validation. I found it unsatisfactory to only present a PDF summary (fig 1) of the WARAN vs FISH comparison, without explaining the different behaviour documented here compared to the literature (overestimation of WARAN) or analyzing the biases pattern, e.g., under moist vs. dry conditions, at different times of the day (solar angles), association with underlying surface types (albedo), and collocated dynamical fields.

*We rephrase parts of the IWC measurement section in order to make the measurement process and derivation of IWC more traceable.*

*The instrumental paper by Afchine et al. (2018) comprehensively studies the capabilities, limitations, uncertainties and deviations of the used hygrometers and inlet configuration. We go no further than that. Having said that, we were actually quite surprised that the inter-instrumental comparison between FISH and WARAN could still be notably improved compared to Afchine et al. (2018), by means of some basic improvements in calibration and data treatment. This does not reduce the inherent deficiencies, but justifies the use of the relatively simple WARAN instrument for this study.*

Moreover, it seems the authors completely ignored the possibility of ice supersaturation in inferring the ice content from the total water measurement. Given how common the UTLS air is found to be in a supersaturated state and how the ice and supersaturated air are intrinsically related in influencing the radiation fields (e.g., Tan et al. 2016, https://doi.org/10.1002/2016GL071144), this is not acceptable. It is understood that independent data not available from the campaign, but at minimum this issue should be recognized and discussed, preferably using the statistics of the supersaturation or its relation to environment conditions obtained from other campaigns. In this regard, it appears especially hand-wavy, and possibly wrong, to inflate the IWC by 5 times in the radiative assessment.

*This issue was raised by all reviewers. We acknowledge the lack of a justification why we pursue our study with such a simple assumption. We are well aware that (near) saturation is by far not the only scenario of relative humidity inside cirrus. Therefore, we change the text to explicitly quote this knowledge and explain why the impact on IWC values is less severe than one might think. The reason is the enhancement of ice particles in the sample flow. In that way, we also effectively avoid to mistake supersaturation as ice crystals.*

**Radiative assessment**

The authors correctly recognize that the radiative assessment is sensitive to the environment conditions coexisting with the cirrus, such as the solar angle and surface albedo. However, it doesn't appear logic to me that they extensively use idealized (nominal) values of these parameters rather than best estimates of them from appropriate datasets. Generally speaking, we don't need another set of sensitivity experiments to illustrate how complex the cirrus radiative effects are but are in great need of measurement data to nail down what exact effects are in the nature. The authors need to either provide convincing arguments as to how the sensitivity computations done here are new or useful (how it can be related to nature), or change their strategy and properly pair their cirrus data with the values of those parameters appropriate to the time and location in their assessment.

*Thanks for the constructive comments. As a major change, the radiative transfer solver has now been switched to DISORT, which certainly increases the resilience of our results to the environmental condition (e.g., bright surfaces and clouds, high solar zenith angles).*

*Concerning sensitivity studies (solar zenith angle, albedo), it should be noted that they were intended as an addition to the discussion of the effects associated with measured IWC vertical fine structures and geometrical thickness of ice clouds, as a way to identify the effects of the individual properties, to*

*make the most of the admittedly low amount of data. This way sensitivity studies enable a more extended view on profiles' radiative effects at Arctic latitudes. We select albedo values and sun elevations which are predominantly typical for Arctic regions (open ocean, snow, 60° < sza > 90°). Naturally, by aid of the data measured during one campaign (POLSTRACC 2016) rather indications of specific radiation effects can be given. Results cannot be representative for the entire Arctic. Note, the title of the article has been changed accordingly. However, to date only a few in-situ measurements have been performed inside Arctic ice clouds. So, our results in combination with sensitivity studies close to reality and especially with future measurements may contribute to a more complete picture on how high latitude ice clouds affect radiation fields.*

*Finally, taking up the aspect of motivation, we added a new figure showing how TOA net irradiances change as a function of different vertical resolutions of synthetic profiles which increase towards those of the measured IWC profiles. Such results may for example help to estimate the uncertainties of models with a coarser vertical resolution, as for example climate models.*

Also, these aspects of the assessment probably can be better documented or explained:

The configuration of the RT model, e.g., how many streams are used in the RT solver, how the scattering angles are discretized, … these aspects all affect the results. The sensitivity of cirrus effect to the solar zenith angle is not well explained in the current paper; unclear how the scattering angle effect (forward scattering) and light path effect respectively affect the result and which dominates.

Possibility of sub-cirrus cloud layers, which are often found in nature and are expected to strongly affect the assessment of the radiative impacts of cirrus.

*The calculation of irradiances and related radiative quantities have now been carried out by using the uvspec-solver DISORT (6 streams). We previously assumed that a two-stream solver is sufficient for calculating irradiances. This may be justified in a broad range of model atmospheres, but is, as we learnt, not appropriate in all cases, especially when a high albedo is combined with semitransparent ice clouds. Many thanks for the comment.*

*Scattering effects and sza-dependent light path effects are now described at the beginning of the result chapter.*

*Yes, sub-cirrus cloud layers (water clouds) act as an additional highly reflecting layer. But during the POLSTRACC campaign microphysical or bulk IWC data of low-level clouds have not been measured due to instrumental constraints, as explained in the text. Investigating effects of underlying clouds would mean to start new sensitivity studies based on a range of different model (water) clouds. We think that this should be done in a separate study.*

Optical depth of the aerosol (haze) layer prescribed – how much does it affect the lower boundary reflectance, and how are the cirrus effect depends on this factor.

*Of course, the definition of the albedo ($F_{up}/F_{down}$) implies that it depends on the atmospheric state. However, the reflection properties of the surface are actually described by a spectrally constant and isotropic bidirectional reflection distribution function (BRDF), a parameter independent of atmospheric conditions. Insofar, the sentence at the lines 271-273 and Eq. 5 are actually misleading. An explanatory sentence concerning the BRDF and the use of the term albedo was added and Eq. 5 has been deleted.*

*Combined effects of the "albedo" and ice clouds are already shown. Studying aerosol effects, however, would again mean to start new sensitivity studies which is expensive and not within the scope of this analysis. Furthermore, there have been no measurements on aerosol during the campaign.*

**Review by anonymous reviewer #4**

**General Comments:**

The cloud radiative effect (CRE) of cirrus clouds tends to be strongest in the Polar Regions since cirrus cloud emissivity tends to be greater than the corresponding albedo, and longwave (LW) radiation tends to dominate over shortwave (SW) radiation in the Polar Regions. This gives Arctic cirrus a potentially elevated status in terms of radiative impact on climate. Moreover, cirrus clouds having visual optical depths $\tau_{vis}$ between 0.3 and 3.0 have the greatest frequency of occurrence (Hong and Liu, 2015, JClim), have a CRE representative of cirrus clouds overall (Hong and Liu, 2016, JClim), and appear to be most abundant in the Arctic during winter (DJF; Mitchell et al., 2018). Thus, the CRE of winter Arctic cirrus might be particularly strong, making the topic of this journal submission of interest.

However, this manuscript was written with a focus on SW radiation with LW radiation arguably secondary in importance. While the SW radiation is more interesting in many respects, the uniqueness of Arctic cirrus in terms of LW radiation should not be ignored. In the results section, it might be instructive to show net irradiance for these surface albedo (and cloudy vs. clear) conditions as a function of time over a 24 hour period. Relating TOA $F_{net}$ (same as CRE) to solar zenith angle is fine but this focus might detract from the fact that most of the time during Arctic winter the sun is not present and $F_{net}$ is determined only by LW radiation. A representative latitude (based on in situ sampling) could be selected for this. This would add perspective for those readers seeking a more representative understanding of Arctic cirrus radiative effects.

*It is correct that the LW range significantly contributes to the radiation budget at Arctic latitudes. Therefore, the LW radiation is now given more consideration in the discussion of the results. In Figs. 8, 9, 10 LW results are now added as horizontal lines to ease interpretation. This way the sza-dependent SW contribution can be read implicitly. Furthermore, it is pointed out how the range of a diurnal course of irradiances can be estimated from the figures if locality and day in the year, and herewith sza are known. Concerning Fig. 12 LW results have already been part of the figure.*

The paper is well written and organized and presents results that appear to be unique. After some minor revisions, it should be appropriate for publication in ACP.

*We thank the reviewer for his/her overall positive assessment of the manuscript.*

**Major Comments:**

1. Figure 9: The results in Fig. 9 (especially 9a) appear to contradict the results in Fig. 17 of Hong and Liu (2015, J. Climate), where $F_{net}$ at the surface is comparable with TOA $F_{net}$ for the same $\tau_{vis}$ used here. Please attempt to explain this discrepancy.
   *An interesting aspect, we try to give an explanation:*
   *Briefly introduced as a small side note: Hong and Liu (2015, Fig. 17) compare net forcings at TOA and BOA, not net irradiances ($F_{net}$).*
   *Hong and Liu (2015) state: "… As a demonstration of how cloud forcing depends on cloud properties, we evaluated the ice cloud radiative effects on the earth–atmosphere system using radiative transfer modeling. In this evaluation, ice cloud radiative effects are estimated over the whole ice cloud spectrum using 4-yr-averaged ice cloud properties …"*
   *We, on the other hand, perform radiative transfer calculations based on ice cloud properties stemming from single measurements, not on averaged ice cloud parameters. The IWC profile, especially on 25 January 2016 is rather special, i.e. it is vertically extended (= increased temperature difference between cloud top and cloud base) and shows an asymmetric shape which result in a more pronounced decoupling of the net forcings at TOA and surface (BOA), especially in the LW range. Averaged ice clouds (ice cloud properties), as treated by Hong and*

*Liu, probably result in more symmetric (constant?) IWC profiles and their clouds could also be geometrically thinner (we didn't find a number right away). Our hypothesis may be supported by the following two Figures comparing TOA and BOA net forcing for measured IWC profiles on 25 January 2016 and 09 March 2016.*

[Figure]

*Figure A3: a) For the vertically extended cloud on 25 January 2016, TOA and BOA radiative forcing curves show significant differences. b) For the vertically thinner cloud on 09 March 2016, the curves are closer together.*

*In a next step, calculations could be performed for a cloud with a further reduced geometrical thickness and a vertically constant IWC. It could be expected that TOA and BOA forcings would continue to converge. However, a series of simulations would be required to further support our thesis. This would be interesting, but is outside the focus of our study.*

2. Lines 352-354: There are evidently some errors in this sentence. The visible optical depths ($\tau_{vis}$) for the Jan. and March case studies are 0.65 and 0.68, respectively (line 265) but here it says both $\tau_{vis}$ are identical. Moreover, $\tau_{vis} = 3$ IWP/($\rho_i D_e$), and multiplying the IWC profiles by a factor of 5 should also increase IWP by this factor, and thus increase $\tau_{vis}$ by a factor of 5. That being so, the 5-fold $\tau_{vis}$ stated for these two case studies should be 3.25 and 3.40 (not 2.94 and 2.85 as stated in the text).

   *Yes, there are typing errors in the sentence which have been corrected now. The reason why $\tau_{vis}$ doesn't result in 3.25 and 3.40 is due to the fact that $r_{eff}$ is again adjusted according to the parameterization after Liou (2008) describing $r_{eff}(z)$ as a function of (the new) IWC(z)*5. A corresponding sentence is added. By the way, $\tau_{vis}$ is output of the radiative transfer model.*

3. Lines 379-380: Note this is due only to changes in SW radiation. Please provide an explanation to conceptually understand this. For example, is this due to the greater "effective" optical depth of the cirrus when incident reflected SW radiation enters cloud base at oblique angles?

   *Most of the paragraph is rewritten; interpreting the forcing curves now takes into account both, the SW and the LW components of radiation.*

4. Lines 382-384: But $\tau_{vis}$ is almost the same for both case studies (0.65 vs. 0.68). Are you sure that a 0.03 change in $\tau_{vis}$ can account for the shift in the snow albedo curves?

   *Thanks for the comment. The 0.03 change in optical thickness of course is not responsible for the shift of the curves. Text is rewritten, besides the SW effects of a low ocean albedo at small sza values a longwave emission at higher ice cloud temperatures on Mar 09, 2016 plays a role.*

**Technical Comments:**

1. Line 29: trough => through?

   *Thanks. We meant "through".*

2. Figure 9: Fig. 8 => Fig. 8a,b?

   *Due to the modifications to Figs. 8 and 9, this reference remains valid as it is.*

*Another note by anonymous reviewer #4 was issued in a separate comment as follows:*

Lines 103-104: Regarding the use of the saturation mixing ratio to estimate the gas phase water content (GWC), consider citing Kramer et al. (2009, ACP, Fig. 7; 2020, ACP, Figs. 6, 7 & 9) to defend this assumption (i.e., RHi ~ 100% inside cirrus clouds). That may be a better option than referencing Heller's PhD thesis. However, measurements in Kramer et al. (2009; 2020) are not representative of Arctic cirrus where homogeneous ice nucleation appears more prevalent (suggesting higher RHi); see Mitchell et al. (2018, ACP).

*We overhauled this section according to the comments made by all reviewers. In doing so, we thank the reviewer for pointing out how to further motivate the "average" 100% RHi-assumption using the already mentioned publications by Krämer et al. (2009, 2020). Mitchell et al. (2018) convincingly state that homogeneous ice nucleation is more prevalent at high latitudes (and therefore the occurrence of high RHi). We include a short discussion involving updraft speeds and conclude that while higher ice supersaturation might be probable in the Arctic, also relaxation times of supersaturation are reduced. We also calculate the overall error for IWC in a worst case assumption.*

**References**

*Afchine, A., Rolf, C., Costa, A., Spelten, N., Riese, M., Buchholz, B., et al. (2018). Ice particle sampling from aircraft – influence of the probing position on the ice water content. Atmos. Meas. Tech., 11, 4015–4031.*

*Baum, B. A., Yang, P., Heymsfield, A. J., Platnick, S., King, M. D., Hu, Y.-X., & Bedka, S. T. (2005). Bulk Scattering Properties for the Remote Sensing of Ice Clouds. Part II: Narrowband Models. J. Appl. Meteorol., 44, 1896–1911.*

*Baum, B. A., Yang, P., Nasiri, S., Heidinger, A. K., Heymsfield, A., & Li, J. (2007). Bulk Scattering Properties for the Remote Sensing of Ice Clouds. Part III: High-Resolution Spectral Models from 100 to 3250 cm-1. J. Appl. Meteorol., 46, 423–434.*

*Birner, T. (2010). Residual Circulation and Tropopause Structure. J. Atmos. Sci., 67, 2582–2600.*

*Hong, Y., & Liu, G. (2015). The Characteristics of Ice Cloud Properties Derived from Cloud Sat and CALIPSO Measurements. J. Clim., 28, 3880–3901.*

*Krämer, M., Schiller, C., Afchine, A., Bauer, R., Gensch, I., Mangold, A., et al. (2009). Ice supersaturations and cirrus cloud crystal numbers. Atmos. Chem. Phys., 9, 3505–3522.*

*Krämer, M., Rolf, C., Luebke, A., Afchine, A., Spelten, N., Costa, A., et al. (2016). A microphysics guide to cirrus clouds – Part 1: Cirrus types. Atmos. Chem. Phys., 16, 3463–3483.*

*Krämer, M., Rolf, C., Spelten, N., Afchine, A., Fahey, D., Jensen, E., et al. (2020). A microphysics guide to cirrus – Part 2: Climatologies of clouds and humidity from observations. Atmos. Chem. Phys., 20, 12569–12608.*

*Liou, K. N., Gu, Y., Yue, Q., & McFarguhar, G. (2008). On the correlation between ice water content and ice crystal size and its application to radiative transfer and general circulation models. Geophys. Res. Lett., 35.*

*Manney, G. L., & Lawrence, Z. D. (2016). The major stratospheric final warming in 2016: dispersal of vortex air and termination of Arctic chemical ozone loss. Atmos. Chem. Phys., 16, 15371–15396.*

*Mitchell, D. L., Garnier, A., Pelon, J., & Erfani, E. (2018). CALIPSO (IIR–CALIOP) retrievals of cirrus cloud ice-particle concentrations. Atmos. Chem. Phys., 18, 17325–17354.*

*Wendisch, M., Pilewskie, P., Pommier, J., Howard, S., Yang, P., Heymsfield, A. J., et al. (2005). Impact of cirrus crystal shape on solar spectral irradiance: A case study for subtropical cirrus. J. Geophys. Res., 110.*

*Wendisch, M., Yang, P., & Pilewskie, P. (2007). Effects of ice crystal habit on thermal infrared radiative properties and forcing of cirrus. J. Geophys. Res., 112.*

---

## Referee Report (RR1)

The authors successfully address all my comments in the revision submission. I don't need to see the manuscript again. Below are a few minor suggestions left for the authors to decide if they feel it is good to take. The suggestions may help further polish the manuscript.

Minor suggestions:

Lines 402-404. Yes, the horizontal solid and dashed gray lines in Fig. 9 are above their counterparts in Fig. 8. In other words, the net upwelling longwave irradiance is smaller at the surface than that at the top of the atmosphere. However, the slopes of the colorful curves in Figs. 8 and 9 are not affected by the offsets (or y-intercepts) of the curves. Are the reduced slopes of the colorful curves in Fig. 9 probably due to the higher proportion of the diffusive downwelling shortwave irradiance at the surface than that at the top of the atmosphere? Is the decrease of the diffusive downwelling shortwave irradiance with increasing sza less pronounced than the decrease of the direct downwelling shortwave irradiance with increasing sza?

Lines 424-425. Does the relative importance of surface albedo vs. cloud albedo play a role in the cloud radiative forcing shown in Fig. 10? Is the sign of cloud shortwave radiative forcing predominately determined by whether the cloud albedo is greater or smaller than the surface albedo at a given sza?

Section 5.2.4 and Lines 540-542. In the authors' response to my major comment in the last review, they show that the longwave heating/cooling rate profile is affected by the temperature profile. It appears that a small lapse rate within the cloud layer results in longwave radiative warming at the cloud base and longwave radiative cooling at the cloud top, whereas a large lapse rate within the cloud layer results in longwave radiative warming at the upper portion of the cloud layer and longwave radiative cooling at the lower portion of the cloud layer. For the selected case on January 25, 2016, the lapse rate between 6 km and the tropopause is near the superadiabatic lapse rate. Not sure if such a steep upper tropospheric temperature gradient is common over the Arctic, but this is interesting to me. One of focuses of this study is the influence of ice cloud radiative effect on the thermal stratification. As introduced in numerous previous studies such as those I listed in the last review, cloud base absorption and top emission result in longwave radiative heating at cloud base and cooling at cloud top in general. Hence, the cloud longwave radiative effect generally tends to decrease the static stability of the cloud layer. However, in this study, the authors show that if the lapse rate is near the superadiabatic lapse rate within the cloud layer, longwave radiation heats the upper cloud layer and cools the lower cloud layer and hence tends to increase the static stability. Would it be helpful if such a contrast is emphasized?

---

## Author Response (AR2)

**Minor revision**

of the manuscript ACP-2022-395: "Investigating the radiative effect of Arctic cirrus measured in situ during the winter 2015/2016" by Marsing, Meerkötter et al.

*Dear referees, dear editor,*

*First of all we thank you very much for re-evaluating our changed manuscript, based on your original comments. We are pleased that the revised manuscript now delivers the quality and comprehensiveness that you expect for publication in ACP. In the following we address the remaining comments.*

*Yours sincerely,*

*Andreas Marsing, on behalf of the authors*

**Comments by anonymous referee #1**

The authors successfully address all my comments in the revision submission. I don't need to see the manuscript again. Below are a few minor suggestions left for the authors to decide if they feel it is good to take. The suggestions may help further polish the manuscript.

**Minor suggestions:**

Lines 402-404. Yes, the horizontal solid and dashed gray lines in Fig. 9 are above their counterparts in Fig. 8. In other words, the net upwelling longwave irradiance is smaller at the surface than that at the top of the atmosphere. However, the slopes of the colorful curves in Figs. 8 and 9 are not affected by the offsets (or y-intercepts) of the curves. Are the reduced slopes of the colorful curves in Fig. 9 probably due to the higher proportion of the diffusive downwelling shortwave irradiance at the surface than that at the top of the atmosphere? Is the decrease of the diffusive downwelling shortwave irradiance with increasing sza less pronounced than the decrease of the direct downwelling shortwave irradiance with increasing sza?

*In other words, with decreasing sza, for example for sza < 65°, differences in SW contribution to the net irradiance ($F_{net}$), when comparing the TOA and BOA cases, become increasingly smaller as a result of strong forward scattering of SW radiation at the ice crystals. Considering the extreme case where the ice cloud would represent a "pane of glass" at small sza values, curves for $F_{net, TOA}$ and $F_{net, BOA}$ would coincide at e.g. sza = 0°. Thus, the connecting line of $F_{net}$ values at sza = 90° and at sza = 0° would result in a smaller slope in the BOA case.*

*Accordingly, we added a sentence at the end of line 404: "At the other end of the x-axis at small sza values, the curves of $F_{net, BOA}$ (Fig. 9) and $F_{net, TOA}$ (Fig. 8) tend to converge for the same albedo values, a consequence of increasing forward scattering of incoming SW radiation at the ice crystals. Generally, with decreasing sza an increasing shortwave component … "*

Lines 424-425. Does the relative importance of surface albedo vs. cloud albedo play a role in the cloud radiative forcing shown in Fig. 10? Is the sign of cloud shortwave radiative forcing predominately determined by whether the cloud albedo is greater or smaller than the surface albedo at a given sza?

*Yes, your statement is correct and illustrative. In line 427 we therefore added: "In general it can be said that the sign of $RF_{SW, TOA}$ is predominantly determined by whether the ice cloud albedo is greater or smaller than the surface albedo. Furthermore, ice cloud layers emit …"*

Section 5.2.4 and Lines 540-542. In the authors' response to my major comment in the last review, they show that the longwave heating/cooling rate profile is affected by the temperature profile. It appears that a small lapse rate within the cloud layer results in longwave radiative warming at the cloud base and longwave radiative cooling at the cloud top, whereas a large lapse rate within the cloud layer results in longwave radiative warming at the upper portion of the cloud layer and longwave radiative cooling at the lower portion of the cloud layer. For the selected case on January 25, 2016, the lapse rate between 6 km and the tropopause is near the superadiabatic lapse rate. Not sure if such a steep upper tropospheric temperature gradient is common over the Arctic, but this is interesting to me. One of focuses of this study is the influence of ice cloud radiative effect on the thermal stratification. As introduced in numerous previous studies such as those I listed in the last review, cloud base absorption and top emission result in longwave radiative heating at cloud base and cooling at cloud top in general. Hence, the cloud longwave radiative effect generally tends to decrease the static stability of the cloud layer. However, in this study, the authors show that if the lapse rate is near the superadiabatic lapse rate within the cloud layer, longwave radiation heats the upper cloud layer and cools the lower cloud layer and hence tends to increase the static stability. Would it be helpful if such a contrast is emphasized?

*This is a very interesting observation, thank you for pointing this out! Indeed we observe rater large lapse rates ($\gtrsim 8\ K\ km^{-1}$) within the cloud layer in many of our profiles. As stated this might not be representative, but seems worth mentioning. Although this is not a purely cloud-induced effect, it is amplified by the presence of the cloud. There we added in line 547: "The strong tropospheric lapse rate in some of our observed cloud profiles induces a stabilizing radiative effect of the cloud in most of the tropospheric heating rate profile inside the cloud layer, with an overall higher (LW) heating rate at the cloud top (warming) than at the cloud bottom (less warming or cooling)."*

**Comments by Yi Huang (referee #3)**

The authors well addressed my comments, especially those on IWC uncertainty and radiative transfer modelling. I appreciate their efforts and am glad to see that the quality of the paper noticeably increased and to find some new/revised results quite interesting, such as the dependence on profile resolution. I think the paper is largely ready to publish as it is. I'd appreciate it if they could address the following further comments - these should be considered minor (non-obligatory) suggestions:

1) separate fig 12 to LW and SW components, to identify which component accounts (more) for the variation;
   *Although an interesting suggestion, we would like to not separate LW and SW components in this section and leave their treatment to future work, which should then include a more detailed discussion of how the artificial cloud layers are set up.*
2) add a column in fig 11 to show the heating rate itself (besides the difference), so that the difference can be appreciated in a context;
   *We agree that showing only the heating rate differences might be unsatisfactory and added another column with the absolute heating rates.*
3) The 5x perturbation experiment of fig 13 looks very interesting in that the effect is shown to not simply scale with IWC. It's however a quite lonely result concerning this dependency (total ice amount). Why not include irradiance results as well and examine it over more values of IWP (like fig 8/9 for SZA dependency)? I am particularly interested in whether the sign change of cloud forcing as reported by Tan et al. (2016, doi:10.1002/2016GL071144, their fig 3j, explained in fig S4) can also happen in the Arctic cirrus case.
   *Studying the sensitivity of the radiative effect on (artificially) varying IWP certainly sounds interesting to us. The 5x perturbation was included to estimate the range that might appear if the measurements were severely affected by undersampling of ice particles (as explained).*

*However, we wanted (and were advised by reviewers) to keep the study as close as possible to the observations, and no overdo with sensitivity results. The few cases also do not fully represent the range of optical thicknesses of Arctic cirrus. Still, a hint on changing signs of cloud radiative forcing can be found in Fig. 10, comparing panels (a) vs. (c) or (b) vs. (d). In our view, a more detailed study on the IWP/optical thickness effect should be based on further observations.*

**Comment by anonymous referee #4**

The authors have satisfactorily replied to all of my review comments/questions and have revised the manuscript accordingly. I have no further comments, other than to consider revising the scaling of tick marks on the upper x-axis in Fig. 6. Major tick marks are in increments of .0025 g/m3 for IWC, but minor tick marks are in increments of .000625 g/m3 (5 tick marks would give increments of .0005 g/m3).

*Yes, scaling of tick marks on the upper x-axis in Fig. 6 should be revised. This has been done.*

**Own changes**

*We adjusted the colouring or added symbols to the data in Figures 7, 8, 9, 10, 11 and 12 to better respect the needs of people with colour blindness handicap.*

*Regardless of the reviewers' comments, we would finally like to make a small change to Figure 7. In Figure 7 the previous red curve should be omitted since by mistake it is still based on the two-stream and not on DISORT calculations. Furthermore, data from the red curve is not used in the radiative transfer calculations and it is not discussed in the text, neither is it planned anymore. The new Figure 7 shows the data which are actually used in our model simulations. This implies a small change also in the corresponding figure caption.*